# Learning Gaussian Mixtures with Generalised Linear Models: Precise Asymptotics in High-dimensions

**Bruno Loureiro**
IdePHICS, EPFL, Lausanne

**Gabriele Sicuro**
Department of Mathematics
King's College London

**Cédric Gerbelot**
Laboratoire de Physique de l'École Normale Supérieure

**Alessandro Pacco**
IdePHICS, EPFL, Lausanne

**Florent Krzakala**
IdePHICS, EPFL, Lausanne

**Lenka Zdeborová**
SPOC, EPFL, Lausanne

## Abstract

Generalised linear models for multi-class classification problems are one of the fundamental building blocks of modern machine learning tasks. In this manuscript, we characterise the learning of a mixture of $K$ Gaussians with generic means and covariances via empirical risk minimisation (ERM) with any convex loss and regularisation. In particular, we prove exact asymptotics characterising the ERM estimator in high-dimensions, extending several previous results about Gaussian mixture classification in the literature. We exemplify our result in two tasks of interest in statistical learning: a) classification for a mixture with sparse means, where we study the efficiency of $\ell_1$ penalty with respect to $\ell_2$; b) max-margin multi-class classification, where we characterise the phase transition on the existence of the multi-class logistic maximum likelihood estimator for $K > 2$. Finally, we discuss how our theory can be applied beyond the scope of synthetic data, showing that in different cases Gaussian mixtures capture closely the learning curve of classification tasks in real data sets.

## 1 Introduction

A recurring observation in modern deep learning practice is that neural networks often defy the standard wisdom of classical statistical theory. For instance, deep neural networks typically achieve good generalisation performances at a regime in which it interpolates the data, a fact at odds with the intuitive bias-variance trade-off picture stemming from classical theory [1–3]. Surprisingly, many of the "exotic" behaviours encountered in deep neural networks have recently been shown to be shared by models as simple as overparametrised linear classifiers [4, 5], e.g., the aforementioned benign over-fitting [6]. Therefore, understanding the generalisation properties of simple models in high-dimensions has proven to be a fertile ground for elucidating some of the challenging statistical questions posed by modern machine learning practice [7–16].

In this manuscript, we pursue this enterprise in the context of a commonly used model for high-dimensional classification problems: the Gaussian mixture. Indeed, it has been recently argued that the features learned by deep neural networks trained on the cross-entropy loss "collapse" in a mixture of well-separated clusters, with the last layer acting as a simple linear classifier [17]. Another observation put forward in [18] is that data obtained using generative adversarial networks behave as Gaussian mixtures. Here, we derive an exact asymptotic formula characterising the performance

35th Conference on Neural Information Processing Systems (NeurIPS 2021).

of generalised linear classifiers trained on $K$ Gaussian clusters with generic covariances and means. Our formula is valid for any convex loss and penalty, encompassing popular tasks in the machine learning literature such as ridge regression, basis pursuit, cross-entropy minimisation and max-margin estimation. This allow us to answer relevant questions for statistical learning, such as: what is the separability threshold for $K$-clustered data? How does regularisation affects estimation? Can different penalties help when the means are sparse? We also extend the observation of [18] showing that the learning curves of binary classification tasks on *real data* are indeed well captured by our asymptotic analysis.

**Model definition —** We consider learning from a $d$-dimensional mixture of $K$ Gaussian clusters $\mathcal{C}_{k \in [K]}$. The data set is obtained by sampling $n$ pairs $(\boldsymbol{x}^\nu, \boldsymbol{y}^\nu)_{\nu \in [n]} \in \mathbb{R}^{d+K}$ identically and independently. We adopt the one-hot encoded representation of the labels, i.e., if $\boldsymbol{x}^\nu \in \mathcal{C}_k$, then $\boldsymbol{y}^\nu = \boldsymbol{e}_k$, $k$th basis vector of $\mathbb{R}^K$. We will denote the matrix of concatenated samples $\boldsymbol{X} \in \mathbb{R}^{d \times n}$. The mixture density then reads:

$$P(\boldsymbol{x}, \boldsymbol{y}) = \sum_{k=1}^{K} y_k \rho_k \mathcal{N}\left(\boldsymbol{x} \,|\, \boldsymbol{\mu}_k, \boldsymbol{\Sigma}_k\right), \tag{1}$$

where $\mathcal{N}(\boldsymbol{x}|\boldsymbol{\mu}, \boldsymbol{\Sigma})$ is the multivariate normal distribution with mean $\boldsymbol{\mu}$ and covariance matrix $\boldsymbol{\Sigma}$. The matrix of concatenated means is denoted $\boldsymbol{M} \in \mathbb{R}^{d \times K}$. In Eq. (1), $\forall k$, $\rho_k = P(\boldsymbol{y} = \boldsymbol{e}_k) \geq 0$, $\boldsymbol{\mu}_k \in \mathbb{R}^d$ and $\boldsymbol{\Sigma}_k \in \mathbb{R}^{d \times d}$ is positive-definite. We will consider the estimator obtained by minimising the following empirical risk:

$$\mathcal{R}(\boldsymbol{W}, \boldsymbol{b}) \equiv \sum_{\nu=1}^{n} \ell\left(\boldsymbol{y}^\nu, \frac{\boldsymbol{W} \boldsymbol{x}^\nu}{\sqrt{d}} + \boldsymbol{b}\right) + \lambda r(\boldsymbol{W}), \tag{2}$$

$$(\boldsymbol{W}^\star, \boldsymbol{b}^\star) \equiv \underset{\boldsymbol{W} \in \mathbb{R}^{K \times d}, \boldsymbol{b} \in \mathbb{R}^K}{\operatorname{argmin}} \mathcal{R}(\boldsymbol{W}, \boldsymbol{b}), \tag{3}$$

where $\boldsymbol{W} \in \mathbb{R}^{K \times d}$ and $\boldsymbol{b} \in \mathbb{R}^K$ are the weights and bias to be learned, $\ell$ is a convex loss function, and $r$ is a regularisation function whose strength is tuned by the parameter $\lambda \geq 0$. For example the loss function $\ell$ can represent the composition of a cross-entropy loss with a softmax thresholding on the linear part of Eq. (2). We will characterise the distribution of the estimator $(\boldsymbol{W}^\star, \boldsymbol{b}^\star)$, and we will evaluate the average training loss defined as

$$\epsilon_\ell = \frac{1}{n} \sum_{\nu=1}^{n} \ell\left(\boldsymbol{y}^\nu, \frac{\boldsymbol{W}^\star \boldsymbol{x}^\nu}{\sqrt{d}} + \boldsymbol{b}^\star\right), \tag{4}$$

as well as the average training error $\epsilon_t$ and generalisation error $\epsilon_g$, defined as the misclassification rates:

$$\epsilon_t = \frac{1}{n} \sum_{\nu=1}^{n} \mathbb{I}\left[\boldsymbol{y}^\nu \neq \hat{\boldsymbol{y}}\left(\frac{\boldsymbol{W}^\star \boldsymbol{x}^\nu}{\sqrt{d}} + \boldsymbol{b}^\star\right)\right], \quad \epsilon_g = \mathbb{E}_{(\boldsymbol{x}^{\text{new}}, \boldsymbol{y}^{\text{new}})}\left[\mathbb{I}\left[\boldsymbol{y}^{\text{new}} \neq \hat{\boldsymbol{y}}\left(\frac{\boldsymbol{W}^\star \boldsymbol{x}^{\text{new}}}{\sqrt{d}} + \boldsymbol{b}^\star\right)\right]\right],$$

where $(\boldsymbol{x}^{\text{new}}, \boldsymbol{y}^{\text{new}})$ is a new unseen data point sampled from the distribution in Eq. (1). In the previous equations, we have used the function $\hat{\boldsymbol{y}} : \mathbb{R}^K \to \mathbb{R}^K$, so that $\hat{y}_k(\boldsymbol{x}) := \mathbb{I}(\max_\kappa x_\kappa = x_k)$.

The **main contributions** in this manuscript are the following:

**(C1)** In Sec. 2 and Appendix A we prove closed-form equations characterizing the asymptotic distribution of the matrix of weights $\boldsymbol{W}^\star \in \mathbb{R}^{K \times d}$, enabling the exact computation of key quantities such as the training and generalisation error. Our proof method solves shortcomings of previous approaches by introducing a novel approximate message-passing sequence, building on recent advances in this framework, that is of independent interest.

**(C2)** In Sec. 3.1 we study the problem of classifying an anisotropic mixture with sparse means, where the strong or weak directions in the data are correlated with the non-zero components of the mean as in [19]. We study how learning the sparsity with an $\ell_1$ penalty improves the classification performance.

**(C3)** In Sec. 3.2 we study the performance of the cross-entropy estimator in the limit of vanishing regularisation $\lambda \to 0^+$ for $K$ Gaussian clusters as a function of the sample complexity $\alpha = {}^n/_d$; we show that a phase transition takes place at a certain value $\alpha_K^\star$ between a regime of complete separability of the data and a regime in which the correct classification of almost all points in the data set is not possible. We also investigate the effect of $\lambda > 0$ regularisation on the generalisation error, comparing the $K > 2$ case with the results given in the literature for $K = 2$ [14, 20].

**(C4)** In Sec. 3.3 we investigate the applicability of our formula beyond the Gaussian assumption by applying it to classification tasks on *real data*. We show that for different tasks and losses, it closely captures the real learning curves, even when data is mapped through a non-linear feature map. This further shows that Gaussian mixtures are a good surrogate model for investigating real classification tasks, as put forward in [18].

**Relation to previous work —** The analysis of Gaussian mixture models in the high-dimensional regime has been the subject of many recent works. Exact asymptotics has been derived for the binary classification case with diagonal covariances in [21–23] for the logistic loss and in [24, 25] for the square loss, both with $\ell_2$ penalty. A similar analysis has been performed in [26] for the hard-margin SVM. These works were generalised to generic convex losses and $\ell_2$ penalty in [14], where it has been also shown that the regularisation term can play an important role in reaching Bayes-optimal performances. Hinge regression with $\ell_1$ penalty and diagonal covariance was treated in [13]. Recently, these asymptotic results were generalised to the case in which both clusters share the same covariance in [27], and finite rate bounds were given in [28, 29] in the case of sub-Gaussian mixtures. Asymptotic results for the multiclass problem with diagonal covariance were derived in [20] for the restricted case of the square loss with $\ell_2$ penalty. Our result unifies all the aforementioned asymptotic formulas, and extends them to the general case of a multiclass problem with generic covariances and arbitrary convex losses and penalties.
From a technical standpoint, in [13, 14, 20, 21, 25, 27, 30] the authors use convex Gaussian comparison inequalities, see e.g. [31, 32], to prove their result. In particular, the proof given in [20] for the multiclass problem harnesses the geometry of least-squares, and it is then stressed that this method breaks down for multiclass problems in which the risk does not factorise over the $K$ clusters (as for the cross-entropy, for example). We solve this problem using an innovative proof technique which has an interest in its own. Our approach is to capture the effect of non-linearity and generic covariances via the rigorous study of an approximate message-passing (AMP) sequence, a family of iterations that admit closed-form asymptotics at each step called *state evolution equations* [33]. Our proof relies on several refinements of AMP methods to handle the full complexity of the problem, notably spatial coupling with matrix valued variables [34–36] and non-separable update functions [37], via a multi-layer approach to AMP [38].
The sparse Gaussian mixture model analysed in Section 3.1 is closely related to the rare/weak features model introduced in [19] and widely studied in the context of sparse linear discriminant analysis [39–42]. It was recently revisited in [28, 29] in the context of ERM with max-margin classifiers. Here, we consider a correlated variation of the model and study the benefit of using a sparsity inducing $\ell_1$ penalty.
The separability transition is a classical topic [43, 44] that has recently witnessed a renewal of interest thanks to its connection to overparametrization. It was studied in [16] in the context of uncorrelated Gaussian data, in [8] in the random features model and in [14, 21] for binary Gaussian mixtures.
Recently, [12, 45, 46] showed that the performance of different regression tasks on real data are well-captured by a teacher-student Gaussian model in high-dimensions for ridge regression, but this turned not to be true for non-linear problems such as logistic classification [12]. Authors of [18] showed instead that data from generative adversarial networks behave like Gaussian mixtures, motivating the modeling of such mixture for real-data in the present paper.

## 2 Technical results

Our main technical result is an exact asymptotic characterization of the distribution of the estimator $W^\star$. Informally, the estimator $W^\star$ and the quantity $W^\star X/\sqrt{d}$ behave asymptotically as non-linear transforms of multivariate Gaussian distributions. These transforms are directly linked to the proximal operators [47, 48] associated to the loss and regularisation functions, summarizing the effect of the cost function landscape on the estimator. The parameters of these Gaussian distributions and proximals can then be computed from the fixed point of a self-contained set of equations. We start by presenting the most generic form of our result in a concentration of measure-like statement in Theorem 1, and discuss an intuitive interpretation of the different quantities involved. Theorem 2 then states how the training and generalisation errors can be computed. All results presented in the experiments section can be obtained from Theorem 1. In Corollary 3 we discuss a particular case where explicit simplifications can be obtained. But first, let's summarise the required assumptions for our result to hold.

**(A1)** The functions $\ell$ (as a function of its second argument) and $r$ are proper, closed, lower semi-continuous convex functions. We assume additionally that either the cost function $\ell(\boldsymbol{y}, \bullet \boldsymbol{X}) + r(\bullet)$ is strictly convex, or that $\ell(\boldsymbol{y}, \bullet)$ is strictly convex in its second argument and $r$ is the $\ell_1$ norm, and that the optimization problem 2 is feasible.

**(A2)** The covariance matrices are positive definite and their spectral norms are bounded.

**(A3)** The mean vectors $\boldsymbol{\mu}_k$ are distributed according to some density $P_{\boldsymbol{\mu}}(\boldsymbol{M})$ such that the following quantity is finite

$$\forall d \qquad \mathbb{E}\left[\left\|\boldsymbol{M}^\top \boldsymbol{M}\right\|_{\mathrm{F}}\right] < +\infty, \tag{5}$$

where $\|\bullet\|_{\mathrm{F}}$ denotes the Frobenius norm.

**(A4)** The number of samples $n$ and dimension $d$ both go to infinity with fixed ratio $\alpha = n/d$, called hereafter the sample complexity. The number of clusters $K$ is finite.

**(A5)** The fixed point of the set of self-consistent equations Eq.(8) exists and is unique.

As specified by assumption **(A1)**, our proof does not apply to any convex problem. We discuss this assumption further in Appendix A.5. We also comment on the existence and uniqueness of the solution to the set of self consistent equations Eq.(8) in Appendix A.6. Before proceeding further, let us specify a useful notation. Suppose that the matrix $\boldsymbol{G} = (G_{ki})_{ki} \in \mathbb{R}^{K \times d}$ is given, alongside the four-index tensor $\mathbf{A} = (A_{ki\,k'i'})_{ki\,k'i'} \in \mathbb{R}^{K \times d} \otimes \mathbb{R}^{K \times d}$. We will use the notation $\boldsymbol{G} \odot \mathbf{A} = \sum_{ki} G_{ki} A_{ki\,k'i'} \in \mathbb{R}^{K \times d}$. Similarly, given a four-index tensor $\mathbf{A}$, we will define $\sqrt{\mathbf{A}}$ as the tensor such that $\mathbf{A} = \sqrt{\mathbf{A}} \odot \sqrt{\mathbf{A}}$. We are now in a position to state our main result.

**Theorem 1** (Concentration properties of the estimator). *Let $\boldsymbol{\xi}_{k \in [K]} \sim \mathcal{N}(\mathbf{0}, \mathbf{I}_K)$ be collection of $K$-dimensional standard normal vectors independent of other quantities. Let also be $\{\boldsymbol{\Xi}_k\}$ a set of $K$ matrices, $\boldsymbol{\Xi}_k \in \mathbb{R}^{K \times d}$, with i.i.d. standard normal entries, independent of other quantities. Under the set of assumptions (A1–A5), for any pseudo-Lispchitz functions of finite order $\phi_1 : \mathbb{R}^{K \times d} \to \mathbb{R}, \phi_2 : \mathbb{R}^{K \times n} \to \mathbb{R}$, the estimator $\boldsymbol{W}^\star$ and the matrix $\boldsymbol{Z}^\star = \frac{1}{\sqrt{d}} \boldsymbol{W}^\star \boldsymbol{X}$ verify:*

$$\phi_1(\boldsymbol{W}^\star) \xrightarrow[n,d \to +\infty]{P} \mathbb{E}_{\boldsymbol{\Xi}}\left[\phi_1(\boldsymbol{G})\right], \qquad \phi_2(\boldsymbol{Z}^\star) \xrightarrow[n,d \to +\infty]{P} \mathbb{E}_{\boldsymbol{\xi}}\left[\phi_2(\boldsymbol{H})\right], \tag{6}$$

*where we have introduced the proximal for the loss:*

$$\boldsymbol{h}_k = \boldsymbol{V}_k^{1/2} \operatorname{Prox}_{\ell(\boldsymbol{e}_k, \boldsymbol{V}_k^{1/2}\bullet)}(\boldsymbol{V}_k^{-1/2}\boldsymbol{\omega}_k) \in \mathbb{R}^K, \qquad \boldsymbol{\omega}_k \equiv \boldsymbol{m}_k + \boldsymbol{b} + \boldsymbol{Q}_k^{1/2}\boldsymbol{\xi}_k, \tag{7}$$

*and $\boldsymbol{H} \in \mathbb{R}^{K \times n}$ is obtained by concatenating each $\boldsymbol{h}_k$, $\rho_k n$ times. We have also introduced the matrix proximal $\boldsymbol{G} \in \mathbb{R}^{K \times d}$:*

$$\boldsymbol{G} = \mathbf{A}^{\frac{1}{2}} \odot \operatorname{Prox}_{r(\mathbf{A}^{\frac{1}{2}}\odot\bullet)}(\mathbf{A}^{\frac{1}{2}} \odot \boldsymbol{B}), \quad \mathbf{A}^{-1} \equiv \sum_k \hat{\boldsymbol{V}}_k \otimes \boldsymbol{\Sigma}_k, \quad \boldsymbol{B} \equiv \sum_k \left(\boldsymbol{\mu}_k \hat{\boldsymbol{m}}_k^\top + \boldsymbol{\Xi}_k \odot \sqrt{\hat{\boldsymbol{Q}}_k \otimes \boldsymbol{\Sigma}_k}\right).$$

*The collection of parameters $(\boldsymbol{Q}_k, \boldsymbol{m}_k, \boldsymbol{V}_k, \hat{\boldsymbol{Q}}_k, \hat{\boldsymbol{m}}_k, \hat{\boldsymbol{V}}_k)_{k \in [K]}$ is given by the fixed point of the following self-consistent equations:*

$$\begin{cases} \boldsymbol{Q}_k = \frac{1}{d}\mathbb{E}_{\boldsymbol{\Xi}}[\boldsymbol{G}\boldsymbol{\Sigma}_k\boldsymbol{G}^\top] \\ \boldsymbol{m}_k = \frac{1}{\sqrt{d}}\mathbb{E}_{\boldsymbol{\Xi}}[\boldsymbol{G}\boldsymbol{\mu}_k] \\ \boldsymbol{V}_k = \frac{1}{d}\mathbb{E}_{\boldsymbol{\Xi}}\left[\left(\boldsymbol{G} \odot \left(\hat{\boldsymbol{Q}}_k \otimes \boldsymbol{\Sigma}_k\right)^{-\frac{1}{2}} \odot (\mathbf{I}_K \otimes \boldsymbol{\Sigma}_k)\right)\boldsymbol{\Xi}_k^\top\right] \end{cases} \quad \begin{cases} \hat{\boldsymbol{Q}}_k = \alpha\rho_k\mathbb{E}_{\boldsymbol{\xi}}\left[\boldsymbol{f}_k\boldsymbol{f}_k^\top\right] \\ \hat{\boldsymbol{V}}_k = -\alpha\rho_k\boldsymbol{Q}_k^{-\frac{1}{2}}\mathbb{E}_{\boldsymbol{\xi}}\left[\boldsymbol{f}_k\boldsymbol{\xi}^\top\right] \\ \hat{\boldsymbol{m}}_k = \alpha\rho_k\mathbb{E}_{\boldsymbol{\xi}}\left[\boldsymbol{f}_k\right] \end{cases} \tag{8}$$

*where $\boldsymbol{f}_k \equiv \boldsymbol{V}_k^{-1}(\boldsymbol{h}_k - \boldsymbol{\omega}_k)$, and the vector $\boldsymbol{b}^\star$ is such that $\sum_k \rho_k\mathbb{E}_{\boldsymbol{\xi}}\left[\boldsymbol{V}_k\boldsymbol{f}_k\right] = \mathbf{0}$ holds.*

The purpose of this statement is to have an asymptotically exact description of the distribution of the estimator, where the dimensions going to infinity are effectively summarized as averages over simple, independent distributions. Those distributions are parametrised by the set of finite-size parameters $(\boldsymbol{Q}_k, \boldsymbol{m}_k, \boldsymbol{V}_k, \hat{\boldsymbol{Q}}_k, \hat{\boldsymbol{m}}_k, \hat{\boldsymbol{V}}_k)_{k \in [K]}$ that can be exactly evaluated and have a clear interpretation. Indeed, the parameters $(\boldsymbol{m}_k, \hat{\boldsymbol{m}}_k)$ and $(\boldsymbol{Q}_k, \hat{\boldsymbol{Q}}_k)$ respectively represent means and covariances of multivariate Gaussians (combined with the original $\boldsymbol{\mu}_k, \boldsymbol{\Sigma}_k$), and the $(\boldsymbol{V}_k, \hat{\boldsymbol{V}}_k)$ parametrise the deformations that should be applied to these Gaussians to obtain the distribution of $\boldsymbol{W}^\star, \boldsymbol{Z}^\star$. The distribution is characterized in a weak sense with concentration of pseudo-Lipschitz (i.e., sufficiently regular) functions, whose definition is reminded in the Appendix A. From this result one can work out a

number of properties of the weights $\boldsymbol{W}^\star$, e.g., training and generalisation error, but also hypothesis tests as done in [49] for the LASSO. Due to the generality of the statement, no direct simplification is possible. However, we will see that in certain specific cases all quantities can be greatly simplified. This is notably the case for diagonal covariance matrices and separable estimators and observables $\phi_1, \phi_2$, where the sums over high-dimensional Gaussians concentrate explicitly to one-dimensional expectations. For instance the results of [14, 20] can be recovered as special cases of the present work. Theorem 1 then allows to obtain the asymptotic values of the generalisation error, of the training loss and of the training error. Their explicit expression is given in the following Theorem.

**Theorem 2** (generalisation error and training loss). *In the hypotheses of Theorem 1, the training loss, the training error and the generalisation error are given by*

$$\epsilon_\ell = \sum_{k=1}^K \rho_k \mathbb{E}_{\boldsymbol{\xi}}[\ell(\boldsymbol{e}_k, \boldsymbol{h}_k)], \qquad \epsilon_t = 1 - \sum_{k=1}^K \rho_k \mathbb{E}_{\boldsymbol{\xi}}\left[\hat{y}_k(\boldsymbol{h}_k)\right], \qquad \epsilon_g = 1 - \sum_{k=1}^K \rho_k \mathbb{E}_{\boldsymbol{\xi}}\left[\hat{y}_k(\boldsymbol{\omega}_k)\right]. \quad (9)$$

**The case of ridge regularisation and diagonal $\boldsymbol{\Sigma}_k$** — The general formulas given above can be remarkably simplified under some assumptions about the choice of the regularisation and about the structure of the covariance matrices $\boldsymbol{\Sigma}_k$. This is the case for instance for the ridge regularisation $r(\boldsymbol{W}) = \|\boldsymbol{W}\|_{\mathrm{F}}^2/2$ and jointly diagonalizable covariances. In this case, Theorem 1 simplifies as follows.

**Corollary 3.** *Under the hypotheses of Theorem 1, let us further assume that a ridge regularisation is adopted, $r(\boldsymbol{W}) = \|\boldsymbol{W}\|_{\mathrm{F}}^2/2$, and that the covariance matrices $\boldsymbol{\Sigma}_k$ have a common set of orthonormal eigenvectors $\{\boldsymbol{v}_i\}_{i=1}^d$, so that, for each $\boldsymbol{\Sigma}_k = \sum_{i=1}^d \sigma_i^k \boldsymbol{v}_i \boldsymbol{v}_i^\top$. Let us also introduce, in the $d \to +\infty$ limit, the joint distribution for the $K$-dimensional vectors $\boldsymbol{\sigma} = (\sigma^1, \ldots, \sigma^K)$ and $\boldsymbol{\mu} = (\mu^1, \ldots, \mu^K)$,*

$$\frac{1}{d} \sum_{i=1}^d \prod_{k=1}^K \delta(\sigma^k - \sigma_i^k)\delta(\mu^k - \sqrt{d}\boldsymbol{\mu}_k^\top \boldsymbol{v}_i) \xrightarrow{d \to +\infty} p(\boldsymbol{\sigma}, \boldsymbol{\mu}), \quad (10)$$

*Then, the first three saddle point equations in eqs. (8) take the form*

$$\begin{cases} \boldsymbol{Q}_k = \mathbb{E}_{\boldsymbol{\sigma},\boldsymbol{\mu}}\left[\sigma^k \left(\lambda \mathbf{I}_K + \sum_{\kappa=1}^K \sigma^\kappa \hat{\boldsymbol{V}}_k\right)^{-2} \left(\sum_{\kappa\kappa'} \mu^\kappa \mu^{\kappa'} \hat{\boldsymbol{m}}_\kappa \hat{\boldsymbol{m}}_{\kappa'}^\top + \sum_{\kappa=1}^K \sigma^\kappa \hat{\boldsymbol{Q}}_k\right)\right], \\ \boldsymbol{m}_k = \mathbb{E}_{\boldsymbol{\sigma},\boldsymbol{\mu}}\left[\mu^k \left(\lambda \mathbf{I}_K + \sum_{\kappa=1}^K \sigma^\kappa \hat{\boldsymbol{V}}_k\right)^{-1} \sum_{\kappa=1}^K \mu^\kappa \hat{\boldsymbol{m}}_\kappa\right], \\ \boldsymbol{V}_k = \mathbb{E}_{\boldsymbol{\sigma},\boldsymbol{\mu}}\left[\sigma^k \left(\lambda \mathbf{I}_K + \sum_{\kappa=1}^K \sigma^\kappa \hat{\boldsymbol{V}}_k\right)^{-1}\right]. \end{cases} \quad (11)$$

**Narrative of the proof** — The proof is detailed in Appendix A. It overcomes problems that existing methods, notably convex Gaussian comparison inequalities [20], have yet to be adapted to. The first main technical difficulty resides in the estimator of interest being a matrix learned with non-linear functions. This makes it impossible to decompose the problem on each row of the estimator, which must be characterised in a probabilistic sense directly as a matrix. The second main difficulty is brought by the mixture of arbitrary covariances. Intuitively, the covariances correlate the estimator with the individual clusters, and therefore the correlation function cannot be represented by a single quantity. In our proof, these points are handled using the AMP and related state-evolution techniques [33, 50–52]. The main idea of the proof is to express the estimator $\boldsymbol{W}^\star$ as the limit of a convergent sequence whose structure enables the decomposition of all correlations and distributions in closed form. AMP iterations can handle matrix valued variables [36, 53], correlations in block-structure [36], non-separable functions [37, 38] and compositions of the previous three, leaving a large choice of possibilities in their design. We thus reformulate the problem in a way that makes the interaction between the estimator and each cluster explicit, effectively introducing a block structure to the problem, and isolate the overlaps with the means $\{\boldsymbol{\mu}_k\}$. We then design a matrix-valued sequence that obeys the update rule of an AMP sequence, in order to benefit from its exact asymptotics, and whose fixed point condition matches the optimality condition of the ERM problem, Eq. (2). Our proof builds on the spatial coupling framework in the AMP literature [36, 54], which shows that the effect of random matrices defined with non-identically distributed blocks can be embedded in an AMP iteration while explicitly keeping the effect of each block. The non-linearities are then obtained by a block decomposition of the proximal operators defined on sets of matrices, acting on

different variables of the AMP sequence and representing the effect of each cluster. The convergence analysis is made possible by the convexity of the problem: the sequence is defined with proximal operators of convex functions which are roughly contractions, and results in converging sequences when combined with the high-dimensional properties of the iteration. It is also interesting to note that the replica method, although heuristic, yet again gives the correct prediction without any hindering from the aforementioned main difficulties, as detailed in Appendix B.

**Universality —** AMP-type proofs are amenable to both finite sample size analysis and universality proofs. For instance, in [55] it is shown that simpler instances of AMP for the LASSO exhibit exponential concentration in the system size, and the i.i.d. Gaussian assumption can be relaxed to independently sampled sub-Gaussian distributions, as shown in [56, 57]. Although these results do not formally encompass our case, their proof method contains most of the required technicalities, and it should be possible to prove similar results in the present setting. Indeed, recent results in [18] suggest that the formula of Theorem 1 and 2 should be universal for all mixtures of concentrated distribution in high-dimension, not only Gaussian ones. As we discuss Sec. 3.3, even real data learning curves are empirically found to follow the behavior of the mixture of Gaussians.

## 3 Results on synthetic and real datasets

In this section we exemplify how Theorem 1 can be employed to compute quantities of interest in different empirical risk minimisation tasks in high-dimensions. In all cases discussed below, eqs. (8) have been solved numerically. A repository with a polished version of the code we used to solve the equations is available on GitHub [58] (see also Appendix B.5).

### 3.1 Correlated sparse mixtures

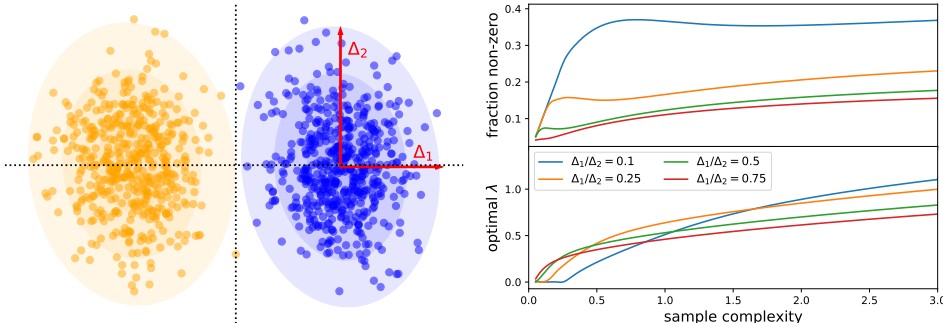

Figure 1: (**Left**) Two-dimensional projection of the Gaussian mixture introduced via Eq. (12) in which the sparse directions of the means are correlated with the weak/strong directions in the data. (**Right**) Fraction of non-zero elements of the lasso estimator (*top*) and optimal regularisation strength (*bottom*) as a function of the sample complexity $\alpha = n/d$ for different anisotropy ratios and fixed sparsity $\rho = 0.1$. Note that for $\Delta_1/\Delta_2 \lesssim 1$ and for low $\alpha$ the optimal error is achieved for vanishing regularisation, which corresponds to the *basis pursuit* algorithm [59].

As a first example, consider a binary classification problem in which the most relevant features live in a subspace of $\mathbb{R}^d$, and can be either weaker or stronger with respect to the irrelevant features. This problem can be modelled with a Gaussian mixture model with sparse means, and where the strong/weak directions of the covariance matrix are correlated with the non-zero components of the means. Mathematically, we consider a data set with $n$ independent samples $(\boldsymbol{x}^\nu, y^\nu) \in \mathbb{R}^d \times \{-1, 1\}$ drawn from a Gaussian mixture $\boldsymbol{x}^\nu \sim \mathcal{N}(y^\nu \boldsymbol{\mu}, \boldsymbol{\Sigma})$ with diagonal covariance $\Sigma_{ij} = \sigma_i \delta_{ij}$ which is correlated with the sparse means:

$$P(\boldsymbol{\mu}, \boldsymbol{\sigma}) = \prod_{i=1}^{d} \left\{ \rho \mathcal{N}(\mu_i | 0, 1) \delta_{\sigma_i, \Delta_1} + (1 - \rho) \delta_{\mu_i, 0} \delta_{\sigma_i, \Delta_2} \right\} \tag{12}$$

where $\rho > 0$ is the fraction of non-zero entries in $\boldsymbol{\mu}$. This model is closely related to the rare/weak features model introduced by Donoho and Jin in [19]. Indeed, in the case $\Delta_1 = \Delta_2 \equiv \Delta$ the

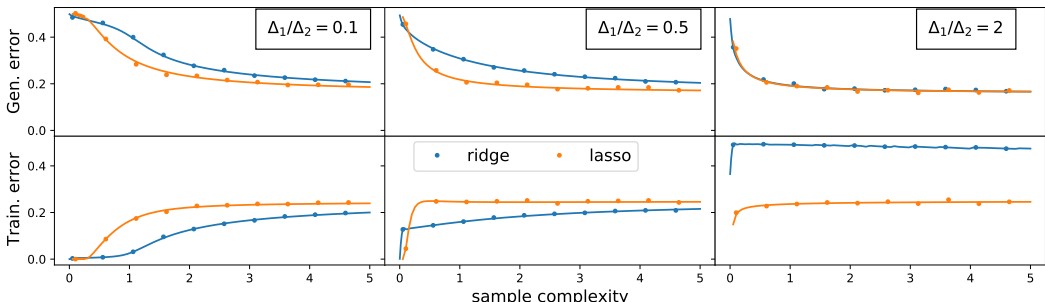

Figure 2: Learning curves for the sparse mixture model defined via Eq. (12) at fixed sparsity $\rho = 0.1$, comparing the performance of the ridge (blue) and the lasso (orange) estimators at optimal regularisation strength $\lambda^*$ and for different anisotropy ratio $\Delta_1/\Delta_2$ (here $\Delta_1 = 0.1$ and we vary $\Delta_2$). Full lines denote the theoretical prediction, and dots denote finite instance simulations with $d = 1000$ using the `ElasticNet` module in the `Scikit-learn` package [60]. Above a certain sample complexity $\alpha$, we can identify two regimes: a) a $\Delta_1/\Delta_2 \lesssim 1$ regime in which the $\ell_1$ penalty improves significantly over $\ell_2$; b) a $\Delta_1/\Delta_2 \gtrsim 1$ regime in which the performance is similar. Interestingly, even though the generalisation error of lasso is considerably better in a), the training loss (i.e. the mse on the labels) is higher, & vice-versa in b).

signal-to-noise ratio of the model is proportional to $\rho/\sqrt{\Delta}$, with $\rho$ and $\Delta^{-1/2}$ playing the roles of the parameters $\epsilon$ and $\mu_0$ setting the "rareness" and "strength" of the features in [19].

The formulas given in Theorem 1 simplify considerably for this model (see Appendix C for details), and therefore can be readily used to characterise the learning performance of different losses and penalties. For instance, one fundamental question we can address is when learning a sparse solution with the $\ell_1$ regularization is advantageous over the usual $\ell_2$. Figure 2 compares the learning curves computed from Theorem 1 for the lasso and ridge estimators, with optimal regularisation strength $\lambda^\star(\alpha) = \arg\min \epsilon_g(\alpha, \lambda)$ at fixed sparsity $\rho = 0.1$. We can see that lasso performs considerably better than ridge in the regime where $\Delta_1/\Delta_2 \lesssim 1$, while it achieves a similar performance when $\Delta_1/\Delta_2 \gtrsim 1$. This is quite intuitive: the sparse directions are uninformative, and therefore learning the relevant features is better when they are stronger. Figure 1 (right) shows how the sparsity of the learned estimator $W^\star$ and the optimal regularisation $\lambda^\star$ depends on the sample complexity $\alpha = n/d$. Interestingly, for $\Delta_1/\Delta_2 = 0.1$ or lower there is a region of small $\alpha$ in which basis pursuit ($\lambda = 0^+$) [59] is optimal, and the sparsity of the estimator has a curious non-monotonic behaviour with $\alpha$.

### 3.2 Separability transition for the cross-entropy loss

We now consider the problem of classifying points of $K$ Gaussian clusters using a cross-entropy loss

$$\ell(\boldsymbol{y}, \boldsymbol{x}) = -\sum_{k=1}^{K} y_k \ln \frac{e^{x_k}}{\sum_{\kappa=1}^{K} e^{x_\kappa}}. \tag{13}$$

Using the results of Theorem 2, we estimate the dependence of the generalisation error $\epsilon_g$ on the sample complexity $\alpha$ and on the regularisation $\lambda$. We assume Gaussian means $\boldsymbol{\mu}_k \sim \mathcal{N}(\boldsymbol{0}, \mathbf{I}_d/d)$ and diagonal covariances $\boldsymbol{\Sigma}_k \equiv \boldsymbol{\Sigma} = \Delta \mathbf{I}_d$. Finally, we adopt a ridge penalty, $r(\boldsymbol{W}) \equiv \|\boldsymbol{W}\|_{\mathrm{F}}^2/2$, and we focus on the case of balanced clusters, i.e., $\rho_k = 1/K$ for the sake of simplicity.

**Separability transition —** In Fig. 3 (left top) we plot the generalisation error $\epsilon_g$ as function of $\alpha$ for $2 \leq K \leq 5$ and $\lambda = 10^{-4}$. The smooth curve is obtained solving the fixed point equations in Theorem 1 and plugging the results in the formulas in Theorem 2. The results of numerical experiments are obtained averaging over $10^2$ instances of the problem solved using the `LogisticRegression` module in the `Scikit-learn` package [60]. An excellent agreement is observed. For each pair $(K, \Delta)$ and for vanishing regularisation $\lambda \to 0^+$ we observe a double-descent-like behaviour in the generalisation error. Indeed, the cusp $\alpha_K^\star(\Delta)$ in the generalisation error corresponds to the point in which the cross-entropy estimator ceases to perfectly interpolate the data, revealing the existence of a separability transition of the type discussed in [16] for Gaussian i.i.d. data. As stressed therein, a phase of perfect separability of the data points corresponds to a regime in which the maximum-likelihood

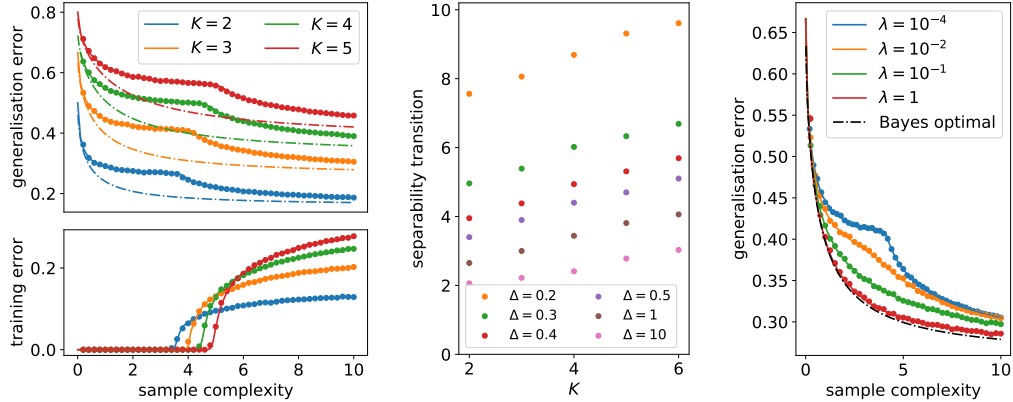

Figure 3: Classification of $K$ Gaussian clusters in $d$ dimensions, having Gaussian means and $\mathbf{\Sigma}_k \equiv \mathbf{\Sigma} = \Delta\mathbf{I}_d$ with $\Delta = 1/2$. In all presented cases, a quadratic regularisation has been adopted. Numerical experiments have been performed using $d = 10^3$. (**Left**) Generalisation error $\epsilon_g$ (*top*) and training error $\epsilon_t$ (*bottom*) as function of $\alpha$ at $\lambda = 10^{-4}$. Theoretical predictions (full lines) are compared with the results of numerical experiments (dots). Dash-dotted lines of the corresponding color represent, for comparison, the Bayes-optimal error. The results of numerical experiments are in agreement with the theoretical predictions in all cases. (**Center**) Separability transition $\alpha_K^\star$ as a function of $K$ in the same setting for different values of $\Delta$. (**Right**) Dependence of the generalisation error on the regularization $\lambda$ for $K = 3$ and $\Delta = 1/2$ in the balanced case, $\rho_k = 1/K$.

estimate does not exist with probability one. This is visible, in the same figure (left bottom), from the training error $\epsilon_t$ that is identically zero for $\alpha < \alpha_K^\star$, and strictly positive otherwise. Our result extends the observations in [14, 21], where an analytic expression for $\alpha_2^\star$ has been given in the case of for $K = 2$, $\boldsymbol{\mu}_1 = -\boldsymbol{\mu}_2$ Gaussian vector, generalising the classical result of Cover [43]. The separability transition point $\alpha_K^\star$ decreases with $\Delta$ and increases with $K$, showing that for larger $K$ it is easier to separate the different clusters: this intuitively follows from the fact that, at fixed $\alpha$ and $\Delta$, each cluster is given by $\alpha d/K$ points, i.e., fewer for increasing $K$ and therefore easier to classify, see Fig. 3 (center).

**The role of regularisation —** In Fig. 3 (right) we compare the performances of the cross-entropy loss with respect to the Bayes-optimal error (detailed in Appendix D) for different strength $\lambda$ of the regularisation, assuming all identical diagonal covariances $\mathbf{\Sigma}_k \equiv \mathbf{\Sigma} = \Delta\mathbf{I}_d$. In the case of balanced clusters (i.e., $\rho_k = 1/K$ for all $k$) it is observed that the generalisation error approaches the Bayes-optimal error for $\lambda \to +\infty$. The same phenomenology has been observed in [14, 24] in the $K = 2$ case with opposite means and generic loss, and in [20] for $K > 2$ for the square loss. Using the concentration results of Section 2, we investigated the robustness of this result in the case of balanced clusters but with different covariances and various losses. First, we considered two opposite *balanced* clusters with $\mathbf{\Sigma}_1 = \Delta_1\mathbf{I}_d$ and $\mathbf{\Sigma}_2 = \Delta_2\mathbf{I}_2$, $\Delta_1 \neq \Delta_2$, and we estimated the generalisation error at fixed sample complexity as function of $\lambda \in [10^{-4}, 10^2]$ using ridge regression. As shown in Fig. 4 (left), the regularisation strength optimising the error is finite, and in particular depends on the sample complexity. This situation is closer to what is observed in real problems with balanced data analysed using logistic regression. Indeed, using

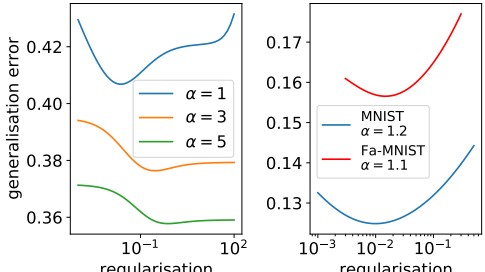

Figure 4: (**Left.**) Generalisation error obtained using ridge regression in the case of two balanced Gaussian clusters having $\mathbf{\Sigma}_1 = \frac{1}{10}\mathbf{I}_d$ and $\mathbf{\Sigma}_2 = \frac{1}{100}\mathbf{I}_d$ as function of $\lambda$ for different values of the sample complexity $\alpha$. (**Right**) Generalisation error $\epsilon_g$ as a function of $\lambda$ at fixed $\alpha$ in the binary classification of MNIST and in the FashionMNIST via logistic regression (see Sec. 3.3 for details).

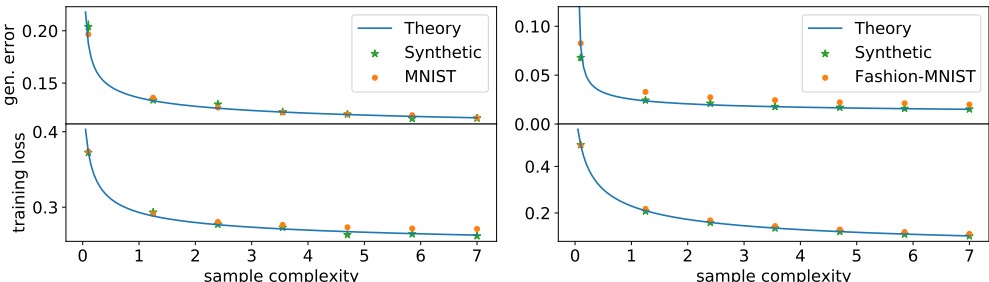

Figure 5: Generalisation error and training loss for the binary classification using the logistic loss on MNIST with $\lambda = 0.05$ (**left**) and on Fashion-MNIST with $\lambda = 1$ (**right**). The results are compared with synthetic data produced from the corresponding Gaussian mixture, and the theoretical prediction.

the covariances from real data sets such as MNIST or Fashion-MNIST yields a similar behaviour, see Fig. 4 (right), with an optimal $\lambda$ that is found to be finite.

### 3.3 Binary classification with real data

A recent line of works has reported that the asymptotic learning curves of simple regression tasks on real data sets can be well approximated by a surrogate Gaussian model matching the first two moments of the data [12, 45, 46]. However, this analysis was fundamentally restricted to least-squares regression, and considerable deviation from the Gaussian model was observed for classification tasks [12]. Authors of [18] have shown that realistic-looking data from trained generative adversarial networks behave like Gaussian mixtures. Here, we pursue these observations and investigate whether Theorem 2 can be used to capture the learning curves of classification tasks on two popular data sets: MNIST [61] and Fashion-MNIST [62]. Our goal is to compare the performances of some classification tasks on them with the predictions provided by the theory for the Gaussian mixture model.

Both data sets consist of $n_{\text{tot}} = 7 \times 10^4$ images $\hat{\boldsymbol{x}}^\mu \in \mathbb{R}^d$, $d = 784$. Each image $\hat{\boldsymbol{x}}^\mu$ is associated to a label $\hat{y}^\mu = \{0, 1, \ldots, 9\}$ specifying the type of represented digit (in the case of MNIST) or item (in the case of Fashion-MNIST). In both cases, we divided the database into two balanced classes (even vs odd digits for MNIST, clothes vs accessories for Fashion-MNIST), relabelling the elements $\hat{\boldsymbol{x}}^\mu$ with $y^\mu \in \{-1, 1\}$ depending on their class, and we selected $n < n_{\text{tot}}$ elements to perform the training, leaving the others for the test of the performances. We adopted a logistic loss with $\ell_2$ regularisation. First, we performed logistic regression on the training real data set, then we tested the learned estimators on the remaining $n_{\text{tot}} - n$ images. At the same time, for each class $k$ of the training set, we empirically estimated the corresponding mean $\boldsymbol{\mu}_k \in \mathbb{R}^d$ and covariance matrix $\boldsymbol{\Sigma}_k \in \mathbb{R}^{d \times d}$. We then assumed that the classification problem on the real database corresponds to a Gaussian mixture model of $K = 2$ clusters with means $\{\boldsymbol{\mu}_k\}_{k \in [2]}$ and covariances $\{\boldsymbol{\Sigma}_k\}_{k \in [2]}$. Under this assumption, we computed the generalisation error and the training loss predicted by the theory inserting the empirical means and covariances in our general formulas. The results are given in Fig. 5, showing a good agreement between the theoretical prediction and the results obtained on MNIST and Fashion-MNIST. In Fig. 5 we also plot, as reference, the results of a classification task performed on synthetic data, obtained generating a genuine Gaussian mixture with the means and covariances of the real data set.

Interestingly, this construction can also be used to analyse the learning curves of classification problems with non-linear feature maps [12], e.g. random features [63]. In this case, we first apply to our data set a feature map $\boldsymbol{x}^\mu = \text{erf}(\boldsymbol{F}\hat{\boldsymbol{x}}^\mu)$, where $\boldsymbol{F} \in \mathbb{R}^{p \times d}$ has i.i.d. Gaussian entries and the erf function is applied component wise. The classification task is then performed on the new data set $\{(\boldsymbol{x}^\nu, y^\nu)\}_{\nu \in [n]}$, the new data points $\boldsymbol{x}^\nu$ living in a $p$-dimensional space. We denote $\gamma = p/d$. We repeat the analysis described above in this new setting. Our results are in Fig. 6 for different values of $\gamma$. Once again, the generalisation error and the training loss are shown to be in a good agreement with both the theoretical prediction and the synthetic data sets obtained plugging in our formulas the real data means and the real data covariance matrices.

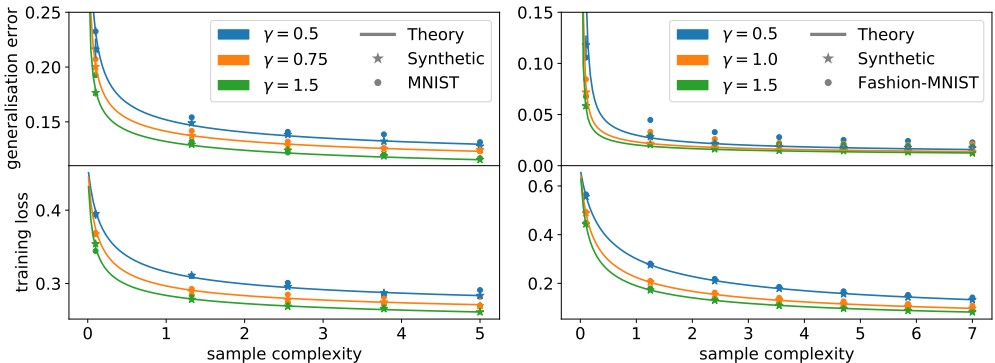

Figure 6: Generalisation error and training loss for the binary classification using the logistic on MNIST at $\lambda = 0.05$ (**left**) and on Fashion-MNIST at $\lambda = 1$ (**right**) in the random feature setting, for different values of $\gamma$, ratio between the number of parameters and the dimensionality of the data. The results are compared with synthetic data produced with the same $\gamma$, and the theoretical prediction.

## Acknowledgements

We thank Raphaël Berthier and Francesca Mignacco for discussions. We acknowledge funding from the ERC under the European Union's Horizon 2020 Research and Innovation Program Grant Agreement 714608- SMiLe, and from the French National Research Agency grants ANR-17-CE23-0023-01 PAIL. GS is grateful to EPFL for its generous hospitality during the finalization of the project.

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
