# Supplementary information for
*Learning Gaussian Mixtures with Generalised Linear Models:*
*Precise Asymptotics in High-dimensions*

**Bruno Loureiro, Gabriele Sicuro, Cédric Gerbelot, Alessandro Pacco, Florent Krzakala, Lenka Zdeborová**

## A Proof of the main results

This appendix presents the proof of the main technical result, Theorem 1. Throughout the whole proof, we assume that the set of conditions from Sec. 2 is verified.

### A.1 Required background

In this Section, we give an overview of the main concepts and tools on approximate message passing algorithms which will be required for the proof.

We start with some definitions that commonly appear in the approximate message-passing literature, see e.g. [33, 36, 37]. The main regularity class of functions we will use is that of pseudo-Lipschitz functions, which roughly amounts to functions with polynomially bounded first derivatives. We include the required scaling w.r.t. the dimensions in the definition for convenience.

**Definition 1** (Pseudo-Lipschitz function). *For $k, K \in \mathbb{N}^*$ and any $n, m \in \mathbb{N}^*$, a function $\boldsymbol{\phi} \colon \mathbb{R}^{n \times K} \to \mathbb{R}^{m \times K}$ is called a* pseudo-Lipschitz *of order $k$ if there exists a constant $L(k, K)$ such that for any $\boldsymbol{x}, \boldsymbol{y} \in \mathbb{R}^{n \times K}$,*

$$\frac{\|\boldsymbol{\phi}(\boldsymbol{x}) - \boldsymbol{\phi}(\boldsymbol{y})\|_{\mathrm{F}}}{\sqrt{m}} \leqslant L(k, K) \left( 1 + \left( \frac{\|\boldsymbol{x}\|_{\mathrm{F}}}{\sqrt{n}} \right)^{k-1} + \left( \frac{\|\boldsymbol{y}\|_{\mathrm{F}}}{\sqrt{n}} \right)^{k-1} \right) \frac{\|\boldsymbol{x} - \boldsymbol{y}\|_{\mathrm{F}}}{\sqrt{n}} \qquad (14)$$

*where $\|\bullet\|_{\mathrm{F}}$ denotes the Frobenius norm. Since $K$ will be kept finite, it can be absorbed in any of the constants.*

For example, the function $f : \mathbb{R}^n \to \mathbb{R}, \boldsymbol{x} \mapsto \frac{1}{n}\|\boldsymbol{x}\|_2^2$ is pseudo-Lipschitz of order 2.

**Moreau envelopes and Bregman proximal operators —** In our proof, we will also frequently use the notions of Moreau envelopes and proximal operators, see e.g. [47, 48]. These elements of convex analysis are often encountered in recent works on high-dimensional asymptotics of convex problems, and more detailed analysis of their properties can be found for example in [12, 31]. For the sake of brevity, we will only sketch the main properties of such mathematical objects, referring to the cited literature for further details. In this proof, we will mainly use proximal operators acting on sets of real matrices endowed with their canonical scalar product. Furthermore, proximals will be defined with matrix valued parameters in the following way: for a given convex function $f \colon \mathbb{R}^{d \times K} \to \mathbb{R}$, a given matrix $\boldsymbol{X} \in \mathbb{R}^{d \times K}$ and a given symmetric positive definite matrix $\boldsymbol{V} \in \mathbb{R}^{K \times K}$ with bounded spectral norm, we will consider operators of the type

$$\underset{\boldsymbol{T} \in \mathbb{R}^{d \times K}}{\operatorname{argmin}} \left\{ f(\boldsymbol{T}) + \frac{1}{2}\mathrm{tr}\left( (\boldsymbol{T} - \boldsymbol{X})\boldsymbol{V}^{-1}(\boldsymbol{T} - \boldsymbol{X})^{\top} \right) \right\} \qquad (15)$$

This operator can either be written as a standard proximal operator by factoring the matrix $\boldsymbol{V}^{-1}$ in the arguments of the trace:

$$\mathrm{Prox}_{f(\bullet \boldsymbol{V}^{1/2})}(\boldsymbol{X}\boldsymbol{V}^{-1/2})\boldsymbol{V}^{1/2} \in \mathbb{R}^{d \times K} \qquad (16)$$

or as a Bregman proximal operator [64] defined with the Bregman distance induced by the strictly convex, coercive function (for positive definite $\boldsymbol{V}$)

$$\boldsymbol{X} \mapsto \frac{1}{2}\mathrm{tr}(\boldsymbol{X}\boldsymbol{V}^{-1}\boldsymbol{X}^{\top}) \qquad (17)$$

which justifies the use of the Bregman resolvent

$$\underset{\boldsymbol{T} \in \mathbb{R}^{d \times K}}{\operatorname{argmin}} \left\{ f(\boldsymbol{T}) + \frac{1}{2}\mathrm{tr}\left( (\boldsymbol{T} - \boldsymbol{X})\boldsymbol{V}^{-1}(\boldsymbol{T} - \boldsymbol{X})^{\top} \right) \right\} = (\mathrm{Id} + \partial f(\bullet)\boldsymbol{V})^{-1}(\boldsymbol{X}) \qquad (18)$$

Many of the usual or similar properties to that of standard proximal operators (i.e. firm non-expansiveness, link with Moreau/Bregman envelopes,...) hold for Bregman proximal operators defined with the function (17), see e.g. [64, 65]. In particular, we will be using the equivalent notion to firmly nonexpansive operators for Bregman proximity operators, called *D-firm* operators. Consider the Bregman proximal defined with a differentiable, strictly convex, coercive function $g : \mathcal{X} \to \mathbb{R}$, where $\mathcal{X}$ is a given input Hilbert space. Let $T$ be the associated Bregman proximal of a given convex function $f : \mathcal{X} \to \mathbb{R}$, i.e., for any $\mathbf{x} \in \mathcal{X}$

$$T(\mathbf{x}) = \underset{\mathbf{y} \in \mathcal{X}}{\operatorname{argmin}} \{f(\mathbf{x}) + D_g(\mathbf{x}, \mathbf{y})\} \tag{19}$$

Then $T$ is *D-firm*, meaning it verifies

$$\langle T\boldsymbol{x} - T\boldsymbol{y}, \nabla g(T\boldsymbol{x}) - \nabla g(T\boldsymbol{y}) \rangle \leqslant \langle T\boldsymbol{x} - T\boldsymbol{y}, \nabla g(\boldsymbol{x}) - \nabla g(\boldsymbol{y}) \rangle \tag{20}$$

for any $\mathbf{x}, \mathbf{y}$ in $\mathcal{X}$.

**Gaussian concentration —** Gaussian concentration properties are at the root of this proof. Such properties are reviewed in more detail, for example, in [12, 37]. We refer the interested reader to this set of works for a detailed and complete discussion.

**Notations —** For any set of matrices $\{\boldsymbol{A}_k \in \mathbb{R}^{n_k \times d_k}\}_{k \in [K]}$ we will use the following notation:

$$\begin{bmatrix} \boldsymbol{A}_1 & & \\ & \boldsymbol{A}_2 & (*) \\ & (*) & \ddots \\ & & & \boldsymbol{A}_K \end{bmatrix} \equiv [\boldsymbol{A}_k]_{k=1}^K \in \mathbb{R}^{(\sum_{k=1}^K n_k) \times (\sum_{k=1}^K d_k)} \tag{21}$$

where the terms denoted by $(*)$ will be zero most of the time.
For a given function $\boldsymbol{\phi} \colon \mathbb{R}^{d \times K} \to \mathbb{R}^{d \times K}$, we write :

$$\boldsymbol{\phi}(\boldsymbol{X}) = \begin{bmatrix} \boldsymbol{\phi}^1(\boldsymbol{X}) \\ \vdots \\ \boldsymbol{\phi}^d(\boldsymbol{X}) \end{bmatrix} \in \mathbb{R}^{d \times K} \tag{22}$$

where each $\boldsymbol{\phi}^i \colon \mathbb{R}^{d \times K} \to \mathbb{R}^K$. We then write the $K \times K$ Jacobian

$$\frac{\partial \boldsymbol{\phi}^i}{\partial \boldsymbol{X}_j}(\boldsymbol{X}) = \begin{bmatrix} \frac{\partial \phi_1^i(\boldsymbol{X})}{\partial X_{j1}} & \cdots & \frac{\partial \phi_1^i(\boldsymbol{X})}{\partial X_{jK}} \\ \vdots & \ddots & \vdots \\ \frac{\partial \phi_K^i(\boldsymbol{X})}{\partial X_{j1}} & \cdots & \frac{\partial \phi_K^i(\boldsymbol{X})}{\partial X_{jK}} \end{bmatrix} \in \mathbb{R}^{K \times K} \tag{23}$$

For a given matrix $\boldsymbol{Q} \in \mathbb{R}^{K \times K}$, we write $\boldsymbol{Z} \in \mathbb{R}^{n \times K} \sim \mathcal{N}(\boldsymbol{0}, \boldsymbol{Q} \otimes \mathbf{I}_n)$ to denote that the lines of $\boldsymbol{Z}$ are sampled i.i.d. from $\mathcal{N}(\boldsymbol{0}, \boldsymbol{Q})$. Note that this is equivalent to saying that $\boldsymbol{Z} = \tilde{\boldsymbol{Z}} \boldsymbol{Q}^{1/2}$ where $\tilde{\boldsymbol{Z}} \in \mathbb{R}^{n \times K}$ is an i.i.d. standard normal random matrix. The notation $\overset{\mathrm{P}}{\simeq}$ denotes convergence in probability.

**Approximate message-passing —** Approximate message-passing algorithms are a statistical physics inspired family of iterations which can be used to solve high dimensional inference problems [66]. One of the central objects in such algorithms are the so called *state evolution equations*, a low-dimensional recursion equations which allow to exactly compute the high dimensional distribution of the iterates of the sequence. In this proof we will use a specific form of matrix-valued approximate message-passing iteration with non-separable non-linearities. In its full generality, the validity of the state evolution equations in this case is an extension of the works of [36, 37] included in [67]. Consider a sequence Gaussian matrices $\boldsymbol{A}(n) \in \mathbb{R}^{n \times d}$ with i.i.d. Gaussian entries, $A_{ij}(n) \sim \mathcal{N}(0, 1/d)$. For each $n, d \in \mathbb{N}$, consider two sequences of pseudo-Lipschitz functions

$$\{\boldsymbol{h}_t : \mathbb{R}^{n \times K} \to \mathbb{R}^{n \times K}\}_{t \in \mathbb{N}} \qquad \{\boldsymbol{e}_t : \mathbb{R}^{d \times K} \to \mathbb{R}^{d \times K}\}_{t \in \mathbb{N}} \tag{24}$$

initialized on $\boldsymbol{u}^0 \in \mathbb{R}^{d \times K}$ in such a way that the limit

$$\lim_{d \to \infty} \frac{1}{d} \big\| \boldsymbol{e}_0(\boldsymbol{u}^0)^\top \boldsymbol{e}_0(\boldsymbol{u}^0) \big\|_{\mathrm{F}} \tag{25}$$

exists and it is finite, and recursively define:

$$\boldsymbol{u}^{t+1} = \boldsymbol{A}^\top \boldsymbol{h}_t(\boldsymbol{v}^t) - \boldsymbol{e}_t(\boldsymbol{u}^t)\langle \boldsymbol{h}'_t \rangle^\top \tag{26}$$

$$\boldsymbol{v}^t = \boldsymbol{A}\boldsymbol{e}_t(\boldsymbol{u}^t) - \boldsymbol{h}_{t-1}(\boldsymbol{v}^{t-1})\langle \boldsymbol{e}'_t \rangle^\top \tag{27}$$

where the dimension of the iterates are $\boldsymbol{u}^t \in \mathbb{R}^{d\times K}$ and $\boldsymbol{v}^t \in \mathbb{R}^{n\times K}$. The terms in brackets are defined as:

$$\langle \boldsymbol{h}'_t \rangle = \frac{1}{d}\sum_{i=1}^n \frac{\partial \boldsymbol{h}_t^i}{\partial \boldsymbol{v}_i}(\boldsymbol{v}^t) \in \mathbb{R}^{K\times K} \quad \langle \boldsymbol{e}'_t \rangle = \frac{1}{d}\sum_{i=1}^d \frac{\partial \boldsymbol{e}_t^i}{\partial \boldsymbol{u}_i}(\boldsymbol{u}^t) \in \mathbb{R}^{K\times K} \tag{28}$$

We define now the *state evolution recursion* on two sequences of matrices $\{\boldsymbol{Q}_{r,s}\}_{s,r\geqslant 0}$ and $\{\hat{\boldsymbol{Q}}_{r,s}\}_{s,r\geqslant 1}$ initialized with $\boldsymbol{Q}_{0,0} = \lim_{d\to\infty}\frac{1}{d}\boldsymbol{e}_0(\boldsymbol{u}^0)^\top \boldsymbol{e}_0(\boldsymbol{u}^0)$:

$$\boldsymbol{Q}_{t+1,s} = \boldsymbol{Q}_{s,t+1} = \lim_{d\to\infty}\frac{1}{d}\mathbb{E}\left[\boldsymbol{e}_s(\hat{\boldsymbol{Z}}^s)^\top \boldsymbol{e}_{t+1}(\hat{\boldsymbol{Z}}^{t+1})\right] \in \mathbb{R}^{K\times K} \tag{29}$$

$$\hat{\boldsymbol{Q}}_{t+1,s+1} = \hat{\boldsymbol{Q}}_{s+1,t+1} = \lim_{d\to\infty}\frac{1}{d}\mathbb{E}\left[\boldsymbol{h}_s(\boldsymbol{Z}^s)^\top \boldsymbol{h}_t(\boldsymbol{Z}^t)\right] \in \mathbb{R}^{K\times K} \tag{30}$$

where $(\boldsymbol{Z}^0,\ldots,\boldsymbol{Z}^{t-1}) \sim \mathcal{N}(\boldsymbol{0}, \{\boldsymbol{Q}_{r,s}\}_{0\leqslant r,s\leqslant t-1}\otimes \mathbf{I}_n)$, $(\hat{\boldsymbol{Z}}^1,\ldots,\hat{\boldsymbol{Z}}^t) \sim \mathcal{N}(\boldsymbol{0}, \{\hat{\boldsymbol{Q}}_{r,s}\}_{1\leqslant r,s\leqslant t}\otimes \mathbf{I}_d)$. Then the following holds

**Theorem 4.** *In the setting of the previous paragraph, for any sequence of pseudo-Lipschitz functions* $\phi_n : (\mathbb{R}^{n\times K} \times \mathbb{R}^{d\times K})^t \to \mathbb{R}$, *for* $n, d \to \infty$:

$$\phi_n(\boldsymbol{u}^0, \boldsymbol{v}^0, \boldsymbol{u}^1, \boldsymbol{v}^1, \ldots, \boldsymbol{v}^{t-1}, \boldsymbol{u}^t) \overset{\mathrm{P}}{\simeq} \mathbb{E}\left[\phi_n\left(\boldsymbol{u}^0, \boldsymbol{Z}^0, \hat{\boldsymbol{Z}}^1, \boldsymbol{Z}^1, \ldots, \boldsymbol{Z}^{t-1}, \hat{\boldsymbol{Z}}^t\right)\right] \tag{31}$$

*where* $(\boldsymbol{Z}^0,\ldots,\boldsymbol{Z}^{t-1}) \sim \mathcal{N}(\boldsymbol{0}, \{\boldsymbol{Q}_{r,s}\}_{0\leqslant r,s\leqslant t-1}\otimes \mathbf{I}_n)$, $(\hat{\boldsymbol{Z}}^1,\ldots,\hat{\boldsymbol{Z}}^t) \sim \mathcal{N}(\boldsymbol{0}, \{\hat{\boldsymbol{Q}}_{r,s}\}_{1\leqslant r,s\leqslant t}\otimes \mathbf{I}_n)$.

**Spatial coupling** As a final premise to our proof, we give the intuition on how to handle a specific form of block random matrix in an AMP sequence. Consider the iteration (26), but this time with a Gaussian matrix defined as:

$$\boldsymbol{A} = \begin{bmatrix} \boldsymbol{A}_1 & & & \\ & \boldsymbol{A}_2 & (0) & \\ & (0) & \ddots & \\ & & & \boldsymbol{A}_K \end{bmatrix} \in \mathbb{R}^{n\times Kd} \tag{32}$$

where $\boldsymbol{A}_k \in \mathbb{R}^{n_k\times d}$ and $\sum_{k=1}^K n_k = n$, which leads to the following form for the products between matrices and non-linearities:

$$\boldsymbol{A}^\top \boldsymbol{h}_t(\boldsymbol{v}^t) = \begin{bmatrix} \boldsymbol{A}_1^\top \boldsymbol{h}_{1,t}(\boldsymbol{v}^t) \\ \boldsymbol{A}_2^\top \boldsymbol{h}_{2,t}(\boldsymbol{v}^t) \\ \vdots \\ \boldsymbol{A}_K^\top \boldsymbol{h}_{K,t}(\boldsymbol{v}^t) \end{bmatrix} \in \mathbb{R}^{Kd\times K} \quad \boldsymbol{A}\boldsymbol{e}_t(\boldsymbol{u}_t) = \begin{bmatrix} \boldsymbol{A}_1 \boldsymbol{e}_{1,t}(\boldsymbol{u}^t) \\ \boldsymbol{A}_2 \boldsymbol{e}_{2,t}(\boldsymbol{u}^t) \\ \vdots \\ \boldsymbol{A}_K \boldsymbol{e}_{K,t}(\boldsymbol{u}^t) \end{bmatrix} \in \mathbb{R}^{n\times K} \tag{33}$$

where the blocks $\boldsymbol{h}_{k,t}(\boldsymbol{v}^t) \in \mathbb{R}^{n_k\times K}, \boldsymbol{e}_{k,t}(\boldsymbol{u}_t) \in \mathbb{R}^{d\times K}$ may depend on their full arguments or only the corresponding blocks depending on their separability. This iteration can be embedded as a subset of the iterates of a larger sequence defined with the full version of the matrix $\boldsymbol{A}$ and non-linearities defined as:

$$\boldsymbol{e}_t : \mathbb{R}^{Kd\times K^2} \to \mathbb{R}^{Kd\times K^2}$$

$$\text{generates} \begin{bmatrix} \boldsymbol{e}_{1,t}(\bullet) & & & \\ & \boldsymbol{e}_{2,t}(\bullet) & (0) & \\ & (0) & \ddots & \\ & & & \boldsymbol{e}_{K,t}(\bullet) \end{bmatrix} \in \mathbb{R}^{Kd\times K^2} \tag{34}$$

$$\boldsymbol{h}_t : \mathbb{R}^{n\times K^2} \to \mathbb{R}^{n\times K^2}$$

$$\text{generates} \begin{bmatrix} \boldsymbol{h}_{1,t}(\bullet) & & & \\ & \boldsymbol{h}_{2,t}(\bullet) & (0) & \\ & (0) & \ddots & \\ & & & \boldsymbol{h}_{K,t}(\bullet) \end{bmatrix} \in \mathbb{R}^{n\times K^2} \tag{35}$$

The original iteration is recovered on the block diagonal of the variables of the iteration. This new setting, however, introduces a richer correlation structure, since each block will be described by a different $K \times K$ covariance according to the state evolution equations. Formally, the new covariance will be a $K^2 \times K^2$ block diagonal matrix. Also, the shape of the Onsager term changes from a matrix of size $K \times K$ to one of size $K^2 \times K^2$ with a $K \times (K \times K)$ block diagonal structure.

## A.2 Reformulation of the problem

We start by reformulating problem (2) in a way that can be treated efficiently using an AMP iteration. With respect to the main part of this paper, we will consider the estimator $\boldsymbol{W} \in \mathbb{R}^{d \times K}$ instead of $\mathbb{R}^{K \times d}$. The normalized (so that the cost does not diverge with the dimension) problem (2) then reads:

$$\min_{\boldsymbol{W} \in \mathbb{R}^{d \times K}, \boldsymbol{b} \in \mathbb{R}^K} \frac{1}{d} \left( L \left( \boldsymbol{Y}, \frac{1}{\sqrt{d}} \boldsymbol{X} \boldsymbol{W} + \boldsymbol{b} \right) + r(\boldsymbol{W}) \right) \tag{36}$$

where we have introduced the function $L : \mathbb{R}^{n \times K} \times \mathbb{R}^{n \times K} \to \mathbb{R}$ acting as

$$\left( \boldsymbol{Y}, \frac{1}{\sqrt{d}} \boldsymbol{X} \boldsymbol{W} + \boldsymbol{b} \right) \mapsto \sum_{\nu=1}^{n} \ell \left( \boldsymbol{y}^\nu, \frac{\boldsymbol{W} \boldsymbol{x}^\nu}{\sqrt{d}} + \boldsymbol{b} \right), \tag{37}$$

the matrix $\boldsymbol{Y} \in \mathbb{R}^{n \times K}$ of concatenated one-hot encoded labels, and the matrix of concatenated means $\boldsymbol{M} \in \mathbb{R}^{K \times d}$ (in the main we took the transpose $\boldsymbol{M} \in \mathbb{R}^{d \times K}$). Until further notice, we will drop the scaling $\frac{1}{d}$ for convenience and study the problem

$$\min_{\boldsymbol{W} \in \mathbb{R}^{d \times K}, \boldsymbol{b} \in \mathbb{R}^K} L \left( \boldsymbol{Y}, \frac{1}{\sqrt{d}} \boldsymbol{X} \boldsymbol{W} + \boldsymbol{b} \right) + r(\boldsymbol{W}) \tag{38}$$

We will write $L_k$ the application of $\ell$ on each row of a sub-block in $\mathbb{R}^{n_k \times K}$. Without loss of generality, we can assume that the samples are grouped by clusters in the data matrix, giving the following form for $\boldsymbol{X} \in \mathbb{R}^{n \times d}$, separating the mean part $\boldsymbol{Y} \boldsymbol{M}$ and centered Gaussian part :

$$\boldsymbol{X} = \boldsymbol{Y} \boldsymbol{M} + \tilde{\boldsymbol{Z}} \boldsymbol{\Sigma} \in \mathbb{R}^{n \times d} \tag{39}$$

where we have introduced the block-diagonal matrix $\tilde{\boldsymbol{Z}}$ and the $Kd \times d$ full-column-rank matrix $\boldsymbol{\Sigma}$

$$\tilde{\boldsymbol{Z}} = \begin{bmatrix} \boldsymbol{Z}_1 & & & \\ & \boldsymbol{Z}_2 & (0) & \\ & (0) & \ddots & \\ & & & \boldsymbol{Z}_K \end{bmatrix} \in \mathbb{R}^{n \times Kd} \qquad \boldsymbol{\Sigma} = \begin{bmatrix} \boldsymbol{\Sigma}_1^{1/2} \\ \boldsymbol{\Sigma}_2^{1/2} \\ \vdots \\ \boldsymbol{\Sigma}_K^{1/2} \end{bmatrix} \in \mathbb{R}^{Kd \times d}. \tag{40}$$

Here $(\boldsymbol{Z}_1, \ldots, \boldsymbol{Z}_K) \in \mathbb{R}^{n_1 \times d} \times \cdots \times \mathbb{R}^{n_K \times d}$ are independent, i.i.d. standard normal matrices.

The product between the data matrix and the weights $\boldsymbol{W} \in \mathbb{R}^{d \times K}$ then reads:

$$\boldsymbol{X} \boldsymbol{W} = \boldsymbol{Y} \boldsymbol{M} \boldsymbol{W} + \tilde{\boldsymbol{Z}} \boldsymbol{\Sigma} \boldsymbol{W} = \begin{bmatrix} \boldsymbol{Y}_1 \boldsymbol{M} \boldsymbol{W} + \boldsymbol{Z}_1 \boldsymbol{\Sigma}_1^{1/2} \boldsymbol{W} \\ \vdots \\ \boldsymbol{Y}_K \boldsymbol{M} \boldsymbol{W} + \boldsymbol{Z}_K \boldsymbol{\Sigma}_K^{1/2} \boldsymbol{W} \end{bmatrix} \in \mathbb{R}^{n \times K} \tag{41}$$

where each $\boldsymbol{Y}_k \in \mathbb{R}^{n_k \times d}$ is a $n_k$ copy of the same label vector. Defining now $\tilde{\boldsymbol{W}} = \boldsymbol{\Sigma} \boldsymbol{W}$, observe that

$$\tilde{\boldsymbol{W}} = \boldsymbol{\Sigma} \boldsymbol{W} \quad \Longrightarrow \quad \boldsymbol{W} = \boldsymbol{\Sigma}^+ \tilde{\boldsymbol{W}}, \tag{42}$$

where

$$\boldsymbol{\Sigma}^+ \equiv \left( \sum_{k=1}^{K} \boldsymbol{\Sigma}_k \right)^{-1} \boldsymbol{\Sigma}^\top \tag{43}$$

is the pseudo-inverse of the matrix $\boldsymbol{\Sigma}$. The optimization problem (2) is thus equivalent to

$$\inf_{\substack{\tilde{\boldsymbol{W}} \in \mathbb{R}^{Kd \times K} \\ \boldsymbol{b} \in \mathbb{R}^K}} \sum_{k=1}^{K} L_k \left( \frac{1}{\sqrt{d}} \boldsymbol{Y}_k \boldsymbol{M} \boldsymbol{W} + \frac{1}{\sqrt{d}} \boldsymbol{Z}_k \tilde{\boldsymbol{W}}_k, \boldsymbol{b} \right) + r \left( \boldsymbol{\Sigma}^+ \tilde{\boldsymbol{W}} \right) \tag{44}$$

Introducing the order parameter $\boldsymbol{m} = \frac{1}{\sqrt{d}}\boldsymbol{M}\boldsymbol{W} \in \mathbb{R}^{K \times K}$, we reformulate Eq.(44) as a constrained optimization problem :

$$\inf_{\boldsymbol{m},\tilde{\boldsymbol{W}},\boldsymbol{b}} \sum_{k=1}^{K} L_k \left( \frac{1}{\sqrt{d}}\boldsymbol{Y}_k\boldsymbol{m} + \frac{1}{\sqrt{d}}\boldsymbol{Z}_k\tilde{\boldsymbol{W}}_k \right) + r\left(\boldsymbol{\Sigma}^+\tilde{\boldsymbol{W}}\right) \tag{45}$$

$$\text{s.t.} \quad \frac{1}{\sqrt{d}}\boldsymbol{M}\boldsymbol{\Sigma}^+\tilde{\boldsymbol{W}} = \boldsymbol{m}$$

whose Lagrangian form, with dual parameters $\hat{\boldsymbol{m}} \in \mathbb{R}^{K \times K}$, reads

$$\inf_{\boldsymbol{m},\tilde{\boldsymbol{W}},\boldsymbol{b}} \sup_{\hat{\boldsymbol{m}}} \sum_{k=1}^{K} L_k \left( \boldsymbol{Y}_k\boldsymbol{m} + \frac{1}{\sqrt{d}}\boldsymbol{Z}_k\tilde{\boldsymbol{W}}_k \right) + r\left(\boldsymbol{\Sigma}^+\tilde{\boldsymbol{W}}\right) + \text{tr}\left( \hat{\boldsymbol{m}}^\top \left( \boldsymbol{m} - \frac{1}{\sqrt{d}}\boldsymbol{M}\boldsymbol{\Sigma}^+\tilde{\boldsymbol{W}} \right) \right). \tag{46}$$

This is a proper, closed, convex, strictly feasible optimization problem, thus strong duality holds and we can invert the order of the inf-sup to focus on the minimization problem in $\tilde{\boldsymbol{W}}$ for fixed $\boldsymbol{m},\hat{\boldsymbol{m}},\boldsymbol{b}$:

$$\inf_{\tilde{\boldsymbol{W}} \in \mathbb{R}^{Kd \times K}} \tilde{L}\left( \frac{1}{\sqrt{d}}\tilde{\boldsymbol{Z}}\tilde{\boldsymbol{W}} \right) + \tilde{r}(\tilde{\boldsymbol{W}}) \tag{47}$$

where we defined the loss term

$$\tilde{L} : \mathbb{R}^{n \times K} \to \mathbb{R}$$
$$\frac{1}{\sqrt{d}}\tilde{\boldsymbol{Z}}\tilde{\boldsymbol{W}} \mapsto \sum_{k=1}^{K} L_k \left( \boldsymbol{Y}_k\boldsymbol{m} + \frac{1}{\sqrt{d}}\boldsymbol{Z}_k\tilde{\boldsymbol{W}}_k \right) = \sum_{k=1}^{K} \sum_{i=1}^{n_k} \ell\left( \left[ \boldsymbol{Y}_k\boldsymbol{m} + \frac{1}{\sqrt{d}}\boldsymbol{Z}_k\tilde{\boldsymbol{W}}_k \right]_i \right) \tag{48a}$$

and the regularisation term

$$\tilde{r} : \mathbb{R}^{Kd \times K} \to \mathbb{R}$$
$$\tilde{\boldsymbol{W}} \mapsto r\left(\boldsymbol{\Sigma}^+\tilde{\boldsymbol{W}}\right) + \text{tr}\left( \hat{\boldsymbol{m}}^\top \left( \boldsymbol{m} - \frac{1}{\sqrt{d}}\boldsymbol{M}\boldsymbol{\Sigma}^+\tilde{\boldsymbol{W}} \right) \right) \tag{48b}$$

where $\boldsymbol{\Sigma}^\top\tilde{\boldsymbol{W}} = \sum_{k=1}^{K} \boldsymbol{\Sigma}_k^{1/2}\boldsymbol{W}_k$ and $\tilde{\boldsymbol{Z}} = [\boldsymbol{Z}_k]_{k=1}^{K} \in \mathbb{R}^{n \times Kd}$ is an i.i.d. standard normal block diagonal matrix as in Eq. (40).

### A.3   Finding the AMP sequence

We now need to find an AMP iteration relating to $\tilde{\boldsymbol{W}}$ that solve the optimization problem in Eq. (47). Although this section is not written as a formal proof, all steps are rigorous. The aim is to give the reader the core intuition on how the AMP iteration is found, otherwise the solution may feel "parachuted". The reader uninterested in the underlying intuition may directly skip to the next section. In order to find the appropriate sequence two key points must be considered :

- the fixed point of the sequence has to match the optimality condition of Eq. (47);
- the update rule of the sequence should have the form Eq. (26) for the state evolution equations to hold.

These two points completely determine the form of the iteration. In the subsequent derivation, we absorb the scaling $\frac{1}{\sqrt{d}}$ in the matrix $\tilde{\boldsymbol{Z}}$, such that the $\boldsymbol{Z}_k \in \mathbb{R}^{n_k \times d}$ have i.i.d. $\mathcal{N}(0, 1/d)$ elements.

**Resolvent of the loss term —**   Going back to problem Eq. (47), its optimality condition will look like :

$$\tilde{\boldsymbol{Z}}^\top \partial\tilde{L}(\boldsymbol{Z}\tilde{\boldsymbol{W}}) + \partial\tilde{r}(\tilde{\boldsymbol{W}}) = 0 \iff \begin{bmatrix} \boldsymbol{Z}_1^\top & & \\ & \boldsymbol{Z}_2^\top & (0) \\ & (0) & \ddots \\ & & & \boldsymbol{Z}_K^\top \end{bmatrix} \begin{bmatrix} \partial\tilde{L}_1(\boldsymbol{Z}_1\tilde{\boldsymbol{W}}_1) \\ \partial\tilde{L}_2(\boldsymbol{Z}_2\tilde{\boldsymbol{W}}_2)) \\ \vdots \\ \partial\tilde{L}_K(\boldsymbol{Z}_K\tilde{\boldsymbol{W}}_K)) \end{bmatrix} + \partial\tilde{r}(\tilde{\boldsymbol{W}}) = 0 \tag{49}$$

where each $\boldsymbol{Z}_k \in \mathbb{R}^{n_k \times d}$, and the subdifferential of $\tilde{L}$ is separable across blocks of size $n_k \times d$, and $\partial \tilde{r}(\tilde{\boldsymbol{W}}) \in \mathbb{R}^{Kd \times K}$. Following the intuition of spatial coupling, we introduce the *full* matrix $\boldsymbol{Z} \in \mathbb{R}^{n \times Kd}$, with i.i.d. $\mathcal{N}(0, 1/d)$ entries. The optimality condition can then be written on the diagonal of a $Kd \times K^2$ matrix:

$$
\boldsymbol{Z}^\top \begin{bmatrix} \partial \tilde{L}_1(\boldsymbol{Z}_1 \tilde{\boldsymbol{W}}_1) & & & \\ & \partial \tilde{L}_2(\boldsymbol{Z}_2 \tilde{\boldsymbol{W}}_2) & (0) & \\ & (0) & \ddots & \\ & & & \partial \tilde{L}_K(\boldsymbol{Z}_K \tilde{\boldsymbol{W}}_K) \end{bmatrix}
$$
$$
+ \begin{bmatrix} \partial \tilde{r}(\tilde{\boldsymbol{W}})_1 & & & \\ & \partial \tilde{r}(\tilde{\boldsymbol{W}})_2 & (0) & \\ & (0) & \ddots & \\ & & & \partial \tilde{r}(\tilde{\boldsymbol{W}})_K \end{bmatrix} = \boldsymbol{0} \quad (50)
$$

where $\partial \tilde{r}(\tilde{\boldsymbol{W}})_k$ represents the $k$-th block of the subdifferential of $\tilde{r}$ which is non-separable across the blocks of $\tilde{\boldsymbol{W}}$. To make the resolvents/proximals appear, we add the argument of the subdifferentials on both sides weighted by a (symmetric) positive definite matrix $\boldsymbol{S}_k \in \mathbb{R}^{K \times K}$ which will be used to allow for Onsager correction while respecting the fixed point condition. Using the notation defined in section A.1

$$
\left[\boldsymbol{Z}_k^\top \partial \tilde{L}_k(\boldsymbol{Z}_k \tilde{\boldsymbol{W}}_k)\right]_{k=1}^K + \left[\partial \tilde{r}(\tilde{\boldsymbol{W}})\right]_{k=1}^K = 0
$$
$$
\iff \left[\boldsymbol{Z}_k^\top \partial \tilde{L}_k(\boldsymbol{Z}_k \tilde{\boldsymbol{W}}_k) + \boldsymbol{Z}_k^\top \boldsymbol{Z}_k \tilde{\boldsymbol{W}}_k \boldsymbol{S}_k^{-1}\right]_{k=1}^K + \left[\partial \tilde{r}(\tilde{\boldsymbol{W}})\right]_{k=1}^K = \left[\boldsymbol{Z}_k^\top \boldsymbol{Z}_k \tilde{\boldsymbol{W}}_k \boldsymbol{S}_k^{-1}\right]_{k=1}^K \quad (51)
$$

for a given set of positive definite matrices $\{\boldsymbol{S}_k\}_{k \in [K]}$. Again, the reason for introducing different $\boldsymbol{S}_k$ on each block is to match the expected structure of the Onsager term. We can introduce the resolvent, formally Bregman resolvent/proximal operator:

$$
\boldsymbol{U}_k \equiv \partial \tilde{L}_k(\boldsymbol{Z}_k \tilde{\boldsymbol{W}}_k) \boldsymbol{S}_k + \boldsymbol{Z}_k \tilde{\boldsymbol{W}}_k \iff \boldsymbol{Z}_k \tilde{\boldsymbol{W}}_k = \boldsymbol{R}_{\tilde{L}_k, \boldsymbol{S}_k}(\boldsymbol{U}_k) \quad (52)
$$

where

$$
\boldsymbol{R}_{\tilde{L}_k, \boldsymbol{S}_k}(\boldsymbol{U}_k) = (\mathrm{Id} + \partial \tilde{L}_k(\bullet)\boldsymbol{S}_k)^{-1}(\boldsymbol{U}_k)
$$
$$
= \underset{\boldsymbol{T} \in \mathbb{R}^{n_k \times K}}{\operatorname{argmin}} \left\{ \tilde{L}_k(\boldsymbol{T}) + \frac{1}{2}\mathrm{tr}\big((\boldsymbol{T} - \boldsymbol{U}_k)\boldsymbol{S}_k^{-1}(\boldsymbol{T} - \boldsymbol{U}_k)^\top\big) \right\}
$$
$$
= \underset{\boldsymbol{T} \in \mathbb{R}^{n_k \times K}}{\operatorname{argmin}} \left\{ L_k(\boldsymbol{T}) + \frac{1}{2}\mathrm{tr}\big((\boldsymbol{T} - (\boldsymbol{Y}_k \boldsymbol{m} + \boldsymbol{U}_k))\boldsymbol{S}_k^{-1}(\boldsymbol{T} - (\boldsymbol{Y}_k \boldsymbol{m} + \boldsymbol{U}_k))^\top\big) \right\} - \boldsymbol{Y}_k \boldsymbol{m}.
$$
$$
\quad (53)
$$

In the previous expressions $\partial \tilde{L}_k \in \mathbb{R}^{n_k \times K}$ and $\boldsymbol{V}_k \in \mathbb{R}^{K \times K}$. The following formulation of the optimality condition is reached:

$$
\left[\boldsymbol{Z}_k^\top \boldsymbol{U}_k \boldsymbol{S}_k^{-1}\right]_{k=1}^K + \left[\partial \tilde{r}(\tilde{\boldsymbol{W}})_k\right]_{k=1}^K = \left[\boldsymbol{Z}_k^\top \boldsymbol{R}_{\tilde{L}_k, \boldsymbol{S}_k}(\boldsymbol{U}_k) \boldsymbol{S}_k^{-1}\right]_{k=1}^K
$$
$$
\iff \left[\boldsymbol{Z}_k^\top \left(\boldsymbol{U}_k - \boldsymbol{R}_{\tilde{L}_k, \boldsymbol{S}_k}(\boldsymbol{U}_k)\right) \boldsymbol{S}_k^{-1}\right]_{k=1}^K + \left[\partial \tilde{r}(\tilde{\boldsymbol{W}})_k\right]_{k=1}^K = 0 \quad (54)
$$

**Resolvent of the regularization term** Determining the block decomposition of the subdifferential of the regularization term is less simple. We would like a block expression in the flavour of:

$$
\left[\partial \tilde{r}(\tilde{\boldsymbol{W}})_k\right]_{k=1}^K + \left[\tilde{\boldsymbol{W}}_k \hat{\boldsymbol{S}}_k^{-1}\right]_{k=1}^K = \left[\tilde{\boldsymbol{W}}_k \hat{\boldsymbol{S}}_k^{-1}\right]_{k=1}^K \quad (55)
$$

At this point it becomes clear that we cannot consider the resolvent as acting on $\tilde{\boldsymbol{W}} \in \mathbb{R}^{Kd \times K}$ otherwise there could be only one $\hat{\boldsymbol{S}} \in \mathbb{R}^{K \times K}$ and there would be a mismatch with the expected form of the Onsager terms. As specified by the definitions Eq.(48), the subdifferential of $\tilde{r}$ is acting on the whole block diagonal matrix $[\tilde{\boldsymbol{W}}_k]_{k=1}^K$, by way of summation due to the action of the pseudo-inverse

$\mathbf{\Sigma}^+$. We can thus consider its proximal acting on $\mathbb{R}^{d \times K^2}$ as $[\tilde{\boldsymbol{W}}_1 \tilde{\boldsymbol{W}}_2 ... \tilde{\boldsymbol{W}}_K]$ (note that we could have also worked directly with a block diagonal matrix in $\mathbb{R}^{Kd \times K^2}$). Proceeding in this way, we can directly write our expression as an application parametrized by another set of positive definite matrices $\{\hat{\boldsymbol{S}}_k\}_{k \in [K]}$.

$$\hat{\boldsymbol{U}} = \left( \mathrm{Id} + \partial \tilde{r}(\bullet)\hat{\boldsymbol{S}} \right) (\tilde{\boldsymbol{W}}) \qquad \tilde{\boldsymbol{W}} = \boldsymbol{R}_{\tilde{r},\hat{\boldsymbol{S}}}(\hat{\boldsymbol{U}}) \tag{56}$$

where

$$\begin{aligned}
\boldsymbol{R}_{\tilde{r},\hat{\boldsymbol{S}}}(\hat{\boldsymbol{U}}) &= \left( \mathrm{Id} + \partial \tilde{r}(\bullet)\hat{\boldsymbol{S}} \right)^{-1} (\hat{\boldsymbol{U}}) \\
&= \underset{\boldsymbol{T} \in \mathbb{R}^{d \times K^2}}{\operatorname{argmin}} \left\{ \tilde{r}(\boldsymbol{T}) + \frac{1}{2}\mathrm{tr}\left( (\boldsymbol{T} - \hat{\boldsymbol{U}})\hat{\boldsymbol{S}}^{-1}(\boldsymbol{T} - \hat{\boldsymbol{U}})^\top \right) \right\}
\end{aligned} \tag{57}$$

where $\hat{\boldsymbol{S}} \in \mathbb{R}^{K^2 \times K^2}$ block diagonal, and $\hat{\boldsymbol{U}} \in \mathbb{R}^{d \times K^2}$. This would lead to the equivalent optimality condition for the regularization part:

$$\hat{\boldsymbol{U}}\hat{\boldsymbol{S}}^{-1} = \boldsymbol{R}_{\tilde{r},\hat{\boldsymbol{S}}}(\hat{\boldsymbol{U}})\hat{\boldsymbol{S}}^{-1} \iff \left[ \hat{\boldsymbol{U}}_k \hat{\boldsymbol{S}}_k^{-1} \right]_{k=1}^K = \left[ \boldsymbol{R}_{\tilde{r},\hat{\boldsymbol{S}},k}(\hat{\boldsymbol{U}})\hat{\boldsymbol{S}}_k^{-1} \right]_{k=1}^K \tag{58}$$

We now need to figure out the block structure of this resolvent since we want to spread it across a block diagonal matrix. Let $\boldsymbol{C} = \sum_{k=1}^K \boldsymbol{\Sigma}_k$, so that $\boldsymbol{\Sigma}^+ = \boldsymbol{C}^{-1}\boldsymbol{\Sigma}^\top$, and the blocks $\boldsymbol{T}_k \in \mathbb{R}^{d \times K}$ are the solution to the minimization problem

$$\begin{aligned}
\min_{\{\boldsymbol{T}_k\}_{k \in [K]} \in (\mathbb{R}^{d \times K})^K} \; & r(\boldsymbol{C}^{-1}\sum_{k=1}^K \boldsymbol{\Sigma}_k^{1/2}\boldsymbol{T}_k) + \frac{1}{2}\mathrm{tr}\left( (\boldsymbol{T} - \hat{\boldsymbol{U}})\hat{\boldsymbol{S}}^{-1}(\boldsymbol{T} - \hat{\boldsymbol{U}}^\top) \right) \\
& + \mathrm{tr}\left( \hat{\boldsymbol{m}}^\top \left( \boldsymbol{m} - \frac{1}{\sqrt{d}}\boldsymbol{M}\boldsymbol{\Sigma}^+\boldsymbol{T} \right) \right)
\end{aligned} \tag{59}$$

Let $\tilde{\boldsymbol{T}} = \boldsymbol{C}^{-1}\sum_{k=1}^K \boldsymbol{\Sigma}_k^{1/2}\boldsymbol{T}_k \in \mathbb{R}^{d \times K}$, and the equivalent reformulation as a constraint optimization problem:

$$\min_{\substack{\boldsymbol{T}_{k \in [K]} \in \mathbb{R}^{d \times K} \\ \tilde{\boldsymbol{T}} \in \mathbb{R}^{d \times K}}} r(\tilde{\boldsymbol{T}}) + \frac{1}{2}\mathrm{tr}\left( (\boldsymbol{T} - \hat{\boldsymbol{U}})\hat{\boldsymbol{S}}^{-1}(\boldsymbol{T} - \hat{\boldsymbol{U}}^\top) \right) + \mathrm{tr}\left( \hat{\boldsymbol{m}}^\top \left( \boldsymbol{m} - \frac{1}{\sqrt{d}}\boldsymbol{M}\tilde{\boldsymbol{T}} \right) \right) \tag{60}$$

$$\text{s.t.} \quad \tilde{\boldsymbol{T}} = \boldsymbol{C}^{-1}\sum_{k=1}^K \boldsymbol{\Sigma}_k^{1/2}\boldsymbol{T}_k$$

This is a feasible convex problem under convex constraint with a strongly convex term, it thus has a unique solution and strong duality holds. Introducing the Lagrange multiplier $\boldsymbol{\lambda} \in \mathbb{R}^{d \times K}$, we get the equivalent representation:

$$\begin{aligned}
\min_{\substack{\boldsymbol{T}_{k \in [K]} \in \mathbb{R}^{d \times K} \\ \tilde{\boldsymbol{T}} \in \mathbb{R}^{d \times K}}} \max_{\boldsymbol{\lambda} \in \mathbb{R}^{d \times K}} \; & r(\tilde{\boldsymbol{T}}) + \sum_{k=1}^K \mathrm{tr}\left( (\boldsymbol{T}_k - \hat{\boldsymbol{U}}_k)\hat{\boldsymbol{S}}_k^{-1}(\boldsymbol{T}_k - \hat{\boldsymbol{U}}_k)^\top \right) \\
& + \mathrm{tr}\left( \boldsymbol{\lambda}^\top \left( \tilde{\boldsymbol{T}} - \boldsymbol{C}^{-1}\sum_{k=1}^K \boldsymbol{\Sigma}_k^{1/2}\boldsymbol{T}_k \right) \right) + \mathrm{tr}\left( \hat{\boldsymbol{m}}^\top \left( \boldsymbol{m} - \frac{1}{\sqrt{d}}\boldsymbol{M}\tilde{\boldsymbol{T}} \right) \right).
\end{aligned} \tag{61}$$

The optimality condition for this problem reads:

$$\partial_{\tilde{\boldsymbol{T}}} : \quad \partial r(\tilde{\boldsymbol{T}}) + \boldsymbol{\lambda} - \frac{1}{\sqrt{d}}\boldsymbol{M}^\top \hat{\boldsymbol{m}} = 0 \tag{62}$$

$$\partial_{\boldsymbol{T}} : \quad (\boldsymbol{T}_k - \boldsymbol{U}_k)\hat{\boldsymbol{S}}_k^{-1} = \boldsymbol{\Sigma}_k^{1/2}\boldsymbol{C}^{-1}\boldsymbol{\lambda} \qquad \forall k \in [K] \tag{63}$$

$$\partial_{\boldsymbol{\lambda}} : \quad \tilde{\boldsymbol{T}} = \boldsymbol{C}^{-1}\sum_{k=1}^K \boldsymbol{\Sigma}_k^{1/2}\boldsymbol{T}_k \tag{64}$$

Using the gradient condition on $\boldsymbol{T}$, we get

$$\sum_{k=1}^{K} \boldsymbol{\Sigma}_k^{1/2}(\boldsymbol{T}_k - \hat{\boldsymbol{U}}_k)\hat{\boldsymbol{S}}_k^{-1} = \boldsymbol{\lambda} \tag{65}$$

The constraint $\tilde{\boldsymbol{T}} = \boldsymbol{C}^{-1}\sum_{k=1}^{K}\boldsymbol{\Sigma}_k^{1/2}\boldsymbol{T}_k$ is solved by $\boldsymbol{T}_k = \boldsymbol{\Sigma}_k^{1/2}\tilde{\boldsymbol{T}}$ which gives the solution for $\boldsymbol{\lambda}$

$$\boldsymbol{\lambda} = \sum_{k=1}^{K} \boldsymbol{\Sigma}_k^{1/2}(\boldsymbol{\Sigma}_k^{1/2}\tilde{\boldsymbol{T}} - \hat{\boldsymbol{U}}_k)\hat{\boldsymbol{S}}_k^{-1} = \sum_{k=1}^{K}\boldsymbol{\Sigma}_k\tilde{\boldsymbol{T}}\hat{\boldsymbol{S}}_k^{-1} - \sum_{k=1}^{K}\boldsymbol{\Sigma}_k^{1/2}\hat{\boldsymbol{U}}_k\hat{\boldsymbol{S}}_k^{-1} \tag{66}$$

and prescribes the following form for $\tilde{\boldsymbol{T}}$, as solution to the problem

$$\partial r(\tilde{\boldsymbol{T}}) + \sum_{k=1}^{K}\boldsymbol{\Sigma}_k\tilde{\boldsymbol{T}}\hat{\boldsymbol{S}}_k^{-1} - \sum_{k=1}^{K}\boldsymbol{\Sigma}_k^{1/2}\hat{\boldsymbol{U}}_k\hat{\boldsymbol{S}}_k^{-1} - \frac{1}{\sqrt{d}}\boldsymbol{M}^\top\hat{\boldsymbol{m}} = 0$$

$$\iff \underset{\tilde{\boldsymbol{T}}}{\operatorname{argmin}}\, r(\tilde{\boldsymbol{T}}) + \frac{1}{2}\sum_{k=1}^{K}\boldsymbol{\Sigma}_k\tilde{\boldsymbol{T}}\hat{\boldsymbol{S}}_k^{-1}\tilde{\boldsymbol{T}} - \left(\sum_{k=1}^{K}\boldsymbol{\Sigma}_k^{1/2}\hat{\boldsymbol{U}}_k\hat{\boldsymbol{S}}_k^{-1} + \frac{1}{\sqrt{d}}\boldsymbol{M}^\top\hat{\boldsymbol{m}}\right)\tilde{\boldsymbol{T}} \tag{67}$$

We then recover $\boldsymbol{T}$ from $\boldsymbol{T} = \boldsymbol{\Sigma}\tilde{\boldsymbol{T}}$. Thus, defining the function

$$\boldsymbol{\eta} : \mathbb{R}^{d \times K^2} \to \mathbb{R}^{d \times K}$$

$$\hat{\boldsymbol{U}} \mapsto \underset{\tilde{\boldsymbol{T}}}{\operatorname{argmin}}\, r(\tilde{\boldsymbol{T}}) + \frac{1}{2}\sum_{k=1}^{K}\boldsymbol{\Sigma}_k\tilde{\boldsymbol{T}}\hat{\boldsymbol{S}}_k^{-1}\tilde{\boldsymbol{T}} - \left(\sum_{k=1}^{K}\boldsymbol{\Sigma}_k^{1/2}\hat{\boldsymbol{U}}_k\hat{\boldsymbol{S}}_k^{-1} + \frac{1}{\sqrt{d}}\boldsymbol{M}^\top\hat{\boldsymbol{m}}\right)\tilde{\boldsymbol{T}} \tag{68}$$

the block decomposition of the resolvent for the regularizer reads:

$$\boldsymbol{R}_{\tilde{r},\hat{\boldsymbol{S}},k}(\hat{\boldsymbol{U}}) = \boldsymbol{\Sigma}_k^{1/2}\boldsymbol{\eta}(\hat{\boldsymbol{U}}) \tag{69}$$

**Matching the optimality condition with the AMP fixed point** The global optimality condition then reads:

$$\left[\boldsymbol{Z}_k^\top\left(\boldsymbol{R}_{\tilde{L}_k,\boldsymbol{S}_k}(\boldsymbol{U}_k) - \boldsymbol{U}_k\right)\boldsymbol{S}_k^{-1}\right]_{k=1}^{K} = \left[(\hat{\boldsymbol{U}}_k - \boldsymbol{R}_{\tilde{r},\hat{\boldsymbol{S}},k}(\hat{\boldsymbol{U}}))\hat{\boldsymbol{S}}_k^{-1}\right]_{k=1}^{K} \tag{70}$$

$$\left[\boldsymbol{Z}_k\boldsymbol{R}_{\tilde{r},\hat{\boldsymbol{S}},k}(\hat{\boldsymbol{U}})\right]_{k=1}^{K} = \left[\boldsymbol{R}_{\tilde{L}_k,\boldsymbol{S}_k}(\boldsymbol{U}_k)\right]_{k=1}^{K} \tag{71}$$

where both equations should be satisfied. We can now define update functions based on the previously obtained block decomposition. The fixed point of the matrix-valued AMP Eq.(26) reads:

$$\mathrm{Id} + \boldsymbol{e}(\boldsymbol{u})\langle\boldsymbol{h}'\rangle^\top = \boldsymbol{Z}^\top\boldsymbol{h}(\boldsymbol{v}) \tag{72}$$

$$\mathrm{Id} + \boldsymbol{h}(\boldsymbol{v})\langle\boldsymbol{e}'\rangle^\top = \boldsymbol{Z}\boldsymbol{e}(\boldsymbol{u}) \tag{73}$$

Matching this fixed point with the optimality condition Eq.(70) suggests the following mapping:

$$\begin{aligned}
\boldsymbol{h}_k(\boldsymbol{U}_k) &= \left(\boldsymbol{R}_{\tilde{L}_k,\boldsymbol{S}_k}(\boldsymbol{U}_k) - \boldsymbol{U}_k\right)\boldsymbol{S}_k^{-1}, & \boldsymbol{S}_k &= \langle\boldsymbol{e}_k'\rangle, \\
\boldsymbol{e}_k(\hat{\boldsymbol{U}}) &= \boldsymbol{R}_{\tilde{r},\hat{\boldsymbol{S}},k}(\hat{\boldsymbol{U}}\hat{\boldsymbol{S}}), & \hat{\boldsymbol{S}}_k &= -\langle\boldsymbol{h}_k'\rangle^{-1},
\end{aligned} \tag{74}$$

where we redefined $\hat{\boldsymbol{U}} \equiv \hat{\boldsymbol{U}}\hat{\boldsymbol{S}}$ in (56), and the subscripts on the non-linearities are block indexes.

## A.4 Proof of Theorem 1 using the AMP sequence

Following the analysis carried out in the previous section, define the following two sequences of non-linearities, for fixed values of the parameters $\hat{m}, m, b$ and any $u \in \mathbb{R}^{d \times K^2}, v \in \mathbb{R}^{n \times K}$:

$$e_t : \mathbb{R}^{Kd \times K^2} \to \mathbb{R}^{Kd \times K^2}$$

$$u \mapsto \begin{bmatrix} e_{1,t}(u) & & & \\ & e_{2,t}(u) & (0) & \\ & (0) & \ddots & \\ & & & e_{K,t}(u) \end{bmatrix} \in \mathbb{R}^{Kd \times K^2} \tag{75}$$

$$h_t : \mathbb{R}^{n \times K^2} \to \mathbb{R}^{n \times K^2}$$

$$v \mapsto \begin{bmatrix} h_{1,t}(v_1) & & & \\ & h_{2,t}(v_2) & (0) & \\ & (0) & \ddots & \\ & & & h_{K,t}t(v_K) \end{bmatrix} \in \mathbb{R}^{n \times K^2} \tag{76}$$

where $Y_k \in \mathbb{R}^{n_k \times K}$ and

$$h_{k,t} : \mathbb{R}^{n_k \times K} \to \mathbb{R}^{n_k \times K}$$

$$v_k \mapsto \left( R_{\tilde{L}_k, V^{k,t}}(v_k) - v_k \right) (V^{k,t})^{-1}$$

$$= \left( \underset{T \in \mathbb{R}^{n_k \times K}}{\operatorname{argmin}} \left\{ \tilde{L}_k(T) + \frac{1}{2} \operatorname{tr} \left( (T - v_k)(V_{k,t})^{-1}(T - v_k)^\top \right) \right\} - v_k \right) (V_{k,t})^{-1}$$

$$= \left( \operatorname{Prox}_{L_k(\bullet(V_{k,t})^{1/2})}((Y_k m + v_k)(V_{k,t})^{-1/2})(V_{k,t})^{1/2} - (Y_k m + v_k) \right)(V_{k,t})^{-1} \tag{77}$$

$$e_{k,t} : \mathbb{R}^{d \times K^2} \to \mathbb{R}^{d \times K}$$

$$u \mapsto \Sigma_k^{1/2} \underset{\tilde{T} \in \mathbb{R}^{d \times K}}{\operatorname{argmin}} r(\tilde{T}) + \frac{1}{2} \sum_{k=1}^K \Sigma_k \tilde{T} \hat{V}_{k,t} \tilde{T} - \left( \sum_{k=1}^K \Sigma_k^{1/2} u_k + \frac{1}{\sqrt{d}} M^\top \hat{m} \right) \tilde{T}$$

$$= \Sigma_k^{1/2} \eta(u(\hat{V}^t)^{-1}) \tag{78}$$

where $(V_t, \hat{V}_t) \in \mathbb{R}^{K^2 \times K^2}$, are defined as the block diagonal matrices $[V_{k,t}]_{k \in [K]}, \left[ \hat{V}_{k,t} \right]_{k \in [K]}$ such that

$$V_{k,t} = \langle (e_{k,t-1})' \rangle \quad \hat{V}_{k,t} = -\langle (h_{k,t})' \rangle \tag{79}$$

using the notation from Eq. (28). Now define the following sequence, initialized with

$$u^0, h^{-1} \equiv 0, \hat{V}_0 \tag{80}$$

such that $\lim_{d \to \infty} \frac{1}{d} \left\| e_0(u^0)^\top e_0(u^0) \right\|_F < +\infty$ and $\hat{V}_0 \in \mathbb{S}_K^{++}$

and recursively define

$$u^{t+1} = Z^\top h_t(v^t) - e_t(u^t) \langle h_t' \rangle^\top \tag{81}$$

$$v^t = Z e_t(u^t) - h_{t-1}(v^{t-1}) \langle e_t' \rangle^\top \tag{82}$$

where $Z \in \mathbb{R}^{n \times Kd}$ has i.i.d. $\mathcal{N}(0, 1/d)$ elements, and in the Jacobians defining $\hat{V}, V$, we used the notation from Eq. (23).

**State evolution equations** The results from section A.3 show that the functions $e^t, h^t$ are proximals operators, and thus are Lipschitz continuous for all $t \in \mathbb{N}$, along with their block restrictions. Therefore the conditions of Theorem 4 are verified and we have the following lemma:

**Lemma 5.** *Consider the sequence defined by Eq.(81), for any fixed $\boldsymbol{m}, \hat{\boldsymbol{m}}, \boldsymbol{b}$. For any sequences of pseudo-Lipschitz functions $\phi_{1,n} : \mathbb{R}^{d \times K^2} \to \mathbb{R}, \phi_{2,n} : \mathbb{R}^{n \times K^2} \to \mathbb{R}$, for any $t \in \mathbb{N}^*$:*

$$\phi_{1,n}(\boldsymbol{u}_1^t, \ldots, \boldsymbol{u}_K^t) \overset{\mathrm{P}}{\simeq} \mathbb{E}\left[\phi_{1,n}(\boldsymbol{H}_1(\hat{\boldsymbol{Q}}_{1,t})^{1/2}, \ldots, \boldsymbol{H}_K(\hat{\boldsymbol{Q}}_{K,t})^{1/2})\right] \tag{83}$$

$$\phi_{2,n}(\boldsymbol{v}_1^t, \ldots, \boldsymbol{v}_K^t) \overset{\mathrm{P}}{\simeq} \mathbb{E}\left[\phi_{1,n}(\boldsymbol{G}_1(\boldsymbol{Q}_{1,t})^{1/2}, \ldots, \boldsymbol{G}_K(\boldsymbol{Q}_{K,t})^{1/2})\right] \tag{84}$$

*where the matrices $\boldsymbol{H}_k \in \mathbb{R}^{d \times K}, \boldsymbol{G}_k \in \mathbb{R}^{n_k \times K}$ are independent matrices with i.i.d. standard normal elements, and at each time step $t \geqslant 1$*

$$\boldsymbol{Q}_{k,t} = \lim_{d \to +\infty} \frac{1}{d} \mathbb{E}\left[\boldsymbol{e}_{k,t}(\{\boldsymbol{H}_k(\hat{\boldsymbol{Q}}_{k,t})^{1/2}(\hat{\boldsymbol{V}}_{k,t})^{-1}\}_{k \in [K]})^\top \boldsymbol{e}_{k,t}(\{\boldsymbol{H}_k(\hat{\boldsymbol{Q}}_{k,t})^{1/2}(\hat{\boldsymbol{V}}_{k,t})^{-1}\}_{k \in [K]})\right] \tag{85}$$

$$\in \mathbb{R}^{K \times K}$$

$$\hat{\boldsymbol{Q}}_{k,t} = \lim_{d \to +\infty} \frac{1}{d} \mathbb{E}\left[\boldsymbol{h}_{k,t-1}(\boldsymbol{G}_k(\boldsymbol{Q}_{k,t-1})^{1/2})^\top \boldsymbol{h}_{k,t-1}(\boldsymbol{G}_k(\boldsymbol{Q}_{k,t-1})^{1/2})\right] \in \mathbb{R}^{K \times K} \tag{86}$$

$$\boldsymbol{V}_{k,t} = \lim_{d \to +\infty} \frac{1}{d} \sum_{i=1}^{d} \frac{\partial \boldsymbol{e}_{k,t-1}(\{\boldsymbol{H}_k(\hat{\boldsymbol{Q}}_{k,t-1})^{1/2}\}_{k \in [K]})}{\partial(\boldsymbol{H}_k(\hat{\boldsymbol{Q}}_{k,t-1})^{1/2})_i} \in \mathbb{R}^{K \times K} \tag{87}$$

$$\hat{\boldsymbol{V}}_{k,t} = -\lim_{d \to +\infty} \frac{1}{d} \sum_{i=1}^{n_k} \frac{\partial \boldsymbol{h}_{k,t}(\boldsymbol{G}_k(\boldsymbol{Q}_{k,t})^{1/2})}{\partial(\boldsymbol{G}_k(\boldsymbol{Q}_{k,t})^{1/2})_i} \in \mathbb{R}^{K \times K} \tag{88}$$

*where the sequence is initialized with $\hat{\boldsymbol{V}}_0, \boldsymbol{e}_0, \boldsymbol{Q}_{0,0} = \lim_{d \to \infty} \frac{1}{d} \|\boldsymbol{e}_0(\boldsymbol{u}^0)^\top \boldsymbol{e}_0(\boldsymbol{u}^0)\|_{\mathrm{F}}$.*

*Proof.* Lemma 5 is a consequence of Theorem 4 whose assumptions have been verified in the paragraph. $\square$

Note that in Lemma 5, we have directly written the block decomposition of the state evolution corresponding to the iteration Eq. (81), which involves the block diagonal matrices $\boldsymbol{Q}_t, \hat{\boldsymbol{Q}}_t, \boldsymbol{V}_t, \hat{\boldsymbol{V}}_t$ which are all in $\mathbb{R}^{K^2 \times K^2}$. Using the notations introduced in section A.1

$$\boldsymbol{V} = [\boldsymbol{V}_k]_{k=1}^K \quad \hat{\boldsymbol{V}} = \left[\hat{\boldsymbol{V}}_k\right]_{k=1}^K \quad \boldsymbol{Q} = [\boldsymbol{Q}_k]_{k=1}^K \quad \hat{\boldsymbol{Q}} = \left[\hat{\boldsymbol{Q}}_k\right]_{k=1}^K \tag{89}$$

Also note that we do not use the full state evolution giving the correlations across all time steps, but only use those at equal times $t$.

**Trajectories and fixed point of the AMP sequence** Now that we have a sequence with state evolution equations, the following two lemmas link the fixed points of this iteration to any optimal solution of problem Eq.(47).

**Lemma 6.** *Consider any fixed point $\boldsymbol{V}, \hat{\boldsymbol{V}}, \boldsymbol{Q}, \hat{\boldsymbol{Q}}$ of the state evolution equations from Lemma 5. For any fixed point $\boldsymbol{u}^*, \boldsymbol{v}^*$ of iteration Eq.(81), the quantity*

$$\boldsymbol{R}_{\tilde{r}, \hat{\boldsymbol{V}}^{-1}}(\boldsymbol{u}^* \hat{\boldsymbol{V}}^{-1}) = \left(\mathrm{Id} + \partial \tilde{r}(\bullet)\hat{\boldsymbol{V}}^{-1}\right)(\boldsymbol{u}^* \hat{\boldsymbol{V}}^{-1}) \tag{90}$$

*is an optimal solution $\tilde{\boldsymbol{W}}^\star$ of problem Eq.( 47). Furthermore*

$$\boldsymbol{R}_{\tilde{L}, \boldsymbol{V}}(\boldsymbol{v}^*) = (\mathrm{Id} + \partial \tilde{L}(\bullet)\boldsymbol{V})(\boldsymbol{v}^*) = \boldsymbol{Z}\tilde{\boldsymbol{W}}^\star \tag{91}$$

*where the block decompositions of each resolvents have been explicitly calculated in section A.3.*

*Proof.* Lemma 6 is a direct consequence of the analysis carried out in section A.3. $\square$

At this point we know the fixed points of the AMP iteration correspond to the optimal solutions of problem Eq.(47). Note that the resolvents/proximals linking the fixed point of the AMP iteration with the solutions of Eq.(47) are Lipschitz continuous, making them acceptable transforms for state evolution observables. However this does not guarantee that the optimal solution is characterized by the fixed point of the state evolution equations. Indeed, we need to show that a converging trajectory can be systematically found for any instance of the problem Eq.(47). This is the purpose of the following lemma.

**Lemma 7.** *Consider iteration Eq.(81), where the parameters $\boldsymbol{Q}, \hat{\boldsymbol{Q}}, \boldsymbol{V}, \hat{\boldsymbol{V}}$ are initialized at any fixed point of the state evolution equations of Lemma 5. For any sequence initialized with $\hat{\boldsymbol{V}}_0 = \hat{\boldsymbol{V}}$ and $\boldsymbol{u}^0$ such that*

$$\lim_{d \to \infty} \frac{1}{d} \boldsymbol{e}_0(\boldsymbol{u}^0)^\top \boldsymbol{e}_0(\boldsymbol{u}^0) = \boldsymbol{Q} \tag{92}$$

*the following holds*

$$\lim_{t \to \infty} \lim_{d \to \infty} \frac{1}{\sqrt{d}} \|\boldsymbol{u}^t - \boldsymbol{u}^\star\|_{\mathrm{F}} = 0 \quad \lim_{t \to \infty} \lim_{d \to \infty} \frac{1}{\sqrt{d}} \|\boldsymbol{v}^t - \boldsymbol{v}^\star\|_{\mathrm{F}} = 0 \tag{93}$$

*Proof.* The proof of Lemma 7 is deferred to subsection A.7. □

Note that the $\boldsymbol{G}$ defined here is not the same as the $\boldsymbol{G}$ in the replica computation. Combining the lemmas 5, 6 and 7 with the pseudo-Lipschitz property, we have reached the following lemma

**Lemma 8.** *For any fixed $\boldsymbol{m}, \hat{\boldsymbol{m}}, \boldsymbol{b}$, consider the fixed point $(\boldsymbol{Q}, \hat{\boldsymbol{Q}}, \boldsymbol{V}, \hat{\boldsymbol{V}})$ of the state evolution equations from Lemma. 5. Then, for any sequences of pseudo-Lipschitz functions $\phi_{1,n} : \mathbb{R}^{d \times K^2} \to \mathbb{R}, \phi_{2,n} : \mathbb{R}^{n \times K} \to \mathbb{R}$, for $n, d \to \infty$*

$$\phi_{1,n}(\tilde{\boldsymbol{W}}^\star) \overset{\mathrm{P}}{\simeq} \mathbb{E}\left[\phi_{1,n}\left(R_{\tilde{r}, \hat{\boldsymbol{V}}}(\boldsymbol{H}\hat{\boldsymbol{Q}}^{1/2}\hat{\boldsymbol{V}}^{-1})\right)\right] \tag{94}$$

$$\phi_{2,n}(\boldsymbol{Z}\tilde{\boldsymbol{W}}^\star) \overset{\mathrm{P}}{\simeq} \mathbb{E}\left[\phi_{2,n}\left(R_{\tilde{L}, \boldsymbol{V}}(\boldsymbol{G}\boldsymbol{Q}^{1/2})\right)\right] \tag{95}$$

*where we remind that $\boldsymbol{G} = [\boldsymbol{G}_k]_{k=1}^K, \boldsymbol{H} = [\boldsymbol{H}_k]_{k=1}^K$ are block diagonal i.i.d. standard normal matrices as in Lemma 5, and $\boldsymbol{Q} = [\boldsymbol{Q}_k]_{k=1}^K$ $\hat{\boldsymbol{Q}} = \left[\hat{\boldsymbol{Q}}_k\right]_{k=1}^K$ are the $K^2 \times K^2$ block diagonal covariances.*

*Proof.* Lemma 8 is a consequence of Lemmas 5,6,7 and applying the pseudo-Lipschitz property along with the fact that the iterates of the AMP have bounded norm using the state evolution and that the estimator also has bounded norm (feasibility assumption). Note that, for a generically non-strictly convex problem, being close to the zero gradient condition does not guarantee being close to the estimator. This is further discussed in Appendix A.5. □

Note that the resolvents are implicitly acting on the block diagonals of their arguments. At this point we are quite close to Theorem 1(details for the exact matching will be given later), but we are missing the equations on $\boldsymbol{m}, \hat{\boldsymbol{m}}, \boldsymbol{b}$.

**Fixed point equations for $\boldsymbol{m}, \hat{\boldsymbol{m}}, \boldsymbol{b}$** We drop the dependence on the bias term $\boldsymbol{b}$ as its solution is very similar to the one for $\boldsymbol{m}, \hat{\boldsymbol{m}}$. To obtain the equations for $\boldsymbol{m}, \hat{\boldsymbol{m}}$, we go back to the complete optimization problem

$$\inf_{\boldsymbol{m}, \tilde{\boldsymbol{W}}, \boldsymbol{b}} \sup_{\hat{\boldsymbol{m}}} L(\boldsymbol{Y}_k \boldsymbol{m} + \boldsymbol{Z}_k \tilde{\boldsymbol{W}}_k) + r\left(\boldsymbol{\Sigma}^+ \tilde{\boldsymbol{W}}\right)$$
$$+ \operatorname{tr}\left(\hat{\boldsymbol{m}}^\top \left(\boldsymbol{m} - \frac{1}{\sqrt{d}} \boldsymbol{M}\boldsymbol{\Sigma}^+ \tilde{\boldsymbol{W}}\right)\right) \tag{96}$$

where we can use strong duality to write the equivalent form

$$\inf_{\boldsymbol{m}, \boldsymbol{b}} \sup_{\hat{\boldsymbol{m}}} L(\boldsymbol{Y}_k \boldsymbol{m} + \boldsymbol{Z}_k \tilde{\boldsymbol{W}}_k^\star) + r\left(\boldsymbol{\Sigma}^+ \tilde{\boldsymbol{W}}\right)$$
$$+ \operatorname{tr}\left(\hat{\boldsymbol{m}}^\top \left(\boldsymbol{m} - \frac{1}{\sqrt{d}} \boldsymbol{M}\boldsymbol{\Sigma}^+ \tilde{\boldsymbol{W}}^\star\right)\right) \tag{97}$$

The gradients w.r.t. $\boldsymbol{m}, \hat{\boldsymbol{m}}$ then read:

$$\partial \hat{\boldsymbol{m}} = \boldsymbol{m} - \frac{1}{\sqrt{d}} \boldsymbol{M}\boldsymbol{\Sigma}^+ \tilde{\boldsymbol{W}}^\star \tag{98}$$

$$\partial \boldsymbol{m} = \hat{\boldsymbol{m}} + \partial_{\boldsymbol{m}} L(\boldsymbol{Y}\boldsymbol{m} + \boldsymbol{Z}\tilde{\boldsymbol{W}}^\star) \tag{99}$$

Uniform convergence of derivatives and conditions for the dominated convergence theorem are verified using similar arguments as in [12, Lemma 12]. We can thus invert limits and derivatives, and expectations and derivatives. To facilitate taking the derivative $\partial_{\boldsymbol{m}}$, we use Lemma 8 (assuming the normalized loss function is pseudo-Lipschitz, which is a very loose assumption verified by most machine learning losses) to obtain, reintroducing the scaling $1/d$

$$\frac{1}{d}L(\boldsymbol{Y}\boldsymbol{m} + \boldsymbol{Z}\tilde{\boldsymbol{W}}^\star) \xrightarrow[d\to\infty]{P} \frac{1}{d}\mathbb{E}\left[L(\boldsymbol{Y}\boldsymbol{m} + \boldsymbol{R}_{\tilde{L},\boldsymbol{V}}(\boldsymbol{G}\boldsymbol{Q}^{1/2}))\right] \tag{100}$$

Using the block decomposition from Eq.(53), the blocks $(\boldsymbol{R}_{\tilde{L},\boldsymbol{V}}(\boldsymbol{G}\boldsymbol{Q}^{1/2}))_k \in \mathbb{R}^{n_k \times K}$ are given by:

$$\operatorname*{argmin}_{\boldsymbol{T}\in\mathbb{R}^{n_k\times K}}\left\{L_k(\boldsymbol{T})+\frac{1}{2}\operatorname{tr}\left((\boldsymbol{T}-(\boldsymbol{Y}_k\boldsymbol{m}+\boldsymbol{G}_k\boldsymbol{Q}_k^{1/2}))\boldsymbol{V}_k^{-1}(\boldsymbol{T}-(\boldsymbol{Y}_k\boldsymbol{m}+\boldsymbol{G}_k\boldsymbol{Q}_k^{1/2}))^\top\right)\right\}-\boldsymbol{Y}_k\boldsymbol{m} \tag{101}$$

Using a block diagonal representation, we can write:

$$\frac{1}{d}L(\boldsymbol{Y}\boldsymbol{m} + \boldsymbol{R}_{\tilde{L},\boldsymbol{V}}(\boldsymbol{G}\boldsymbol{Q}^{1/2})) = \frac{1}{d}L(R_{L,\boldsymbol{V}}(\boldsymbol{Y}\boldsymbol{m} + \boldsymbol{G}\boldsymbol{Q}^{1/2}))$$
$$= \frac{1}{d}\mathcal{M}_{L,\boldsymbol{V}}(\boldsymbol{Y}\boldsymbol{m} + \boldsymbol{G}\boldsymbol{Q}^{1/2})-$$
$$\frac{1}{2d}\operatorname{tr}\left((\boldsymbol{R}_{L,\boldsymbol{V}}(\boldsymbol{Y}\boldsymbol{m}+\boldsymbol{G}\boldsymbol{Q}^{1/2})-(\boldsymbol{Y}\boldsymbol{m}+\boldsymbol{G}\boldsymbol{Q}^{1/2}))\boldsymbol{V}^{-1}(\boldsymbol{R}_{L,\boldsymbol{V}}(\boldsymbol{Y}\boldsymbol{m}+\boldsymbol{G}\boldsymbol{Q}^{1/2})-(\boldsymbol{Y}\boldsymbol{m}+\boldsymbol{G}\boldsymbol{Q}^{1/2}))^\top\right) \tag{102}$$

where we have introduced the Bregman-envelope [65] with respect to the distance Eq. (17)

$$\mathcal{M}_{L,\boldsymbol{V}}(\boldsymbol{Y}\boldsymbol{m} + \boldsymbol{G}\boldsymbol{Q}^{1/2}) =$$
$$\min_{\boldsymbol{T}}\left\{L(\boldsymbol{T}) + \frac{1}{2}\operatorname{tr}\left((\boldsymbol{T}-(\boldsymbol{Y}\boldsymbol{m}+\boldsymbol{G}\boldsymbol{Q}^{1/2}))\boldsymbol{V}^{-1}(\boldsymbol{T}-(\boldsymbol{Y}\boldsymbol{m}+\boldsymbol{G}\boldsymbol{Q}^{1/2}))^\top\right)\right\} \tag{103}$$

Then, using the state evolution equations from Lemma 5 and Stein's lemma, we can write:

$$\frac{1}{d}L(\boldsymbol{Y}\boldsymbol{m} + \boldsymbol{R}_{\tilde{L},\boldsymbol{V}}(\boldsymbol{G}\boldsymbol{Q}^{1/2})) = \frac{1}{d}\mathcal{M}_{L,\boldsymbol{V}}(\boldsymbol{Y}\boldsymbol{m} + \boldsymbol{G}\boldsymbol{Q}^{1/2}) - \frac{1}{2}\operatorname{tr}(\boldsymbol{V}^\top\boldsymbol{Q}) \tag{104}$$

Taking the gradient w.r.t. $\boldsymbol{m}$ using the expression for the derivative of a Bregman envelope [65], we get:

$$\partial_{\boldsymbol{m}}L(\boldsymbol{Y}\boldsymbol{m} + \boldsymbol{R}_{\tilde{L},\boldsymbol{V}}(\boldsymbol{G}\boldsymbol{Q}^{1/2})) = \frac{1}{d}\boldsymbol{Y}^\top\left(\boldsymbol{Y}\boldsymbol{m} + \boldsymbol{G}\boldsymbol{Q}^{1/2} - \boldsymbol{R}_{L,\boldsymbol{V}}(\boldsymbol{Y}\boldsymbol{m} + \boldsymbol{G}\boldsymbol{Q}^{1/2})\right)\boldsymbol{V}^{-1} \tag{105}$$

which prescribes, using Lemma 8

$$\hat{m} \overset{\mathrm{P}}{\simeq} \frac{1}{d}\boldsymbol{Y}^\top\left(\boldsymbol{R}_{L,\boldsymbol{V}}(\boldsymbol{Y}\boldsymbol{m} + \boldsymbol{G}\boldsymbol{Q}^{1/2}) - \boldsymbol{Y}\boldsymbol{m} + \boldsymbol{G}\boldsymbol{Q}^{1/2}\right)\boldsymbol{V}^{-1} \tag{106}$$

For $\boldsymbol{m}$, we use the block decomposition from Eq.(67), which simplifies the pseudo-inverse $\boldsymbol{\Sigma}^+$ in Eq. (98) to give, using Lemma 8 again

$$m \overset{\mathrm{P}}{\simeq} \frac{1}{\sqrt{d}}\boldsymbol{M}\boldsymbol{\eta}(\boldsymbol{H}\hat{\boldsymbol{Q}}^{1/2}\hat{\boldsymbol{V}}^{-1}) \tag{107}$$

where the function $\boldsymbol{\eta}$ acts on the block diagonal and is defined by Eq.(68). Using those results and the definition of $\tilde{\boldsymbol{W}}$, the solution $\boldsymbol{W}^\star$ and the quantity $\boldsymbol{X}\boldsymbol{W}^\star$ are characterized, in the pseudo-Lipschitz sense of Theorem 1, by the fixed point of the system of equations (the first four equations are meant

for all $1 \leqslant k \leqslant K$):

$$\boldsymbol{Q}_k = \lim_{d \to +\infty} \frac{1}{d} \mathbb{E}\left[ \boldsymbol{e}_k(\{\boldsymbol{H}_k(\hat{\boldsymbol{Q}}_k)^{1/2}\hat{\boldsymbol{V}}_k^{-1}\}_{k\in[K]})^\top \boldsymbol{e}_k(\{\boldsymbol{H}_k(\hat{\boldsymbol{Q}}_k)^{1/2}\hat{\boldsymbol{V}}_k^{-1}\}_{k\in[K]}) \right] \in \mathbb{R}^{K\times K} \quad (108)$$

$$\hat{\boldsymbol{Q}}_k = \lim_{d \to +\infty} \frac{1}{d} \mathbb{E}\left[ \boldsymbol{h}_k(\boldsymbol{G}_k \boldsymbol{Q}_k^{1/2})^\top \boldsymbol{h}_k(\boldsymbol{G}_k \boldsymbol{Q}_k^{1/2}) \right] \in \mathbb{R}^{K\times K} \quad (109)$$

$$\boldsymbol{V}_k = \lim_{d \to +\infty} \frac{1}{d} \sum_{i=1}^{d} \mathbb{E}\left[ \frac{\partial \boldsymbol{e}_k(\{\boldsymbol{H}_k(\hat{\boldsymbol{Q}}_k)^{1/2}\}_{k\in[K]})}{\partial(\boldsymbol{H}_k(\hat{\boldsymbol{Q}}_k)^{1/2})_i} \right] \in \mathbb{R}^{K\times K} \quad (110)$$

$$\hat{\boldsymbol{V}}_k = -\lim_{d \to +\infty} \frac{1}{d} \sum_{i=1}^{n_k} \mathbb{E}\left[ \frac{\partial \boldsymbol{h}_{k,t}(\boldsymbol{G}_k(\boldsymbol{Q}_{k,t})^{1/2})}{\partial(\boldsymbol{G}_k(\boldsymbol{Q}_k)^{1/2})_i} \right] \in \mathbb{R}^{K\times K} \quad (111)$$

$$\boldsymbol{m} = \frac{1}{\sqrt{d}} \mathbb{E}\left[ \boldsymbol{M}\eta(\boldsymbol{H}\hat{\boldsymbol{Q}}^{1/2}\hat{\boldsymbol{V}}^{-1}) \right] \in \mathbb{R}^{K\times K} \quad (112)$$

$$\hat{\boldsymbol{m}} = \frac{1}{d} \boldsymbol{Y}^\top \left( \boldsymbol{R}_{L,\boldsymbol{V}}(\boldsymbol{Y}\boldsymbol{m} + \boldsymbol{G}\boldsymbol{Q}^{1/2}) - \boldsymbol{Y}\boldsymbol{m} + \boldsymbol{G}\boldsymbol{Q}^{1/2} \right) \boldsymbol{V}^{-1} \in \mathbb{R}^{K\times K} \quad (113)$$

Using the explicit form of the different functions given in section A.3 and Stein's lemma for the derivatives, these equations match those of Theorem 1. This completes the proof.

## A.5 On the strict convexity assumption

If the optimization problem defining $\boldsymbol{W}^\star$ is strictly convex, there is only one minimizer and the provided proof is enough. Additionally it is shown in [68] that for any loss function that is strictly convex in its argument and penalized with the $\ell_1$ norm, provided the data is sampled from a continuous distribution, the solution is unique with probability one regardless of the rank of the design matrix. Thus finding a point verifying the optimality condition of (47) is also enough to complete the proof. For generic convex (non-strictly) problems a more careful analysis could be performed in the same spirit as the one of [51]. Empirically the result still holds.

## A.6 On the uniqueness of the solution to the fixed point equations (108)

It is possible to reconstruct Bregman envelopes on problem (47) for the loss and regularization as we have done for the loss in the previous section. We can then show that the fixed point equations (108) are the optimality condition of a convex-concave problem involving both Bregman envelopes and linear combinations of the order parameters. In the same spirit as [12, 49], this problem should be asymptotically strictly convex. This is supported by the simulations presented in the experiments sections but left as an assumption in the main paper.

## A.7 Proof of Lemma 7

This proof follows a similar argument to the one used to control the trajectory of the AMP studied in [50]. Note that, because of the way the AMP is initialized using the fixed point of the state evolution equations, for any $t \geqslant 1$ the following holds:

$$\lim_{d \to +\infty} \frac{1}{d} \mathbb{E}\left[ \boldsymbol{e}(\boldsymbol{u}^t)^\top \boldsymbol{e}(\boldsymbol{u}^t) \right] = \boldsymbol{Q} \in \mathbb{R}^{K^2 \times K^2} \quad (114)$$

$$\lim_{d \to +\infty} \frac{1}{d} \mathbb{E}\left[ \boldsymbol{h}(\boldsymbol{v}^t)^\top \boldsymbol{h}(\boldsymbol{v}^t) \right] = \hat{\boldsymbol{Q}} \in \mathbb{R}^{K^2 \times K^2} \quad (115)$$

where

$$\boldsymbol{e}(\boldsymbol{u}^t) = (Id + \partial\tilde{r}(\bullet)\hat{\boldsymbol{V}}^{-1})^{-1}(\boldsymbol{u}^t\hat{\boldsymbol{V}}^{-1}) \quad \boldsymbol{h}(\boldsymbol{v}^t) = \left( \left( Id + \partial\tilde{L}(\bullet)\boldsymbol{V} \right)^{-1}(\boldsymbol{v}^t) - \boldsymbol{v}^t \right) \boldsymbol{V}^{-1} \quad (116)$$

then the limit we are looking for reads:

$$\lim_{d \to \infty} \frac{1}{d} \|\boldsymbol{u}^t - \boldsymbol{u}^{t-1}\|_F^2 = \lim_{d \to \infty} 2(\hat{\boldsymbol{Q}} - \frac{1}{d}\mathrm{tr}((\boldsymbol{u}^t)^\top\boldsymbol{u}^{t-1}))$$

$$\lim_{d \to \infty} \frac{1}{d} \|\boldsymbol{v}^t - \boldsymbol{v}^{t-1}\|_F^2 = 2(\boldsymbol{Q} - \frac{1}{d}\mathrm{tr}((\boldsymbol{v}^t)^\top\boldsymbol{v}^{t-1})) \quad (117)$$

We thus need to study the correlation between successive iterates. At each time step, denote $(\hat{C}_t, C_t)$ in $\mathbb{R}^{K^2 \times K^2}$ the correlation matrices between iterates at times $t, t-1$ describing the Gaussian fields respectively associated to $\boldsymbol{u}^t, \boldsymbol{v}^t$ i.e.,

$$\lim_{d \to \infty} \frac{1}{d} \text{tr}((\boldsymbol{u}^t)^\top \boldsymbol{u}^{t-1}) = \hat{C}_t \quad \lim_{d \to \infty} \frac{1}{d} \text{tr}((\boldsymbol{v}^t)^\top \boldsymbol{v}^{t-1}) = C_t \tag{118}$$

we can then write the block diagonal Gaussian fields $\hat{\boldsymbol{Z}}^t, \hat{\boldsymbol{Z}}^{t-1}, \boldsymbol{Z}^t, \boldsymbol{Z}^{t-1}$ in $\mathbb{R}^{Kd \times K^2}$ and in the following way

$$\hat{\boldsymbol{Z}}^t \sim \boldsymbol{H}(\hat{C}_t)^{1/2} + \boldsymbol{H}'(\hat{\boldsymbol{Q}} - \hat{C}_t)^{1/2} \tag{119}$$

$$\hat{\boldsymbol{Z}}^{t-1} \sim \boldsymbol{H}(\hat{C}_t)^{1/2} + \boldsymbol{H}''(\hat{\boldsymbol{Q}} - \hat{C}_t)^{1/2} \tag{120}$$

$$\boldsymbol{Z}^t \sim \boldsymbol{G}(C_t)^{1/2} + \boldsymbol{G}'(\boldsymbol{Q} - C_t)^{1/2} \tag{121}$$

$$\boldsymbol{Z}^{t-1} \sim \boldsymbol{G}(C_t)^{1/2} + \boldsymbol{G}''(\boldsymbol{Q} - C_t)^{1/2} \tag{122}$$

where the matrices $\boldsymbol{H}, \boldsymbol{H}', \boldsymbol{H}''$ are in $\mathbb{R}^{Kd \times K^2}$, $\boldsymbol{G}, \boldsymbol{G}', \boldsymbol{G}''$ are in $\mathbb{R}^{n \times K^2}$ and all have i.i.d. standard normal elements. The recursion describing the evolution of these correlations then reads :

$$C_{t+1} = \frac{1}{d}\mathbb{E}\left[e(\boldsymbol{H}\hat{C}_t^{1/2} + \boldsymbol{H}'(\hat{\boldsymbol{Q}} - \hat{C}_t)^{1/2})^\top e(\boldsymbol{H}\hat{C}_t^{1/2} + \boldsymbol{H}''(\hat{\boldsymbol{Q}} - \hat{C}_t)^{1/2})\right] \tag{123}$$

$$\hat{C}_t = \frac{1}{d}\mathbb{E}\left[h(\boldsymbol{G}C_t^{1/2} + \boldsymbol{G}'(\boldsymbol{Q} - C_t)^{1/2})^\top h(\boldsymbol{G}C_t^{1/2} + \boldsymbol{G}''(\boldsymbol{Q} - C_t)^{1/2})\right] \tag{124}$$

Integrating out the independent $\boldsymbol{H}', \boldsymbol{H}''$ first, we get

$$C_{t+1} = \int_{\mathbb{R}^{Kd \times K^2}} d\mu(\boldsymbol{H})\mathbf{I}(H)^\top \mathbf{I}(\boldsymbol{H}) \tag{125}$$

where $\mathbf{I}(\boldsymbol{H}) = \int_{\mathbb{R}^{Kd \times K^2}} d\mu(\boldsymbol{H}')e(\boldsymbol{H}\hat{C}_t^{1/2} + \boldsymbol{H}'(\hat{\boldsymbol{Q}} - \hat{C}_t)^{1/2})$. So $C^t$ is symmetric positive definite, assuming the resolvents aren't trivial. The same argument applied to $\hat{C}^t$ shows it is also symmetric positive definite. From [64], the operators

$$(Id + \partial \tilde{r}(\bullet)\hat{\boldsymbol{V}}^{-1})^{-1}(\bullet) \quad \left(Id + \partial \tilde{L}(\bullet)\boldsymbol{V}\right)^{-1}(\bullet) \tag{126}$$

are *D-firm* w.r.t. the Bregman distances induced by the differentiable, strictly convex functions $\frac{1}{2}\text{tr}(X\hat{\boldsymbol{V}}X^\top)$ and $\frac{1}{2}\text{tr}(\boldsymbol{X}\boldsymbol{V}^{-1}\boldsymbol{X}^\top)$ respectively. Recall

$$\boldsymbol{e}(\boldsymbol{u}^t) = (Id + \partial \tilde{r}(\bullet)\hat{\boldsymbol{V}}^{-1})^{-1}(\boldsymbol{u}^t\hat{\boldsymbol{V}}^{-1}) \quad \boldsymbol{h}(\boldsymbol{v}^t) = \left(\left(Id + \partial \tilde{L}(\bullet)\boldsymbol{V}\right)^{-1}(\boldsymbol{v}^t) - \boldsymbol{v}^t\right)\boldsymbol{V}^{-1} \tag{127}$$

Then, using the definition of *D-firm*

$$\langle \boldsymbol{e}(\hat{\boldsymbol{Z}}^t) - \boldsymbol{e}(\hat{\boldsymbol{Z}}^{t-1}), \left(\boldsymbol{e}(\hat{\boldsymbol{Z}}^t) - \boldsymbol{e}(\hat{\boldsymbol{Z}}^{t-1})\right)\hat{\boldsymbol{V}}\rangle \leqslant \langle \boldsymbol{e}(\hat{\boldsymbol{Z}}^t) - \boldsymbol{e}(\hat{\boldsymbol{Z}}^{t-1}), (\hat{\boldsymbol{Z}}^t - \hat{\boldsymbol{Z}}^{t-1})\hat{\boldsymbol{V}}^{-1}\hat{\boldsymbol{V}}\rangle \tag{128}$$

then, adding the normalization by $\frac{1}{d}$, using the representation Eq.(119-122), taking expectations and applying the matrix form of Stein's lemma, see for example [67] Lemma 12, we get:

$$\text{tr}((\boldsymbol{Q} - C_{t+1})\hat{\boldsymbol{V}}) \leqslant \text{tr}((\hat{\boldsymbol{Q}} - \hat{C}_t)\boldsymbol{V}) \tag{129}$$

Using a similar argument on $\boldsymbol{h}$, we get

$$\text{tr}((\hat{\boldsymbol{Q}} - \hat{C}_t)\boldsymbol{V}) \leqslant \text{tr}((\boldsymbol{Q} - C_t)\hat{\boldsymbol{V}}) \tag{130}$$

and

$$\text{tr}(C_{t+1}\hat{\boldsymbol{V}}) \geqslant \text{tr}(C_t\hat{\boldsymbol{V}}) \tag{131}$$

thus the sequence $\text{tr}(C_{t+1}\hat{\boldsymbol{V}})$ is a bounded (above) monotone (increasing) sequence, and therefore converges. Since $\hat{\boldsymbol{V}}$ is positive definite and given the iteration defining $C_{t+1}$ from $C_t$, any fixed point of this iteration is a fixed point of $\text{tr}(C_t\hat{\boldsymbol{V}})$. Assuming there is only one fixed point to the set of self-consistent equations Eq.(8) (see previous section), the proof is complete. (A similar argument can be carried out on $\hat{C}_t$).

# B  Replica computation

## B.1  Setting of the problem

In this Section we give a full derivation of the results in Theorem 1 and Theorem 2 by means of the replica approach, a standard method developed in the realm of statistical physics of disordered systems [69]. In the general computation, we will consider the classification problem of $K$ clusters, assuming a dataset $\{(\boldsymbol{x}^\nu, \boldsymbol{y}^\nu)\}_{\nu \in [n]}$ of $n$ independent datapoints where, as in the main text, the labels $\boldsymbol{y}$ takes value in a set of $K$ elements, $\boldsymbol{y}^\nu \in \{\boldsymbol{e}_k\}_k$, with $\boldsymbol{e}_k \in \mathbb{R}^L$. The elements of the dataset are independently generated by a mixture density in the form

$$P(\boldsymbol{x}, \boldsymbol{y}) = \sum_{k=1}^{K} \mathbb{I}(\boldsymbol{y} = \boldsymbol{e}_k) \rho_k \mathcal{N}(\boldsymbol{x} \,|\, \boldsymbol{\mu}_k, \boldsymbol{\Sigma}_k), \quad \sum_{k=1}^{K} \rho_k = 1. \tag{132}$$

We will perform our classification task searching for a set of parameters $(\boldsymbol{W}^\star, \boldsymbol{b}^\star)$ that will allow us to construct an estimator. The parameters will be chosen by minimising an empirical risk function in the form

$$\mathcal{R}(\boldsymbol{W}, \boldsymbol{b}) \equiv \sum_{\nu=1}^{n} \ell\left(\boldsymbol{y}^\nu, \frac{\boldsymbol{W}\boldsymbol{x}^\nu}{\sqrt{d}} + \boldsymbol{b}\right) + \lambda r(\boldsymbol{W}), \tag{133}$$

i.e., they are given by

$$(\boldsymbol{W}^\star, \boldsymbol{b}^\star) \equiv \underset{\boldsymbol{W} \in \mathbb{R}^{L \times d}, \, \boldsymbol{b} \in \mathbb{R}^L}{\operatorname{argmin}} \mathcal{R}(\boldsymbol{W}, \boldsymbol{b}). \tag{134}$$

We will say that $\boldsymbol{W} \in \mathbb{R}^{L \times d}$ and $\boldsymbol{b} \in \mathbb{R}^L$ are the weights and bias to be learned respectively, $\ell$ is a convex loss function with respect to its second argument, and $r$ is a regularisation function whose strength is tuned by the parameter $\lambda \geq 0$. Finally, we will assume that a classifier $\boldsymbol{\varphi} \colon \mathbb{R}^L \to \{\boldsymbol{e}_k\}_k$ is given, such that, once $(\boldsymbol{W}^\star, \boldsymbol{b}^\star)$ are obtained, a new point $\boldsymbol{x}$ is assigned to the label

$$\boldsymbol{x} \mapsto \boldsymbol{\varphi}\left(\frac{\boldsymbol{W}^\star \boldsymbol{x}}{\sqrt{d}} + \boldsymbol{b}^\star\right) \in \{\boldsymbol{e}_k\}_k. \tag{135}$$

The described setting is slightly more general than the one given in Theorem 1. As a consequence of the fact that we choose $L$-dimensional labels, the order parameters that appear in the computation are $L$ dimensional vectors or $L \times L$ matrices. A typical "high-dimensional encoding" is the one-hot encoding convention adopted in Theorem 1, where $L = K$ and $\{\boldsymbol{e}_k\}_k \subset \mathbb{R}^K$ is the canonical basis of $\mathbb{R}^K$. In this case, the adopted classifier is

$$\boldsymbol{\varphi}(\boldsymbol{x}) \equiv \hat{\boldsymbol{y}}(\boldsymbol{x}), \quad \hat{y}_k(\boldsymbol{x}) = \mathbb{I}(\max_\kappa x_\kappa = x_k). \tag{136}$$

Assuming *scalar* labels $\{e_k\}_k \in \mathbb{R}$, we deal with scalar order parameters. For example, in the case of binary classification ($K = 2$) it is common to adopt $L = 1$ and $\{e_1, e_2\} = \{+1, -1\}$. In this case $\varphi(x) = \operatorname{sign}(x)$, see also Section C.2.

## B.2  Gibbs minimisation

The problem stated in Section 1 is formulated as an optimisation problem. We can tackle such optimisation problem introducing a Gibbs measure over the weights $(\boldsymbol{W}, \boldsymbol{b})$, namely

$$\mu_\beta(\boldsymbol{W}, \boldsymbol{b}) \propto e^{-\beta \mathcal{R}(\boldsymbol{W}, \boldsymbol{b})} = \underbrace{e^{-\beta r(\boldsymbol{W})}}_{P_w(\boldsymbol{W})} \prod_{\nu=1}^{n} \underbrace{\exp\left[-\beta \ell\left(\boldsymbol{y}^\nu, \frac{\boldsymbol{W}\boldsymbol{x}^\nu}{\sqrt{d}} + \boldsymbol{b}\right)\right]}_{P_y(\boldsymbol{y}|\boldsymbol{W}, \boldsymbol{b})}. \tag{137}$$

The parameter $\beta > 0$ is introduced for convenience: in the $\beta \to +\infty$ limit, the Gibbs measure concentrates on the values $(\boldsymbol{W}^\star, \boldsymbol{b}^\star)$ which minimize the empirical risk $\mathcal{R}(\boldsymbol{W}, \boldsymbol{b})$ and are therefore the goal of the learning process. The functions $P_y$ and $P_w$ can be interpreted as a (unnormalised) likelihood and prior distribution respectively. Our analysis will go through the computation of the average free energy density associated to such Gibbs measure, i.e.,

$$f_\beta = -\lim_{\substack{n,d \to +\infty \\ n/d = \alpha}} \mathbb{E}_{\{(\boldsymbol{x}, \boldsymbol{y})\}}\left[\frac{\ln \mathcal{Z}_\beta}{d\beta}\right], \tag{138}$$

where $\mathbb{E}_{\{(\boldsymbol{x},\boldsymbol{y})\}}[\bullet]$ is the average over the training dataset, and we have introduced the partition function

$$\mathcal{Z}_\beta \equiv \int e^{-\beta \mathcal{R}(\boldsymbol{W},\boldsymbol{b})} \mathrm{d}\boldsymbol{W} \tag{139}$$

To perform the computation of such quantity, we use the so-called replica method, i.e., we compute

$$-\lim_{\substack{n,d\to+\infty \\ n/d=\alpha}} \mathbb{E}_{\{(\boldsymbol{x},\boldsymbol{y})\}}\left[\frac{\ln \mathcal{Z}_\beta}{d\beta}\right] = \lim_{\substack{n,d\to+\infty \\ n/d=\alpha}} \lim_{s\to 0} \frac{1 - \mathbb{E}_{\{(\boldsymbol{x},\boldsymbol{y})\}}[\mathcal{Z}_\beta^s]}{sd\beta}, \tag{140}$$

### B.3 Replica approach

We proceed in our calculation considering the bias vector assuming no prior on $\boldsymbol{b}$, which will play a role of an extra parameter. The equations for the bias $\boldsymbol{b}$ will be derived extremising with respect to it the final result for the free energy. We need to evaluate

$$\mathbb{E}_{\{(\boldsymbol{x},\boldsymbol{y})\}}[\mathcal{Z}_\beta^s] = \prod_{a=1}^s \int \mathrm{d}\boldsymbol{W}^a P_w(\boldsymbol{W}^a) \left(\sum_k \rho_k \mathbb{E}_{\boldsymbol{x}|\boldsymbol{y}=\boldsymbol{e}_k}\left[\prod_{a=1}^s P_y\left(\boldsymbol{e}_k \left| \frac{\boldsymbol{W}^a \boldsymbol{x}}{\sqrt{d}} + \boldsymbol{b}\right.\right)\right]\right)^n. \tag{141}$$

Let us take the inner average introducing a new variable $\boldsymbol{\eta}$,

$$\mathbb{E}_{\boldsymbol{x}|\boldsymbol{y}=\boldsymbol{e}_k}\left[\prod_{a=1}^s P_y\left(\boldsymbol{e}_k\left|\frac{\boldsymbol{W}^a \boldsymbol{x}}{\sqrt{d}} + \boldsymbol{b}\right.\right)\right] = \prod_{a=1}^s \int \mathrm{d}\boldsymbol{\eta}^a P_y(\boldsymbol{e}_k|\boldsymbol{\eta}^a) \mathbb{E}_{\boldsymbol{x}}\left[\prod_{a=1}^s \delta\left(\boldsymbol{\eta}^a - \frac{\boldsymbol{W}^a \boldsymbol{x}}{\sqrt{d}} + \boldsymbol{b}\right)\right]$$

$$= \prod_{a=1}^s \int \mathrm{d}\boldsymbol{\eta}^a P_y(\boldsymbol{e}_k|\boldsymbol{\eta}^a) \mathcal{N}\left(\boldsymbol{\eta}\left|\frac{\boldsymbol{W}^a \boldsymbol{\mu}_k}{\sqrt{d}} - \boldsymbol{b}; \frac{\boldsymbol{W}^a \boldsymbol{\Sigma}_k \boldsymbol{W}^{b\top}}{d}\right.\right). \tag{142}$$

We can write then

$$\mathbb{E}_{\{(\boldsymbol{x},\boldsymbol{y})\}}[\mathcal{Z}_\beta^s] =$$

$$= \prod_{a=1}^n \int \mathrm{d}\boldsymbol{W}^a P_w(\boldsymbol{W}^a) \left(\sum_k \rho_k \prod_{a=1}^s \int \mathrm{d}\boldsymbol{\eta}^a P_y(\boldsymbol{e}_k|\boldsymbol{\eta}^a) \mathcal{N}\left(\boldsymbol{\eta}; \frac{\boldsymbol{W}^a \boldsymbol{\mu}_k}{d} + \boldsymbol{b}; \frac{\boldsymbol{W}^a \boldsymbol{\Sigma}_k \boldsymbol{W}^{b\top}}{d}\right)\right)^n$$

$$= \left(\prod_{k=1}^K \prod_{a\le b} \iint \frac{\mathrm{d}\boldsymbol{Q}_k^{ab} \mathrm{d}\hat{\boldsymbol{Q}}_k^{ab}}{(2\pi)^{L^2/2}}\right) \left(\prod_k \prod_a \int \frac{\mathrm{d}\boldsymbol{m}_k^a \mathrm{d}\hat{\boldsymbol{m}}_k^a}{(2\pi)^{L/2}}\right) e^{-d\beta\Phi^{(s)}}. \tag{143}$$

where we introduced the *order parameters*

$$\boldsymbol{Q}_k^{ab} = \frac{\boldsymbol{W}^a \boldsymbol{\Sigma}_k \boldsymbol{W}^{b\top}}{d} \in \mathbb{R}^{L\times L}, \quad a,b=1,\dots,s, \tag{144}$$

$$\boldsymbol{m}_k^a = \frac{\boldsymbol{W}^a \boldsymbol{\mu}_k}{\sqrt{d}} \in \mathbb{R}^L, \quad a=1,\dots,s, \tag{145}$$

and the replicated free-energy

$$\beta\Phi^{(s)}(\boldsymbol{Q},\boldsymbol{m},\hat{\boldsymbol{Q}},\hat{\boldsymbol{m}},\boldsymbol{b}) = \sum_{k=1}^K \sum_a \hat{\boldsymbol{m}}_k^{a\top} \boldsymbol{m}_k^a + \sum_{k=1}^K \sum_{a\le b} \mathrm{tr}\left[\hat{\boldsymbol{Q}}_k^{ab\top} \boldsymbol{Q}_k^{ab}\right]$$

$$- \frac{1}{d} \ln \prod_{a=1}^s \int P_w(\boldsymbol{W}^a) \mathrm{d}\boldsymbol{W}^a \prod_k \left(\prod_{a\le b} e^{\mathrm{tr}[\hat{\boldsymbol{Q}}_k^{ab\top} \boldsymbol{W}^a \boldsymbol{\Sigma}_k \boldsymbol{W}^{b\top}]} \prod_a e^{\sqrt{d}\hat{\boldsymbol{m}}_k^{a\top} \boldsymbol{W}^a \boldsymbol{\mu}_k}\right)$$

$$- \alpha \ln \sum_k \rho_k \prod_{a=1}^s \int \mathrm{d}\boldsymbol{\eta}^a P_y(\boldsymbol{e}_k|\boldsymbol{\eta}^a) \mathcal{N}\left(\boldsymbol{\eta}|\boldsymbol{m}_k^a + \boldsymbol{b}, \boldsymbol{Q}_k^{ab}\right). \tag{146}$$

At this point, the free energy $f_\beta$ should be computed extremizing with respect to all the order parameters by virtue of the Laplace approximation (in addition to $\boldsymbol{b}$),

$$f_\beta = \lim_{s\to 0} \mathop{\mathrm{Extr}}_{\{\boldsymbol{m},\boldsymbol{Q},\hat{\boldsymbol{m}},\hat{\boldsymbol{Q}}\},\boldsymbol{b}} \frac{\Phi^{(s)}(\boldsymbol{Q},\boldsymbol{m},\hat{\boldsymbol{Q}},\hat{\boldsymbol{m}},\boldsymbol{b})}{s}. \tag{147}$$

However, the convexity of the problem allows us to make an important simplification.

**Replica symmetric ansatz —** Before taking the $s \to 0$ limit we make the assumptions

$$Q_k^{aa} = \begin{cases} \boldsymbol{R}_k, & a = b \\ \boldsymbol{Q}_k & a \neq b \end{cases} \qquad \hat{\boldsymbol{Q}}_k^{aa} = \begin{cases} -\frac{1}{2}\boldsymbol{R}_k, & a = b \\ \hat{\boldsymbol{Q}}_k & a \neq b \end{cases} \tag{148}$$

$$\boldsymbol{m}_k^a = \boldsymbol{m}_k \qquad\qquad \hat{\boldsymbol{m}}_k^a = \hat{\boldsymbol{m}}_k \quad \forall a$$

This ansatz is justified by the fact that we are assuming $\ell$ and $r$ to be convex, and $\lambda > 0$. In this case, the problem admit one solution only that, therefore, coincide with the replica symmetric solution, in which overlaps between two replicas do not depend on the chosen replicas. By means of the replica symmetric hypotesis, we can write

$$\boldsymbol{Q}_k^{ab} \mapsto \mathbf{Q}_k \equiv \mathbf{I}_{s,s} \otimes (\boldsymbol{R}_k - \boldsymbol{Q}_k) + \mathbf{1}_s \otimes \boldsymbol{Q}_k. \tag{149}$$

The inverse matrix is therefore

$$\mathbf{Q}_k^{-1} = \mathbf{1}_s \otimes (\boldsymbol{R}_k - \boldsymbol{Q}_k)^{-1} - \mathbf{I}_{s,s} \otimes [(\boldsymbol{R}_k + (s-1)\boldsymbol{Q}_k)^{-1}\boldsymbol{Q}_k(\boldsymbol{R}_k - \boldsymbol{Q}_k)^{-1}], \tag{150}$$

whereas

$$\begin{aligned} \det \mathbf{Q}_k &= \det(\boldsymbol{R}_k - \boldsymbol{Q}_k)^{s-1} \det(\boldsymbol{R}_k + (s-1)\boldsymbol{Q}_k) \\ &= 1 + s \ln \det(\boldsymbol{R}_k - \boldsymbol{Q}_k) + s \operatorname{tr}\left[(\boldsymbol{R}_k - \boldsymbol{Q}_k)^{-1}\boldsymbol{Q}_k\right] + o(s). \end{aligned} \tag{151}$$

If we denote $\boldsymbol{V}_k \equiv \boldsymbol{R}_k - \boldsymbol{Q}_k$

$$\begin{aligned} \ln \sum_k \rho_k &\prod_{a=1}^s \int \mathrm{d}\boldsymbol{\eta}^a P_y(\boldsymbol{e}_k|\boldsymbol{\eta}^a)\mathcal{N}\left(\boldsymbol{\eta}\big|\boldsymbol{m}_k^a + \boldsymbol{b}, \boldsymbol{Q}_k^{ab}\right) \\ &= s \sum_k \rho_k \mathbb{E}_{\boldsymbol{\xi}} \ln \left(\int \frac{\mathrm{d}\boldsymbol{\eta} P_y(\boldsymbol{e}_k|\boldsymbol{\eta})}{\sqrt{\det(2\pi \boldsymbol{V}_k)}} e^{-\frac{1}{2}(\boldsymbol{\eta} - \boldsymbol{m}_k - \boldsymbol{b} - \boldsymbol{Q}_k^{1/2}\boldsymbol{\xi})^\top \boldsymbol{V}_k^{-1}(\boldsymbol{\eta} - \boldsymbol{b} - \boldsymbol{m}_k - \boldsymbol{Q}_k^{1/2}\boldsymbol{\xi})}\right) + o(s) \\ &= s \sum_k \rho_k \mathbb{E}_{\boldsymbol{\xi}} \left[\ln Z\left(\boldsymbol{e}_k, \boldsymbol{m}_k + \boldsymbol{b} + \boldsymbol{Q}_k^{1/2}\boldsymbol{\xi}, \boldsymbol{V}_k\right)\right] + o(s), \quad (152) \end{aligned}$$

with $\boldsymbol{\xi} \sim \mathcal{N}(\mathbf{0}, \mathbf{I}_L)$ is a normally distributed vector and we have introduced the function

$$Z\left(\boldsymbol{e}_k, \boldsymbol{m}, \boldsymbol{V}\right) \equiv \int \frac{\mathrm{d}\boldsymbol{\eta} P_y(\boldsymbol{e}_k|\boldsymbol{\eta})}{\sqrt{\det(2\pi \boldsymbol{V})}} e^{-\frac{1}{2}(\boldsymbol{\eta} - \boldsymbol{m})^\top \boldsymbol{V}^{-1}(\boldsymbol{\eta} - \boldsymbol{m})} \tag{153}$$

On the other hand, denoting by $\hat{\boldsymbol{V}}_k = \hat{\boldsymbol{R}}_k + \hat{\boldsymbol{Q}}_k$,

$$\begin{aligned} \frac{1}{d}\ln \prod_{a=1}^s &\left(\int P_w(\boldsymbol{W}^a)\mathrm{d}\boldsymbol{W}^a \prod_k e^{-\frac{1}{2}\operatorname{tr}[\hat{\boldsymbol{V}}_k^\top \boldsymbol{W}^a \boldsymbol{\Sigma}_k(\boldsymbol{W}^a)^\top] + \sqrt{d}\hat{\boldsymbol{m}}_k^\top \boldsymbol{W}^a \boldsymbol{\mu}_k} \prod_{b,k} e^{\frac{1}{2}\operatorname{tr}[\hat{\boldsymbol{Q}}_k \boldsymbol{W}^a \boldsymbol{\Sigma}_k(\boldsymbol{W}^b)^\top]}\right) = \\ &= \frac{s}{d}\mathbb{E}_{\boldsymbol{\Xi}}\ln\left[\int P_w(\boldsymbol{W})\mathrm{d}\boldsymbol{W}\prod_k \exp\left(-\frac{\operatorname{tr}\left[\hat{\boldsymbol{V}}_k^\top \boldsymbol{W}\boldsymbol{\Sigma}_k\boldsymbol{W}^\top\right]}{2} + \sqrt{d}\hat{\boldsymbol{m}}_k^\top \boldsymbol{W}\boldsymbol{\mu}_k + \boldsymbol{\Xi}_k \odot \sqrt{\hat{\boldsymbol{Q}}_k \otimes \boldsymbol{\Sigma}_k} \odot \boldsymbol{W}\right)\right] \\ &\qquad\qquad\qquad\qquad\qquad\qquad\qquad\qquad\qquad\qquad\qquad\qquad\qquad\qquad\qquad\qquad + o(s). \quad (154) \end{aligned}$$

In the expression above we have used the tensorial product $\hat{\boldsymbol{Q}} \otimes \boldsymbol{\Sigma} = (\hat{Q}_{kk'}\Sigma_{ij})_{ki,k'j'}$. Given a matrix $\boldsymbol{B} \in \mathbb{R}^{L \times d}$ and the tensors $\mathbf{A}, \mathbf{A}' \in \mathbb{R}^{L \times d} \otimes \mathbb{R}^{L \times d}$, we denote $(\boldsymbol{B} \odot \mathbf{A})_{ki} \equiv \sum_{k'i'} B_{k'i'}A_{k'i'\,ki} \in \mathbb{R}^{L \times d}$, $(\mathbf{A} \odot \boldsymbol{B})_{ki} \equiv \sum_{k'i'} A_{ki\,k'i'}B_{k'i'} \in \mathbb{R}^{L \times d}$ and $(\mathbf{A} \odot \mathbf{A}')_{ki\,k'i'} = \sum_{\kappa j} A_{ki\,\kappa j}A_{\kappa j\,k'i'}$. In this way, we define $\sqrt{\mathbf{A}}$ as the tensor such that $\mathbf{A} = \sqrt{\mathbf{A}} \odot \sqrt{\mathbf{A}}$. Finally, we have also introduced a set of $k$ matrices $\boldsymbol{\Xi}_k \in \mathbb{R}^{L \times d}$ with i.i.d. random Gaussian entries with zero mean and variance 1, and the average over them $\mathbb{E}_{\boldsymbol{\Xi}}[\bullet]$. Therefore, the (replicated) *replica symmetric* free-energy is given by

$$\begin{aligned} \lim_{s \to 0} \frac{\beta}{s}\Phi_{\mathrm{RS}}^{(s)} &= \sum_{k=1}^K \hat{\boldsymbol{m}}_k^\top \boldsymbol{m}_k + \frac{1}{2}\sum_{k=1}^K \operatorname{tr}\left[\hat{\boldsymbol{V}}_k^\top \boldsymbol{Q}_k\right] - \frac{1}{2}\sum_{k=1}^K \operatorname{tr}\left[\hat{\boldsymbol{Q}}_k^\top \boldsymbol{V}_k\right] - \frac{1}{2}\sum_{k=1}^K \operatorname{tr}\left[\hat{\boldsymbol{V}}_k^\top \boldsymbol{V}_k\right] \\ &\quad - \alpha\beta\Psi_{\mathrm{out}}(\boldsymbol{m}, \boldsymbol{Q}, \boldsymbol{V}) - \beta\Psi_w(\hat{\boldsymbol{m}}, \hat{\boldsymbol{Q}}, \hat{\boldsymbol{V}}) \end{aligned} \tag{155}$$

where we have defined two contributions

$$\Psi_{\text{out}}(\boldsymbol{m}, \boldsymbol{Q}, \boldsymbol{V}) \equiv \beta^{-1} \sum_k \rho_k \mathbb{E}_{\boldsymbol{\xi}_k} \ln Z\left(\boldsymbol{e}_k, \boldsymbol{\omega}_k, \boldsymbol{V}_k\right) \tag{156}$$

$$\Psi_w(\hat{\boldsymbol{m}}, \hat{\boldsymbol{Q}}, \hat{\boldsymbol{V}}) \equiv \frac{1}{\beta d} \mathbb{E}_{\boldsymbol{\xi}} \ln \left( \int P_w(\boldsymbol{W}) \mathrm{d}\boldsymbol{W} \prod_k e^{-\frac{\operatorname{tr}\left[\hat{\boldsymbol{V}}_k^\top \boldsymbol{W} \boldsymbol{\Sigma}_k \boldsymbol{W}^\top\right]}{2} + \sqrt{d} \hat{\boldsymbol{m}}_k^\top \boldsymbol{W} \boldsymbol{\mu}_k + \boldsymbol{\Xi}_k \odot \sqrt{\hat{\boldsymbol{Q}}_k \otimes \boldsymbol{\Sigma}_k} \odot \boldsymbol{W}} \right) \tag{157}$$

and introduced, for future convenience,

$$\boldsymbol{\omega}_k \equiv \boldsymbol{m}_k + \boldsymbol{b} + \boldsymbol{Q}_k^{1/2} \boldsymbol{\xi}_k. \tag{158}$$

Note that we have separated the contribution coming from the chosen loss (the so-called *channel* part $\Psi_{\text{out}}$) from the contribution depending on the regularisation (the *prior* part $\Psi_w$). To write down the saddle-point equations in the $\beta \to +\infty$ limit, let us first rescale our order parameters as $\hat{\boldsymbol{m}}_k \mapsto \beta \hat{\boldsymbol{m}}_k$, $\hat{\boldsymbol{Q}}_k \mapsto \beta^2 \hat{\boldsymbol{Q}}_k$, $\hat{\boldsymbol{V}}_k \mapsto \beta \hat{\boldsymbol{V}}_k$ and $\boldsymbol{V}_k \mapsto \beta^{-1} \boldsymbol{V}_k$. For $\beta \to +\infty$ the channel part is

$$\Psi_{\text{out}}(\boldsymbol{m}, \boldsymbol{Q}, \boldsymbol{V}) = -\sum_k \rho_k \mathbb{E}_{\boldsymbol{\xi}} \left[ \mathcal{M}_{\ell(\boldsymbol{e}_k, \boldsymbol{V}_k^{1/2} \bullet)} \left( \boldsymbol{V}_k^{-1/2} \boldsymbol{\omega}_k \right) \right]. \tag{159}$$

Here and in the following the quantity

$$\mathcal{M}_{f(\bullet)}(\mathbf{u}) \equiv \min_{\mathbf{v} \in \text{domain}(\mathbf{v})} \left[ \frac{1}{2} \|\mathbf{v} - \mathbf{u}\|_{\text{F}}^2 + f(\mathbf{v}) \right] \tag{160}$$

is the Moreau envelope of $f: \text{domain}(\mathbf{v}) \to \mathbb{R}$, whereas $\| \bullet \|_{\text{F}}$ is the Frobenius norm. We can write the contribution $\Psi_{\text{out}}$ in terms of a proximal

$$\boldsymbol{h}_k = \boldsymbol{V}_k^{1/2} \text{Prox}_{\ell(\boldsymbol{e}_k, \boldsymbol{V}_k^{1/2} \bullet)}(\boldsymbol{V}_k^{-1/2} \boldsymbol{\omega}_k) \equiv \boldsymbol{V}_k^{1/2} \arg\min_{\mathbf{u} \in \mathbb{R}^L} \left[ \frac{1}{2} \|\mathbf{u} - \boldsymbol{V}_k^{-1/2} \boldsymbol{\omega}_k\|_{\text{F}}^2 + \ell(\boldsymbol{e}_k, \boldsymbol{V}_k^{1/2} \mathbf{u}) \right]. \tag{161}$$

as

$$\Psi_{\text{out}}(\boldsymbol{m}, \boldsymbol{Q}, \boldsymbol{V}) = -\sum_k \rho_k \mathbb{E}_{\boldsymbol{\xi}} \left[ \frac{1}{2} \|\boldsymbol{V}_k^{-1/2} \boldsymbol{h}_k - \boldsymbol{V}_k^{-1/2} \boldsymbol{\omega}_k\|_{\text{F}}^2 + \ell(\boldsymbol{e}_k, \boldsymbol{h}_k) \right] \tag{162}$$

A similar expression can be obtained for $\Psi_w$. Defining

$$\mathbf{A} = \left( \sum_k \hat{\boldsymbol{V}}_k \otimes \boldsymbol{\Sigma}_k \right)^{-1}, \qquad \boldsymbol{B} = \sqrt{d} \sum_k \boldsymbol{\mu}_k \hat{\boldsymbol{m}}_k^\top + \sum_k \boldsymbol{\Xi}_k \odot \sqrt{\hat{\boldsymbol{Q}}_k \otimes \boldsymbol{\Sigma}_k}. \tag{163}$$

$\Psi_w$ can be written as

$$\Psi_w(\hat{\boldsymbol{m}}, \hat{\boldsymbol{Q}}, \hat{\boldsymbol{V}}) = \frac{1}{2d} \mathbb{E}_{\boldsymbol{\xi}} \left[ \boldsymbol{B} \odot \mathbf{A} \odot \boldsymbol{B} \right]$$
$$+ \frac{1}{\beta d} \mathbb{E}_{\boldsymbol{\xi}} \ln \left[ \int \mathrm{d}\boldsymbol{W} \exp\left( -\frac{\beta}{2} \|\mathbf{A}^{-1/2} \odot \boldsymbol{W} - \mathbf{A}^{1/2} \odot \boldsymbol{B}\|_{\text{F}}^2 - \beta r(\boldsymbol{W}) \right) \right]. \tag{164}$$

It follows that, for $\beta \to +\infty$,

$$\Psi_w(\hat{\boldsymbol{m}}, \hat{\boldsymbol{Q}}, \hat{\boldsymbol{V}}) = \frac{1}{2d} \mathbb{E}_{\boldsymbol{\xi}} \left[ \boldsymbol{B} \odot \mathbf{A} \odot \boldsymbol{B} \right] - \frac{1}{d} \mathbb{E}_{\boldsymbol{\xi}} \left[ \mathcal{M}_{r(\mathbf{A}^{1/2} \odot \bullet)}(\mathbf{A}^{1/2} \odot \boldsymbol{B}) \right]. \tag{165}$$

As before, let us introduce the proximal

$$\boldsymbol{G} = \mathbf{A}^{1/2} \odot \text{Prox}_{r(\mathbf{A}^{1/2} \odot \bullet)}(\mathbf{A}^{1/2} \odot \boldsymbol{B}) \in \mathbb{R}^{L \times d} \tag{166}$$

We can rewrite the prior contribution $\Psi_w$ as

$$\Psi_w(\hat{\boldsymbol{m}}, \hat{\boldsymbol{Q}}, \hat{\boldsymbol{V}}) = \frac{1}{2d} \mathbb{E}_{\boldsymbol{\Xi}} \left[ \boldsymbol{B} \odot \mathbf{A} \odot \boldsymbol{B} \right] - \frac{1}{d} \mathbb{E}_{\boldsymbol{\Xi}} \left[ \frac{\|\mathbf{A}^{-1/2} \odot \boldsymbol{G} - \mathbf{A}^{1/2} \odot \boldsymbol{B}\|_{\text{F}}^2}{2} + r(\boldsymbol{G}) \right]. \tag{167}$$

The parallelism between the two contributions is evident, aside from the different dimensionality of the involved objects. The replica symmetric free energy in the $\beta \to +\infty$ limit is computed extremising with respect to the introduced order parameters,

$$f_{\mathrm{RS}} = \operatorname*{Extr}_{\substack{m, Q, V, b \\ \hat{m}, \hat{Q}, \hat{V}}} \left[ \sum_{k=1}^{K} \hat{m}_k^\top m_k + \frac{1}{2} \sum_{k=1}^{K} \mathrm{tr}\left[ \hat{V}_k^\top Q_k \right] - \frac{1}{2} \sum_{k=1}^{K} \mathrm{tr}\left[ \hat{Q}_k^\top V_k \right] \right.$$
$$\left. - \frac{1}{2} \sum_{k=1}^{K} \mathrm{tr}\left[ \hat{V}_k^\top V_k \right] - \alpha \Psi_{\mathrm{out}}(m, Q, V) - \Psi_w(\hat{m}, \hat{Q}, \hat{V}) \right]. \quad (168)$$

To do so, we have to write down a set of saddle-point equations and solve them.

**Saddle-point equations —** The saddle-point equations are derived straightforwardly from the obtained free energy extremising with respect to all parameters. A first set of equations is obtained from $\Psi_{\mathrm{out}}$ as[1]

$$\hat{Q}_k = \alpha \rho_k \mathbb{E}_{\boldsymbol\xi} \left[ f_k f_k^\top \right], \quad (169a)$$

$$\hat{V}_k = -\alpha \rho_k Q_k^{-1/2} \mathbb{E}_{\boldsymbol\xi} \left[ f_k \boldsymbol\xi^\top \right], \quad (169b)$$

$$\hat{m}_k = \alpha \rho_k \mathbb{E}_{\boldsymbol\xi} \left[ f_k \right], \quad (169c)$$

$$b = \sum_k \rho_k \mathbb{E}_{\boldsymbol\xi} \left[ h_k - m_k \right] \iff \sum_k \rho_k \mathbb{E}_{\boldsymbol\xi} \left[ V_k f_k \right] = 0. \quad (169d)$$

where for brevity we have denoted

$$f_k \equiv V_k^{-1}(h_k - \omega_k). \quad (170)$$

Similarly, the saddle-point equations from $\Psi_{\mathrm{out}}$ are

$$V_k = \frac{1}{d} \mathbb{E}_{\boldsymbol\Xi} \left[ \left( G \odot \left( \hat{Q}_k \otimes \Sigma_k \right) \right)^{-1/2} \odot \left( I_k \otimes \Sigma_k \right) \right) \Xi_k^\top \right] \quad (171a)$$

$$Q_k = \frac{1}{d} \mathbb{E}_{\boldsymbol\xi} \left[ G \Sigma_k G^\top \right] \quad (171b)$$

$$m_k = \frac{1}{\sqrt{d}} \mathbb{E}_{\boldsymbol\xi} \left[ G \mu_k \right]. \quad (171c)$$

To obtain the replica symmetric free energy, therefore, the given set of equation has to be solved, and the result then plugged in Eq. (168). No further simplification can be obtained in the most general setting. We will explore however some simple (but important) applications in Appendix C. Before going on, however, it is important to express the relevant quantities for learning, i.e., the training and generalization errors, in terms of the obtained order parameters.

## B.4 Training and test errors

The order parameters introduced to solve the problem allow us to reach our ultimate goal of computing the average errors of the learning process. We will start from the estimation of the training loss. The complication in computing this quantity is that the order parameters found in the learning process are, of course, correlated with the dataset used for the learning itself. We need to compute

$$\epsilon_\ell \equiv \frac{1}{n} \sum_{\nu=1}^{n} \ell\left( y^\nu, \frac{W^\star x^\nu}{\sqrt{d}} + b^\star \right) \quad (172)$$

in the $n \to +\infty$ limit. Denoting for brevity $\ell_k(x) \equiv \ell(e_k, x)$, the best way to proceed is to observe that $\mathbb{E}_{\{(y^\nu, x^\nu)\}_\nu}[\mathcal{R}(W^\star, b^\star)] = -\lim_{\beta \to +\infty} \mathbb{E}_{\{(y^\nu, x^\nu)\}_\nu}[\partial_\beta \ln \mathcal{Z}_\beta] = \lambda \mathbb{E}_{\{(y^\nu, x^\nu)\}_\nu}[r(W^\star)] + \epsilon_\ell$, where

$$\epsilon_\ell = -\lim_{\beta \to +\infty} \partial_\beta(\beta \Psi_{\mathrm{out}}) = \lim_{\beta \to +\infty} \sum_k \rho_k \int \ell_k(\boldsymbol\eta) \frac{e^{-\frac{\beta}{2}(\boldsymbol\eta - m_k^\star)^\top V_k^{\star -1}(\boldsymbol\eta - m_k^\star) - \beta \ell_k(\boldsymbol\eta)}}{\sqrt{\det(2\pi\beta^{-1} V^\star)} Z(e_k, \omega_k^\star, \beta^{-1} V_k^\star)} \mathrm{d}\boldsymbol\eta. \quad (173)$$

---

[1]To obtain the equation for $\hat{V}$ it is convenient to use Stein's lemma, so that $\mathbb{E}[\partial_\xi f_k] = \mathbb{E}[f_k \boldsymbol\xi^\top]$.

In the $\beta \to +\infty$ limit, the integral concentrates on the minimizer of the exponent, that is, by definition, the proximal $\boldsymbol{h}_k$. In conclusion, $\epsilon_\ell = \sum_k \rho_k \mathbb{E}[\ell(\boldsymbol{h}_k)]$. By means of the same concentration result, the training error is

$$\epsilon_t = \frac{1}{n} \sum_{\nu=1}^{n} \mathbb{I}\left( \boldsymbol{\varphi}\left( \frac{\boldsymbol{W}^\star \boldsymbol{x}^\nu}{\sqrt{d}} + \boldsymbol{b}^\star \right) \neq \boldsymbol{y}^\nu \right) \xrightarrow{n \to +\infty} \sum_{k=1}^{K} \rho_k \mathbb{E}_{\boldsymbol{\xi}} \left[ \mathbb{I}(\boldsymbol{\varphi}(\boldsymbol{h}_k) \neq \boldsymbol{e}_k) \right]. \tag{174}$$

The expressions above hold in general, but, as anticipated, important simplifications can occur in the set of saddle-point equations (169) and (171) depending on the choice of the loss $\ell$ and of the regularization function $r$.

The generalisation (or test) error can be written instead as

$$\epsilon_g = \mathbb{E}_{\boldsymbol{y}^{\mathrm{new}}, \boldsymbol{x}^{\mathrm{new}}} \left[ \mathbb{I}\left( \boldsymbol{\varphi}\left( \frac{\boldsymbol{W}^\star \boldsymbol{x}^{\mathrm{new}}}{\sqrt{d}} + \boldsymbol{b}^\star \right) \neq \boldsymbol{y}^{\mathrm{new}} \right) \right]. \tag{175}$$

This expression can be rewritten as

$$\epsilon_g = \sum_k \rho_k \int \mathbb{I}(\boldsymbol{\varphi}(\boldsymbol{\eta}) = \boldsymbol{e}_k) \mathbb{E}_{\boldsymbol{x}^{\mathrm{new}}} \left[ \delta\left( \boldsymbol{\eta} - \frac{\boldsymbol{W}^\star \boldsymbol{x}^{\mathrm{new}}}{\sqrt{d}} - \boldsymbol{b}^\star \right) \right] \mathrm{d}\boldsymbol{\eta} \tag{176}$$

Once again, we write

$$\mathbb{E}_{\boldsymbol{x}^{\mathrm{new}}} \left[ \delta\left( \boldsymbol{\eta} - \frac{\boldsymbol{W}^\star \boldsymbol{x}^{\mathrm{new}}}{\sqrt{d}} - \boldsymbol{b}^\star \right) \right] \xrightarrow{d \to +\infty} \mathcal{N}(\boldsymbol{\eta} | \boldsymbol{m}_k^\star + \boldsymbol{b}^\star, \boldsymbol{Q}_k^\star) \tag{177}$$

so that

$$\epsilon_g = \sum_{k=1}^{K} \rho_k \mathbb{E}_{\boldsymbol{\xi}} \left[ \mathbb{I}\left( \boldsymbol{\varphi}\left( \boldsymbol{m}_k^\star + \boldsymbol{Q}_k^{\star 1/2} \boldsymbol{\xi} + \boldsymbol{b}^\star \right) \neq \boldsymbol{e}_k \right) \right]. \tag{178}$$

This can be easily computed numerically once that the order parameters are given.

### B.5 A note on the numerical integration of the saddle-point equations

To estimate $\epsilon_g$, $\epsilon_t$ and $\epsilon_\ell$ we first need to find the fixed-point solutions of the saddle-point equations (169) and (171). The simplest numerical strategy consists in updating, in a self-consistent way, the order parameters until their variation according to, e.g., the Frobenius norm is smaller than a given threshold value (that we adopted to be $10^{-5}$). In the simplest setting, i.e., the one discussed in Corollary 3, the update of $(\boldsymbol{m}_k, \boldsymbol{Q}_k, \boldsymbol{V}_k)_{k \in [K]}$ is performed explicitly using eq. (11), where $\mathbb{E}_{\boldsymbol{\sigma}, \boldsymbol{\mu}}[\bullet]$ is a shorthand for the sum over the eigenvalues and eigenvectors of the assigned covariance matrices. The update of $(\hat{\boldsymbol{m}}_k, \hat{\boldsymbol{Q}}_k, \hat{\boldsymbol{V}}_k)_{k \in [K]}$ (right hand side of eq. (8)) is more involved, as it requires the computation of the proximal followed by a Gaussian average. Such average has been performed using a Monte Carlo strategy, i.e., by solving the equation for the proximal for a large number ($10^4 - 10^5$) of instances of $\boldsymbol{\xi}$ and averaging the solution. We remark that in the case of the square loss, the proximal can be computed analytically and the integration can be performed explicitly, highly simplifying the fixed-point equations (see below eq. (191)). We have found that in practice fluctuations due to the adopted Monte Carlo pool were small enough to be negligible compared with the outcomes of direct numerical experiments.

The convergence to the the correct fixed point is guaranteed (in principle) by the convexity of the problem. However, a few delicate aspects have to be taken into account in the update process described above.

1. The update requires the computation of the proximals $\boldsymbol{G}$ and $\boldsymbol{h}_k$. Such computations can be performed analytically in some specific cases only (for example, in the case of ridge regression). The existence of a unique solution is guaranteed by the strong convexity of the problem defining the proximal. In our study of the cross-entropy loss function, for example, we computed the proximals $\boldsymbol{h}_k$ numerically solving Eq. (194). In this problem, however, additional numerical instabilities emerged in the $\lambda \to 0$ limit, due the fact that the discontinuity in the gradient appear, see Eq. (198). We solved this issue performing an annealing in $\lambda$, i.e., solving for the proximal for decreasing values of the regularization strength.

2. The numerical solution of the saddle-point equations might suffer numerical instabilities due to the operations of inversion involved, see, e.g., the equation for $\hat{V}_k$ in (169), which requires the inversion of $Q_k$. It is convenient, in such cases, to rewrite the equation in an equivalent form which is numerically more stable. For example, in the aforementioned equation, we can observe that $f_k$ satisfies the equation $f_k + \partial_x \ell_k(V_k f_k + \omega_k) = 0$ so that $\partial_{\omega_k} f_k = -(I_K + \partial_x^2 \ell_k(V_k f_k + \omega_k)V_k)^{-1}\partial_x^2 \ell_k(V_k f_k + \omega_k)$. Using Stein's lemma,

$$\hat{V}_k = -\alpha\rho_k \mathbb{E}_{\xi}\left[\partial_{\xi} f_k\right] = \alpha\rho_k \mathbb{E}_{\xi}\left[\left(I_K + \partial_x^2 \ell_k(V_k f_k + \omega_k)V_k\right)^{-1}\partial_x^2 \ell_k(V_k f_k + \omega_k)\right].$$

(179)

We found this equation numerically more stable than the one given in (169) when dealing with the cross-entropy loss.

Our implementation can be found at [58].

# C  Some relevant particular cases

In this Appendix, we will specify the saddle-point equations for the multiclass classification problem for different choices of the loss function $\ell$ and of the regularisation function $r$. From the analysis developed in the previous Appendices, it is clear that the choices of $\ell$ and $r$ impact separately the set of equations (169) and (171) respectively. Once the order parameters are found, it is possible to estimate the training and generalisation errors as, for example, in Section B.4.

## C.1  The case of $\ell_2$ regularization

In this Section we consider the relevant case of quadratic regularization, $r(\boldsymbol{W}) = 1/2\|\boldsymbol{W}\|_{\mathrm{F}}^2$. In this case the computation of $\Psi_w$ can be performed explicitly via a Gaussian integration,

$$\frac{1}{\beta}\Psi_w(\hat{\boldsymbol{m}}, \hat{\boldsymbol{Q}}, \hat{\boldsymbol{V}}) = \frac{1}{2d}\operatorname{tr}\ln\mathbf{S} - \frac{K\ln\beta}{2\beta} + \frac{1}{2}\operatorname{tr}\left[\mathbf{S}\odot\left(\sum_{kk'}\hat{\boldsymbol{m}}_k\hat{\boldsymbol{m}}_{k'}^\top\otimes\boldsymbol{\mu}_k\boldsymbol{\mu}_{k'}^\top + \frac{1}{d}\sum_k\hat{\boldsymbol{Q}}_k\otimes\boldsymbol{\Sigma}_k\right)\right]. \tag{180}$$

Here we have introduced, for notation compactness,

$$\mathbf{S} \equiv \left(\lambda\mathbf{I}_K\otimes\mathbf{I}_d + \sum_\kappa\hat{\boldsymbol{V}}_\kappa\otimes\boldsymbol{\Sigma}_\kappa\right)^{-1}. \tag{181}$$

This form of $\Psi_w$ allows us to write in a simpler way the set of eqs. (171), that can be re-written as

$$\begin{aligned}
\boldsymbol{Q}_k &= \operatorname{tr}_d\left[(\mathbf{I}_K\otimes\boldsymbol{\Sigma}_k)\odot\mathbf{S}\odot\left(\sum_{kk'}\hat{\boldsymbol{m}}_k\hat{\boldsymbol{m}}_{k'}^\top\otimes\boldsymbol{\mu}_k\boldsymbol{\mu}_{k'}^\top + \frac{1}{d}\sum_\kappa\hat{\boldsymbol{Q}}_\kappa\otimes\boldsymbol{\Sigma}_\kappa\right)\odot\mathbf{S}\right] \\
\boldsymbol{m}_k &= \sum_{k'}\operatorname{tr}_d\left[\mathbf{S}\odot\left(\hat{\boldsymbol{m}}_{k'}\otimes\boldsymbol{\mu}_{k'}\boldsymbol{\mu}_k^\top\right)\right] \\
\boldsymbol{V}_k &= \frac{1}{d}\operatorname{tr}_d\left[(\mathbf{I}_K\otimes\boldsymbol{\Sigma}_k)\odot\mathbf{S}\right].
\end{aligned} \tag{182}$$

In the previous equations, by $\operatorname{tr}_d$ we denoted the trace with respect to the components living in the $d$-dimensional space of the dataset.

**Jointly diagonal covariances —**  Suppose now that $\boldsymbol{\Sigma}_k = \sum_i\sigma_i^k\boldsymbol{v}_i\boldsymbol{v}_i^\top$ for all $k$, i.e., the covariance matrices share the same basis of eigenvectors $\{\boldsymbol{v}_i\}_i$. Then, denoting $\mu_i^k \equiv \sqrt{d}\boldsymbol{\mu}_k^\top\boldsymbol{v}_i$

$$\begin{aligned}
\boldsymbol{Q}_k &= \frac{1}{d}\sum_{i=1}^d\sigma_i^k\left(\lambda\mathbf{I}_K + \sum_\kappa\sigma_i^\kappa\hat{\boldsymbol{V}}_\kappa\right)^{-1}\left(\sum_{kk'}\mu_i^k\mu_i^{k'}\hat{\boldsymbol{m}}_k\hat{\boldsymbol{m}}_{k'}^\top + \sum_\kappa\sigma_i^\kappa\hat{\boldsymbol{Q}}_\kappa\right)\left(\lambda\mathbf{I}_K + \sum_\kappa\sigma_i^\kappa\hat{\boldsymbol{V}}_\kappa\right)^{-1} \\
\boldsymbol{m}_k &= \frac{1}{d}\sum_{i=1}^d\sum_{k'}\mu_i^k\mu_i^{k'}\left(\lambda\mathbf{I}_K + \sum_\kappa\sigma_i^\kappa\hat{\boldsymbol{V}}_\kappa\right)^{-1}\hat{\boldsymbol{m}}_{k'} \\
\boldsymbol{V}_k &= \frac{1}{d}\sum_{i=1}^d\sigma_i^k\left(\lambda\mathbf{I}_K + \sum_\kappa\sigma_i^\kappa\hat{\boldsymbol{V}}_\kappa\right)^{-1}.
\end{aligned} \tag{183}$$

Introducing the joint density

$$\frac{1}{d}\sum_{i=1}^d\prod_{\kappa=1}^K\delta(\sigma^\kappa - \sigma_i^\kappa)\delta(\mu^\kappa - \mu_i^\kappa) \xrightarrow{d\to+\infty} \rho(\boldsymbol{\sigma}, \boldsymbol{\mu}), \tag{184}$$

then we can write the saddle-point equations given in Corollary 3

$$\boldsymbol{Q}_k = \mathbb{E}_{\boldsymbol{\sigma},\boldsymbol{\mu}} \left[ \sigma^k \left( \lambda \mathbf{I}_K + \sum_\kappa \sigma^\kappa \hat{\boldsymbol{V}}_\kappa \right)^{-1} \left( \sum_{kk'} \mu^k \mu^{k'} \hat{\boldsymbol{m}}_k \hat{\boldsymbol{m}}_{k'}^\top + \sum_\kappa \sigma^\kappa \hat{\boldsymbol{Q}}_\kappa \right) \left( \lambda \mathbf{I}_K + \sum_\kappa \sigma^\kappa \hat{\boldsymbol{V}}_\kappa \right)^{-1} \right]$$

$$\boldsymbol{m}_k = \mathbb{E}_{\boldsymbol{\sigma},\boldsymbol{\mu}} \left[ \mu^k \left( \lambda \mathbf{I}_K + \sum_\kappa \sigma^\kappa \hat{\boldsymbol{V}}_\kappa \right)^{-1} \sum_\kappa \mu^\kappa \hat{\boldsymbol{m}}_\kappa \right]$$

$$\boldsymbol{V}_k = \mathbb{E}_{\boldsymbol{\sigma},\boldsymbol{\mu}} \left[ \sigma^k \left( \lambda \mathbf{I}_K + \sum_\kappa \sigma^\kappa \hat{\boldsymbol{V}}_\kappa \right)^{-1} \right].$$

(185)

where the expectations $\mathbb{E}_{\boldsymbol{\sigma},\boldsymbol{\mu}}$ are taken with respect to the joint distribution $\rho$.

### C.1.1 Uniform covariances

Let us consider the simpler case $\boldsymbol{\Sigma}_k \equiv \Delta \mathbf{I}_d$, with $\Delta > 0$. In this case, the saddle-point equations can take a more compact form that is particularly suitable for a numerical solution. Moreover, for reasons of symmetry we can write

$$\boldsymbol{Q}_k \equiv \boldsymbol{Q}, \quad \boldsymbol{V}_k \equiv \boldsymbol{V}, \quad \hat{\boldsymbol{Q}}_k \equiv \frac{1}{K\Delta} \hat{\boldsymbol{Q}}_k, \quad \hat{\boldsymbol{V}}_k \equiv \frac{1}{K\Delta} \hat{\boldsymbol{V}}, \quad \forall k.$$

(186)

Let us define the following $K \times K$ matrices

- $\mathbf{M} \in \mathbb{R}^{K \times K}$ (resp. $\hat{\mathbf{M}} \in \mathbb{R}^{K \times K}$) is the matrix obtained concatenenating the vectors $\boldsymbol{m}_k$ (resp. $\hat{\boldsymbol{m}}_k$);
- $\boldsymbol{\Theta} = \left( \boldsymbol{\mu}_k^\top \boldsymbol{\mu}_{k'} \right)_{kk'}$ is the Gram matrix of the means;
- $\boldsymbol{F} \in \mathbb{R}^{K \times K}$ is the matrix obtained concatenenating the vectors $\boldsymbol{f}_k$;
- $\boldsymbol{H} \in \mathbb{R}^{K \times K}$ is the matrix obtained concatenenating the vectors $\boldsymbol{h}_k$;
- $\boldsymbol{\Pi} = \mathrm{diag}(\rho_k) \in \mathbb{R}^{K \times K}$ is a diagonal matrix with elements $\Pi_{kk'} = \delta_{kk'}\rho_k$.

The saddle-point equations then can be rewritten as

$$\boldsymbol{Q} = \Delta \left( \lambda \mathbf{I}_K + \hat{\boldsymbol{V}} \right)^{-1} \left( \hat{\boldsymbol{Q}} + \hat{\mathbf{M}} \boldsymbol{\Theta} \hat{\mathbf{M}}^\top \right) \left( \lambda \mathbf{I}_K + \hat{\boldsymbol{V}} \right)^{-1} \qquad \hat{\boldsymbol{Q}} = \alpha \Delta \mathbb{E}_{\boldsymbol{\Xi}} \left[ \boldsymbol{F} \boldsymbol{\Pi} \boldsymbol{F}^\top \right]$$

$$\mathbf{M} = \left( \lambda \mathbf{I}_K + \hat{\boldsymbol{V}} \right)^{-1} \hat{\mathbf{M}} \boldsymbol{\Theta} \qquad\qquad\qquad\qquad \hat{\boldsymbol{V}} = -\alpha \Delta \boldsymbol{Q}^{-1/2} \mathbb{E}_{\boldsymbol{\Xi}} \left[ \boldsymbol{F} \boldsymbol{\Pi} \boldsymbol{\Xi}^\top \right]$$

$$\hat{\mathbf{M}} = \alpha \mathbb{E}_{\boldsymbol{\Xi}} \left[ \boldsymbol{F} \boldsymbol{\Pi} \right]$$

$$\boldsymbol{V} = \Delta \left( \lambda \mathbf{I}_K + \hat{\boldsymbol{V}} \right)^{-1}, \qquad\qquad\qquad\qquad \boldsymbol{b} = \mathbb{E}_{\boldsymbol{\Xi}} [(\boldsymbol{H} - \mathbf{M}) \boldsymbol{\Pi} \mathbf{1}_K].$$

(187)

Here and in the following $\mathbf{1}_K$ is the vector of $K$ components all equal to $1$. These expressions are particularly suitable for a numerical implementation, because involve matrix multiplications and inversions of $K$-dimensional objects only.

**Quadratic loss** —  If we consider a quadratic loss $\ell(\boldsymbol{y},\boldsymbol{x}) = \frac{1}{2}(\boldsymbol{y}-\boldsymbol{x})^2$, then an explicit formula for the proximal can be found, namely

$$\boldsymbol{f}_k = (\mathbf{I}_K + \boldsymbol{V})^{-1}(\boldsymbol{e}_K - \boldsymbol{\omega}_k)$$

(188)

so that the second set of saddle-point equations (187) can be written as

$$\hat{\boldsymbol{Q}} = \alpha (\mathbf{I}_K + \boldsymbol{V})^{-1} \left[ (\mathbf{I}_K - \mathbf{M} - \boldsymbol{b} \otimes \mathbf{1}_K) \boldsymbol{\Pi} (\mathbf{I}_K - \mathbf{M} - \boldsymbol{b} \otimes \mathbf{1}_K)^\top + \boldsymbol{Q} \right] (\mathbf{I}_K + \boldsymbol{V})^{-1}$$

$$\hat{\mathbf{M}} = \alpha (\mathbf{I}_K + \boldsymbol{V})^{-1} (\mathbf{I}_K - \mathbf{M} - \boldsymbol{b} \otimes \mathbf{1}_K) \boldsymbol{\Pi}$$

(189)

$$\hat{\boldsymbol{V}} = \alpha \Delta (\mathbf{I}_K + \boldsymbol{V})^{-1}.$$

Observe at this point that we can explicitly solve for $\boldsymbol{V}$ using the equation for it in eqs. (187). In particular, $\boldsymbol{V}$ satisfies the equation $\lambda \boldsymbol{V}^2 + (\alpha + \lambda - \Delta)\boldsymbol{V} = \Delta \mathbf{I}_K$. Being $\boldsymbol{V}$ positive definite, it follows that it is diagonal, $\boldsymbol{V} = V \mathbf{I}_K$ with diagonal element

$$V = \frac{\Delta(1-\alpha) - \lambda + \sqrt{(\Delta - \alpha\Delta - \lambda)^2 + 4\Delta\lambda}}{2\lambda}, \quad \hat{V} = \frac{\alpha\Delta}{1+V},$$

(190)

so that

$$Q=\frac{\Delta}{(\lambda+\Delta\hat{V})^2}\left(\hat{Q}+\hat{\mathbf{M}}\Theta\hat{\mathbf{M}}^\top\right)$$

$$\mathbf{M}=\frac{\hat{\mathbf{M}}\Theta}{\lambda+\Delta\hat{V}},$$

$$\boldsymbol{b}=(\mathbf{I}_K-\mathbf{M})\Pi\mathbf{1}_K,$$

$$\hat{Q}=\frac{\alpha\left[(\mathbf{I}_K-\mathbf{M}-\boldsymbol{b}\otimes\mathbf{1}_K)\Pi(\mathbf{I}_K-\mathbf{M}-\boldsymbol{b}\otimes\mathbf{1}_K)^\top+Q\right]}{(1+V)^2}$$

$$\hat{\mathbf{M}}=-\frac{\alpha(\mathbf{I}_K-\mathbf{M}-\boldsymbol{b}\otimes\mathbf{1}_K)\Pi}{1+V}. \tag{191}$$

In the $\lambda\to 0$ limit, for $\alpha<1$ it is convenient to rescale $\hat{Q}\mapsto\lambda^2\hat{Q}$ and $\hat{\mathbf{M}}\mapsto\lambda\hat{\mathbf{M}}$, so that

$$Q=\Delta(1-\alpha)^2\left(\hat{Q}+\hat{\mathbf{M}}\Theta\hat{\mathbf{M}}^\top\right),$$

$$\mathbf{M}=(1-\alpha)\hat{\mathbf{M}}\Theta,$$

$$\boldsymbol{b}=(\mathbf{I}_K-\mathbf{M})\Pi\mathbf{1}_K,$$

$$\hat{Q}=\frac{\alpha\left[(\mathbf{I}_K-\mathbf{M}-\boldsymbol{b}\otimes\mathbf{1}_K)\Pi(\mathbf{I}_K-\mathbf{M}-\boldsymbol{b}\otimes\mathbf{1}_K)^\top+Q\right]}{\Delta^2(1-\alpha)^2},$$

$$\hat{\mathbf{M}}=-\frac{\alpha(\mathbf{I}_K-\mathbf{M}-\boldsymbol{b}\otimes\mathbf{1}_K)\Pi}{\Delta(1-\alpha)}. \tag{192}$$

**Cross-entropy loss —** We consider now the relevant case of the cross entropy loss

$$\ell(\boldsymbol{y},\boldsymbol{x}) = -\sum_{k=1}^K y_k \ln \frac{e^{x_k}}{\sum_{\kappa=1}^K e^{x_\kappa}}. \tag{193}$$

If $\boldsymbol{y}\in\{\boldsymbol{e}_k\}_{k\in[K]}$, the loss can be written in the form $\ell(\boldsymbol{y},\boldsymbol{x})=-\boldsymbol{y}^\top\boldsymbol{x}+\ln\sum_\kappa e^{x_\kappa}$. If we introduce the *softmax function* $\mathbf{soft}\colon\mathbb{R}^K\to\mathbb{R}^K$

$$\partial_{\boldsymbol{x}}\ell(\boldsymbol{y},\boldsymbol{x}) = -\boldsymbol{y}+\mathbf{soft}(\boldsymbol{x}), \qquad \mathrm{soft}_k(\boldsymbol{x})\equiv\frac{\exp{(x_k)}}{\sum_\kappa\exp{(x_\kappa)}} \tag{194}$$

the proximal equation for the cross-entropy loss is the solution of the equations:

$$\boldsymbol{V}^{-1}(\boldsymbol{h}_k-\boldsymbol{\omega}_k)-\boldsymbol{e}_k+\mathbf{soft}(\boldsymbol{h}_k)=\mathbf{0}\iff\boldsymbol{f}_k=\boldsymbol{e}_k-\mathbf{soft}(\boldsymbol{V}\boldsymbol{f}_k+\boldsymbol{\omega}_k)\quad\forall k\in[K], \tag{195}$$

having only one solution for which, however, there is no closed-form expression. The equation can be solved numerically, and in this way we obtained the results in Section 3.2.

The saddle-point equations can be written rescaling $\boldsymbol{Q}\mapsto\lambda^{-2}\boldsymbol{Q}$, $\boldsymbol{V}\mapsto\lambda^{-1}\boldsymbol{V}$, $\mathbf{M}\mapsto\lambda^{-1}\mathbf{M}$, $\boldsymbol{b}\mapsto\lambda^{-1}\boldsymbol{b}$, $\hat{V}\mapsto\lambda\hat{V}$. They become

$$\boldsymbol{Q}=\Delta\left(\mathbf{I}_K+\hat{\boldsymbol{V}}\right)^{-1}\left(\hat{\boldsymbol{Q}}+\hat{\mathbf{M}}\Theta\hat{\mathbf{M}}^\top\right)\left(\mathbf{I}_K+\hat{\boldsymbol{V}}\right)^{-1},$$

$$\mathbf{M}=\left(\mathbf{I}_K+\hat{\boldsymbol{V}}\right)^{-1}\hat{\mathbf{M}}\Theta$$

$$\boldsymbol{V}=\Delta\left(\mathbf{I}_K+\hat{\boldsymbol{V}}\right)^{-1},$$

$$\hat{\boldsymbol{Q}}=\alpha\Delta\mathbb{E}_{\boldsymbol{\Xi}}\left[\boldsymbol{F}\Pi\boldsymbol{F}^\top\right],$$

$$\hat{\boldsymbol{V}}=-\alpha\Delta\boldsymbol{Q}^{-1/2}\mathbb{E}_{\boldsymbol{\Xi}}\left[\boldsymbol{F}\Pi\boldsymbol{\Xi}^\top\right],$$

$$\hat{\mathbf{M}}=\alpha\mathbb{E}_{\boldsymbol{\Xi}}\left[\boldsymbol{F}\Pi\right],$$

$$\boldsymbol{b}=\mathbb{E}_{\boldsymbol{\Xi}}\left[(\boldsymbol{H}-\mathbf{M})\Pi\right], \tag{196}$$

so that the dependence on $\lambda$ disappears everywhere except in the equation for the proximal $\boldsymbol{f}_k$

$$\boldsymbol{f}_k=\arg\min_{\boldsymbol{x}}\left[\frac{1}{2}\boldsymbol{x}^\top\boldsymbol{V}\boldsymbol{x}+\lambda\ell\left(\boldsymbol{e}_k,\frac{\boldsymbol{V}\boldsymbol{x}+\boldsymbol{\omega}_k}{\lambda}\right)\right], \tag{197}$$

which, in the $\lambda\to 0$ limit, becomes

$$\boldsymbol{f}_k=\arg\min_{\boldsymbol{x}}\left[\frac{1}{2}\boldsymbol{x}^\top\boldsymbol{V}\boldsymbol{x}+\min_\mu\{(\boldsymbol{e}_\mu-\boldsymbol{e}_k)^\top(\boldsymbol{V}\boldsymbol{x}+\boldsymbol{\omega}_k)\}\right]. \tag{198}$$

Note that in this limit, minimising the cross-entropy loss yields precisely the max-margin estimator [70].

## C.2 The $K=2$ case with scalar labels

The formulas for the $K=2$ case can be derived directly from the general analysis given above imposing $L=1$. In particular, let us assume that the two clusters are labeled with $e_1=+1$ and $e_2=-1$. Using as classifier

$$\varphi(x)=\mathrm{sign}(x) \tag{199}$$

the expression of the average errors is

$$\epsilon_g = \sum_{k\in[2]} \rho_k \mathbb{E}_\xi[\theta\left((-1)^k \omega_k^\star\right)] = \sum_{k\in[2]} \frac{\rho_k}{2}\mathrm{erfc}\left((-1)^{k-1}\frac{m_k^\star + b^\star}{\sqrt{2q_k^\star}}\right),$$

$$\epsilon_t = \sum_{k\in[2]} \rho_k \mathbb{E}_\xi[\theta\left((-1)^k h_k^\star\right)], \tag{200}$$

$$\epsilon_\ell = \sum_{k\in[2]} \rho_k \mathbb{E}_\xi[\ell((-1)^k, h_k^\star)].$$

We will further explore this case, considering some special cases in the following.

### C.2.1   Example: $\ell_1$ regularization

In this Section we derive the saddle-point equations for the the case in which the two cluster have opposite means $\boldsymbol{\mu}_1 = -\boldsymbol{\mu}_2 \equiv \boldsymbol{\mu}$, and the same diagonal covariance matrix, $\boldsymbol{\Sigma}_1 = \boldsymbol{\Sigma}_2 \equiv \boldsymbol{\Sigma}$, with $\Sigma_{ij} = \sigma_i \delta_{ij}$ and $\sigma_i > 0$. In this case, for symmetry reasons, the overlaps simplify and we have:

$$V_1 = V_2 \equiv V, \qquad\qquad q_1 = q_2 \equiv q, \qquad\qquad m_+ = -m_- \equiv m, \qquad (201)$$

$$\hat{V}_+ = \hat{V}_- \equiv \frac{1}{2}\hat{V}, \qquad\qquad \hat{q}_+ = \hat{q}_- \equiv \frac{1}{2}\hat{q}, \qquad\qquad \hat{m}_+ = -\hat{m}_- \equiv \frac{1}{2}\hat{m}. \qquad (202)$$

We define

$$\frac{1}{d}\sum_{i=1}^d \delta(\sigma - \sigma_i)\delta(\mu - \sqrt{d}\mu_i) \xrightarrow{d\to+\infty} p(\sigma, \mu) \tag{203}$$

joint distribution of the covariance diagonal elements and of the mean elements. We will denote $\mathbb{E}_{\mu,\sigma}[\bullet]$ the average with respect to this measure. We will focus in particular on the form of the saddle-point equations obtained from the prior contribution assuming $\ell_1$ regularization, i.e., $r(\boldsymbol{w}) = \sum_i |w_i|$, and let us introduce the corresponding *soft-thresholding operator*:

$$\mathrm{Prox}_{\lambda|\cdot|}(x) = \mathrm{sign}(x)\max\{|x| - \lambda, 0\}. \tag{204}$$

Observe that $\mathrm{Prox}_{\alpha\lambda|\cdot|}(\alpha x) = \alpha \mathrm{Prox}_{\lambda|\cdot|}(x)$ for $\alpha > 0$. Its derivative given by $\mathrm{Prox}'_{\lambda|\cdot|}(x) = \theta(|x| > \lambda)$. The saddle point equations from the prior part simply read:

$$V = \frac{1}{\hat{V}} \mathbb{E}_{\mu,\sigma,\xi}\left[\mathrm{Prox}'_{\frac{\lambda}{\sigma\hat{V}}|\cdot|}\left(\frac{\hat{m}\mu + \sqrt{\hat{q}}\sigma\xi}{\hat{V}\sigma}\right)\right], \tag{205}$$

$$q = \mathbb{E}_{\mu,\sigma,\xi}\left[\sigma\left(\mathrm{Prox}_{\frac{\lambda}{\sigma\hat{V}}|\cdot|}\left(\frac{\hat{m}\mu + \sqrt{\hat{q}}\sigma\xi}{\hat{V}\sigma}\right)\right)^2\right], \tag{206}$$

$$m = \mathbb{E}_{\mu,\sigma,\xi}\left[\mu\mathrm{Prox}_{\frac{\lambda}{\sigma\hat{V}}|\cdot|}\left(\frac{\hat{m}\mu + \sqrt{\hat{q}}\sigma\xi}{\hat{V}\sigma}\right)\right]. \tag{207}$$

The averages over $\xi$ can be performed explicitely using the simple expression of the proximal in this case. If we define the auxiliary functions

$$\phi_\pm^0(v, u, \lambda) \equiv \frac{1}{2}\mathrm{erfc}\left(\frac{\lambda \pm v}{\sqrt{2u}}\right)$$

$$\phi_\pm^1(u, v, \lambda) = \sqrt{\frac{u}{2\pi}}e^{-\frac{(v\pm\lambda)^2}{2u}} - \frac{v\pm\lambda}{2}\mathrm{erfc}\left(\frac{\lambda\pm v}{\sqrt{2u}}\right), \tag{208}$$

$$\phi_\pm^2(v, u, \lambda) = -\sqrt{\frac{u}{2\pi}}e^{-\frac{(\lambda\pm v)^2}{2u}}(\lambda \pm v) + \frac{u + (\lambda\pm v)^2}{2}\mathrm{erfc}\left(\frac{\lambda\pm v}{\sqrt{2u}}\right).$$

then

$$V = \frac{1}{\hat{V}}\mathbb{E}_{\mu,\sigma}\left[\phi_+^0(\mu\hat{m}, \sigma\hat{q}, \lambda) + \phi_-^0(\mu\hat{m}, \sigma\hat{q}, \lambda)\right]$$

$$q = \mathbb{E}_{\mu,\sigma}\left[\frac{\phi_+^2(\mu\hat{m}, \sigma\hat{q}, \lambda) + \phi_-^2(\mu\hat{m}, \sigma\hat{q}, \lambda)}{\sigma\hat{V}^2}\right], \tag{209}$$

$$m = \mathbb{E}_{\mu,\sigma}\left[\frac{\mu\phi_-^1(\mu\hat{m}, \sigma q, \lambda) - \mu\phi_+^1(\mu\hat{m}, \sigma q, \lambda)}{\sigma\hat{V}}\right].$$

**Gaussian means, homogenous covariances —** If $p(\mu, \sigma) = \mathcal{N}(\mu|0, 1)\delta(\sigma - \Delta)$, i.e., the means have i.i.d. Gaussian entries and $\boldsymbol{\Sigma} = \Delta \mathbf{I}_d$, then

$$V = \frac{1}{\hat{V}} \mathbb{E}_z \left[ \operatorname{erfc}\left( \frac{\lambda + \hat{m}z}{\sqrt{2\Delta\hat{q}}} \right) \right],$$

$$q = \frac{1}{\Delta\hat{V}^2} \left\{ -\frac{e^{-\frac{1}{2}\frac{\lambda^2}{\hat{m}^2 + \Delta\hat{q}}}}{\sqrt{2\pi(\hat{m}^2 + \Delta\hat{q})}} \frac{2(\Delta\hat{q})^2\lambda}{\hat{m}^2 + \Delta\hat{q}} + \mathbb{E}_z \left[ (\lambda + \hat{m}z)^2 \operatorname{erfc}\left( \frac{\lambda + \hat{m}z}{\sqrt{2\Delta\hat{q}}} \right) \right] \right\}, \tag{210}$$

$$m = \frac{1}{\Delta\hat{V}} \left\{ \frac{e^{-\frac{1}{2}\frac{\lambda^2}{\hat{m}^2 + \Delta\hat{q}}}}{\sqrt{2\pi(\hat{m}^2 + \Delta\hat{q})}} \frac{2\Delta\hat{q}\hat{m}\lambda}{\hat{m}^2 + \Delta\hat{q}} + \mathbb{E}_{z \sim \mathcal{N}(0,1)} \left[ (\lambda + \hat{m}z)\, z \operatorname{erfc}\left( \frac{\lambda + \hat{m}z}{\sqrt{2\Delta\hat{q}}} \right) \right] \right\},$$

with $z \sim \mathcal{N}(0, 1)$.

**Covariance correlated with sparse means —** In Section 3.1 we considered the case of sparse means correlated with the covariance matrices. In particular, we considered

$$p(\sigma, \mu) = p\mathcal{N}(\mu|0, 1)\delta(\sigma - \Delta_1) + (1 - p)\delta(\mu)\delta(\sigma - \Delta_0). \tag{211}$$

The saddle-point equations are therefore

$$V = \frac{1}{\hat{V}} \left[ p\mathbb{E}_\mu \left[ \operatorname{erfc}\left( \frac{\lambda + \hat{m}\mu}{\sqrt{2\Delta_1\hat{q}}} \right) \right] + (1 - p)\operatorname{erfc}\left( \frac{\lambda}{\sqrt{2\Delta_0\hat{q}}} \right) \right] \tag{212}$$

$$q = \frac{p}{\Delta_1\hat{V}^2} \left\{ -\frac{e^{-\frac{1}{2}\frac{\lambda^2}{\hat{m}^2 + \Delta_1\hat{q}}}}{\sqrt{2\pi(\hat{m}^2 + \Delta_1\hat{q})}} \frac{2(\Delta_1\hat{q})^2\lambda}{\hat{m}^2 + \Delta_1\hat{q}} + \mathbb{E}_z \left[ (\lambda + \hat{m}z)^2 \operatorname{erfc}\left( \frac{\lambda + \hat{m}z}{\sqrt{2\Delta_1\hat{q}}} \right) \right] \right\}$$

$$- \lambda(1 - p)\sqrt{\frac{\Delta_0\hat{q}}{2\pi}} e^{-\frac{\lambda^2}{2\Delta_0 q}} + \frac{1 - p}{2}(\Delta_0\hat{q} + \lambda^2)\operatorname{erfc}\left( \frac{\lambda}{\sqrt{2\Delta_0\hat{q}}} \right) \tag{213}$$

$$m = \frac{p}{\Delta_1\hat{V}} \left\{ \frac{e^{-\frac{1}{2}\frac{\lambda^2}{\hat{m}^2 + \Delta_1\hat{q}}}}{\sqrt{2\pi(\hat{m}^2 + \Delta_1\hat{q})}} \frac{2\Delta_1\hat{q}\hat{m}\lambda}{\hat{m}^2 + \Delta_1\hat{q}} + \mathbb{E}_z \left[ (\lambda + \hat{m}z)\, z \operatorname{erfc}\left( \frac{\lambda + \hat{m}z}{\sqrt{2\Delta_1\hat{q}}} \right) \right] \right\}. \tag{214}$$

In Section 3.1 we compare the performance obtained adopting an $\ell_1$ regularization with the corresponding one obtained using $\ell_2$, $r(\boldsymbol{w}) = \sum_i w_i^2$. For the sake of completeness, we give here the expression of the saddle-point equations in that case as well. In this case, the prior term $\Psi_w$ can be written explicitly after a Gaussian integration as

$$\Psi_w(\hat{m}, \hat{Q}, \hat{V}) = -\frac{1}{2d} \operatorname{tr} \ln\left( \lambda \mathbf{I}_d + \hat{V}\boldsymbol{\Sigma} \right) + \frac{1}{2} \operatorname{tr}\left[ \left( \lambda \mathbf{I}_d + \hat{V}\boldsymbol{\Sigma} \right)^{-1} \left( \hat{m}_k^2 \boldsymbol{\mu}\boldsymbol{\mu}^\top + \frac{\hat{q}}{d}\boldsymbol{\Sigma} \right) \right]. \tag{215}$$

In the setting given by eq. (211) the saddle point equations are then

$$q = p\frac{\hat{m}^2\Delta_1 + \hat{q}\Delta_1^2}{(\lambda + \hat{V}\Delta_1)^2} + \frac{(1 - p)\hat{q}\Delta_0^2}{(\lambda + \hat{V}\Delta_0)^2} \tag{216a}$$

$$V = p\frac{\Delta_1}{\lambda + \hat{V}\Delta_1} + \frac{(1 - p)\Delta_0}{\lambda + \hat{V}\Delta_0} \tag{216b}$$

$$m = \frac{\hat{m}p}{\lambda + \hat{V}\Delta_1}. \tag{216c}$$

## D Bayes optimal error

In this Appendix, we derive a formula for the Bayes optimal classification error in the case of $K$ clusters with the same covariance $\boldsymbol{\Sigma}_k = \Delta \mathbf{I}_d$ in the large $d$ limit, assuming that a dataset $\{(\boldsymbol{x}^\nu, \boldsymbol{y}^\nu)\}_{\nu \in [n]}$ of correctly labeled points is available. As usual, we will assume $n/d = \alpha$ finite. The distribution of a pair $(\boldsymbol{y}, \boldsymbol{x})$ is given by

$$p(\boldsymbol{y}, \boldsymbol{x}|\boldsymbol{M}) = \sum_k y_k \frac{\rho_k \exp\left(-\frac{1}{2\Delta}\|\boldsymbol{x} - \boldsymbol{\mu}_k\|^2\right)}{(2\pi\Delta)^{\frac{d}{2}}}. \tag{217}$$

where $\boldsymbol{M} \in \mathbb{R}^{d \times K}$ is the matrix of concatenated means $\boldsymbol{\mu}_k$ *estimated* from the dataset, so that

$$p(\boldsymbol{M}|\{\boldsymbol{y}^\nu, \boldsymbol{x}^\nu\}_\nu) \propto p(\{\boldsymbol{x}^\nu\}_\nu|\boldsymbol{M}, \{\boldsymbol{y}^\nu\}_\nu) P_{\boldsymbol{\mu}}(\boldsymbol{M})$$

$$\propto P_{\boldsymbol{\mu}}(\boldsymbol{M}) \prod_{\nu=1}^n \sum_k y_k^\nu \exp\left(-\frac{1}{2\Delta}\|\boldsymbol{x}^\nu - \boldsymbol{\mu}_k\|^2\right). \tag{218}$$

We will assume in the following the distribution

$$P_{\boldsymbol{\mu}}(\boldsymbol{M}) = \frac{\exp\left(-\frac{d}{2}\mathrm{tr}[\boldsymbol{M}\boldsymbol{\Theta}^{-1}\boldsymbol{M}^\top]\right)}{(2\pi)^{\frac{Kd}{2}} d^{-K/2}|\boldsymbol{\Theta}|^{1/2}} \tag{219}$$

where $\boldsymbol{\Theta} \in \mathbb{R}^{K \times K}$ is a given positive definite covariance matrix. In this way

$$\mathbb{E}\left[\boldsymbol{M}^\top \boldsymbol{M}\right] = \boldsymbol{\Theta}. \tag{220}$$

The conditional distribution for the label $\boldsymbol{y}^0$ of a new point $\boldsymbol{x}^0$,

$$p(\boldsymbol{y}^0|\boldsymbol{x}^0, \{\boldsymbol{y}^\nu, \boldsymbol{x}^\nu\}_\nu) \propto \mathbb{E}_{\boldsymbol{M}|\{\boldsymbol{y}^\nu, \boldsymbol{x}^\nu\}_\nu}[p(\boldsymbol{y}, \boldsymbol{x}|\boldsymbol{M})]$$

$$= \int \mathrm{d}\boldsymbol{M} P_{\boldsymbol{\mu}}(\boldsymbol{M}) \sum_k y_k^0 \rho_k \exp\left(-\frac{\|\boldsymbol{x}^0 - \boldsymbol{\mu}_k\|^2}{2\Delta}\right) \prod_{\nu=1}^n \sum_k y_k^\nu \exp\left(-\frac{\|\boldsymbol{x}^\nu - \boldsymbol{\mu}_k\|^2}{2\Delta}\right). \tag{221}$$

If $\boldsymbol{n} = (n_k)_k$ is the vector of the number of examples $n_k$ in the class $k$, then

$$p(\boldsymbol{y}^0|\boldsymbol{x}^0, \{\boldsymbol{y}^\nu, \boldsymbol{x}^\nu\}_\nu) \propto \int \mathrm{d}\boldsymbol{M} P_{\boldsymbol{\mu}}(\boldsymbol{M}) \prod_{k=1}^K \left[\rho_k^{y_k^0} \exp\left(-\sum_{\nu=0}^n \frac{y_k^\nu \|\boldsymbol{x}^\nu - \boldsymbol{\mu}_k\|^2}{2\Delta}\right)\right]$$

$$= \exp\left[\sum_k y_k^0 \left(\ln\rho_k - \frac{\|\boldsymbol{x}\|^2}{2\Delta}\right) - \frac{1}{2}\ln\det\left(1 + \frac{1}{d\Delta}\mathrm{diag}(\boldsymbol{n} + \boldsymbol{y}^0)\boldsymbol{\Theta}\right)\right]$$

$$\times \exp\left[\frac{1}{2\Delta}\mathrm{tr}\left[\left(\sum_{\nu=0}^n \boldsymbol{y}^\nu \otimes \boldsymbol{x}^\nu\right)^\top (d\Delta\boldsymbol{\Theta}^{-1} + \mathrm{diag}(\boldsymbol{n} + \boldsymbol{y}))^{-1}\left(\sum_{\nu=0}^n \boldsymbol{y}^\nu \otimes \boldsymbol{x}^\nu\right)\right]\right]. \tag{222}$$

In the following we will denote by $\star$ the true label of $\boldsymbol{x}$. Let $\boldsymbol{\Pi} = \mathrm{diag}(\rho_k)$. Then we can write the previous expression as

$$p(\boldsymbol{y}^0|\boldsymbol{x}^0, \{\boldsymbol{y}^\nu, \boldsymbol{x}^\nu\}_\nu) \propto \exp\left[\sum_k y_k \left(\ln\rho_k - \frac{\|\boldsymbol{x}^0\|^2}{2\Delta}\right) - \frac{1}{2}\ln\det\left(1 + \frac{1}{\Delta}\alpha\boldsymbol{\Pi}\boldsymbol{\Theta}\right)\right]$$

$$\times \exp\left[\frac{1}{2\Delta}\mathrm{tr}\left[\left(\frac{1}{d}\sum_{\nu=0}^n \boldsymbol{y}^\nu \otimes \boldsymbol{x}^\nu\right)^\top (\Delta\boldsymbol{\Theta}^{-1} + \alpha\boldsymbol{\Pi})^{-1}\left(\sum_{\nu=0}^n \boldsymbol{y}^\nu \otimes \boldsymbol{x}^\nu\right)\right]\right] \tag{223}$$

Observe now that

$$\frac{1}{d\Delta}\boldsymbol{x}^0 \sum_{\nu=1}^n y_k^\nu \boldsymbol{x}^\nu \xrightarrow{n,d \to +\infty} \alpha\rho_k \frac{\Theta_{\star,k} + \eta_k Z_k}{\Delta}, \qquad \eta_k \equiv \sqrt{\Delta\left(1 + \frac{\Delta}{\alpha\rho_k}\right)}, \quad Z_k \sim \mathcal{N}(0, 1),$$

$$\tag{224}$$

so that, defining the vector $\boldsymbol{a}^\star = (a_k)_{k\in[K]}$ with elements

$$a_k^\star \equiv \alpha\rho_k \frac{\Theta_{\star,k} + \eta_k Z_k}{\Delta}, \tag{225}$$

and neglecting the $\boldsymbol{y}^0$-independent contributions, the expression above can be rewritten as

$$p(\boldsymbol{y}^0|\boldsymbol{x}^0, \{\boldsymbol{y}^\nu, \boldsymbol{x}^\nu\}_\nu) \propto \exp\left[\sum_k y_k^0 \ln\rho_k + \left(\boldsymbol{a}^\star + \frac{1}{2}\boldsymbol{y}^0\right)^\top \left(\Delta\boldsymbol{\Theta}^{-1} + \alpha\boldsymbol{\Pi}\right)^{-1}\boldsymbol{y}^0\right] \tag{226}$$

where we have also used the fact that $\|\boldsymbol{x}^0\|^2 = d\Delta + O(1)$. This means that the Bayes optimal generalization error is

$$\varepsilon_g^{\text{BO}} = \sum_k \rho_k \mathbb{P}\left[\arg\max_\kappa \left(\ln\rho_\kappa + \left(\boldsymbol{a}^k + \frac{1}{2}\boldsymbol{e}_\kappa\right)^\top \left(\Delta\boldsymbol{\Theta}^{-1} + \alpha\boldsymbol{\Pi}\right)^{-1}\boldsymbol{e}_\kappa\right) \neq k\right]. \tag{227}$$

If $\boldsymbol{\Theta} = \mathbf{I}_K$ and the clusters have same weights, $\rho_k \equiv 1/K \Leftrightarrow \boldsymbol{\Pi} = 1/K\mathbf{I}_K$, then $\eta_k \equiv \eta$ and

$$\varepsilon_g^{\text{BO}} = \mathbb{P}\left[\frac{1}{\eta} < \max_{\kappa\in[K-1]} Z_\kappa + Z\right], \tag{228}$$

that is the formula given in [20].

# E  Experiments with real data

In this Appendix we discuss the experiments of Section 3.3 with real data sets.

**Numerical details —** Consider a real data set $\{(\boldsymbol{x}^\nu, y^\nu)\}_{\nu=1}^{n_{\text{tot}}}$ with $n_{\text{tot}}$ samples which we assume are independent. As a pre-processing step we center, normalise and flatten the inputs $\boldsymbol{x}^\nu$ into $d$-dimensional vectors. For both the MNIST [61] and Fashion-MNIST [62] data sets used in the experiments we have normalised the inputs by 255, such that components $x_i^\nu \in [0, 1]$. In what follows we focus on binary classification tasks and encode the labels as $y^\nu \in \{-1, 1\}$. For example, for the MNIST and Fashion-MNIST data sets we have $d = 784$ and $n_{\text{tot}} = 7 \times 10^4$, and we split the inputs into two classes depending on the task of interest, e.g. odd vs. even digits and clothes vs. accessories items, respectively. Define the empirical distribution over the data set:

$$\hat{P}(\boldsymbol{x}, y) = \frac{1}{n_{\text{tot}}} \sum_{\nu=1}^{n_{\text{tot}}} \delta(\boldsymbol{x} - \boldsymbol{x}^\nu) \delta(y - y^\nu) \tag{229}$$

The question we want to answer is: how well can we approximate the learning curves $(\epsilon_g, \epsilon_t)$ on a given ERM classification task by approximating $\hat{P}$ with a Gaussian mixture distribution? To answer this question, we consider a Gaussian mixture distribution $P_2$ as defined in Eq. (1) with the same means and covariances as $\hat{P}$:

$$\hat{\boldsymbol{\mu}}_k = \frac{1}{n_{\text{tot}}} \sum_{\nu=1}^{n_{\text{tot}}} \boldsymbol{x}^\nu \, \mathbb{I}\left(\boldsymbol{x}^\nu \in \mathcal{C}_k\right), \qquad \hat{\boldsymbol{\Sigma}}_k = \frac{1}{n_{\text{tot}}} \sum_{\nu=1}^{n_{\text{tot}}} (\boldsymbol{x}^\nu - \boldsymbol{\mu}_k)(\boldsymbol{x}^\nu - \boldsymbol{\mu}_k)^\top \, \mathbb{I}\left(\boldsymbol{x}^\nu \in \mathcal{C}_k\right) \tag{230}$$

for $k \in \{+, -\}$ labelling the two clusters. Similarly, the class probabilities $\rho_k$ are also estimated from the full data set:

$$\hat{\rho}_k = \frac{1}{n_{\text{tot}}} \sum_{\nu=1}^{n_{\text{tot}}} \mathbb{I}\left(\boldsymbol{x}^\nu \in \mathcal{C}_k\right). \tag{231}$$

The parameters $(\hat{\boldsymbol{\mu}}_k, \hat{\boldsymbol{\Sigma}}_k, \hat{\rho}_k)$ completely characterise the approximating Gaussian mixture distribution $P_2$, and together with Theorem 1 can be used to compute the theoretical learning curves $(\epsilon_g, \epsilon_t)$ as in Fig. 5 of the main. Note that this discussion can be easily generalised to the case in which a non-linear feature map $\boldsymbol{\varphi} : \mathbb{R}^d \to \mathbb{R}^p$ is applied to the data prior to fitting. The only difference is that the empirical distribution $\hat{P}$ is defined over the features $\{(\boldsymbol{v}^\nu, y^\nu)\}_{\nu=1}^{n_{\text{tot}}}$ where $\boldsymbol{v}^\nu = \boldsymbol{\varphi}(\boldsymbol{x}^\nu)$, and the Gaussian mixture approximation $P_2$ is defined with respect to the empirical features distribution. Figure 6 of the main manuscript shows an example where a random feature map $\boldsymbol{v} = \text{erf}(\boldsymbol{F}\boldsymbol{x})$ with $\boldsymbol{F} \in \mathbb{R}^{p \times d}$ a random Gaussian projection applied to MNIST and fashion MNIST before the fitting with different ratios $\gamma = p/d$.

The theoretical learning curves are then compared with two sets of finite instance simulations. First, we simulate the learning problem on synthetic data sampled from the approximating Gaussian mixture distribution $P_2$, and the learning curves are computed by averaging over 10 instances of the problem. Second, we simulate the learning problem on the real data set. The real data set is split into training and test sets, and for a given sample complexity $\alpha = n/d$ we sub-sample $n = \alpha d$ points from the training set. The averaged learning curves are computed over different instances of the sub-sampling, with replacement.

**Discussion —** As expected, we find good agreement between theory and simulations with synthetic data drawn from the approximating Gaussian mixture distribution $P_2$, even for relatively small input dimensions (e.g. $d = 784$ for MNIST). Surprisingly, we have found that in many cases the Gaussian mixture is a good approximation to the real data curves, see Figs. 5 and 6 for examples of logistic regression on input space and with random features. Figure 7 shows an example where the feature map $\boldsymbol{\varphi}$ is given by removing the last layer of the following fully-connected 2-layer neural network pre-trained on the full MNIST odd vs. even data set:

```
Sequential(
  (0): Linear(in_features=784, out_features=784, bias=False)
  (1): ReLU()
  (2): Linear(in_features=784, out_features=1, bias=False)
  (3): Tanh()
)
```

with the training performed by minimising the square loss with the Adam optimiser and random initialisation. However, we have also found cases in which the approximation is not as sharp, see blue curves in Fig. 10. Understanding the factors determining the quality of the approximation in real data sets is an interesting question we expect to address in future work.

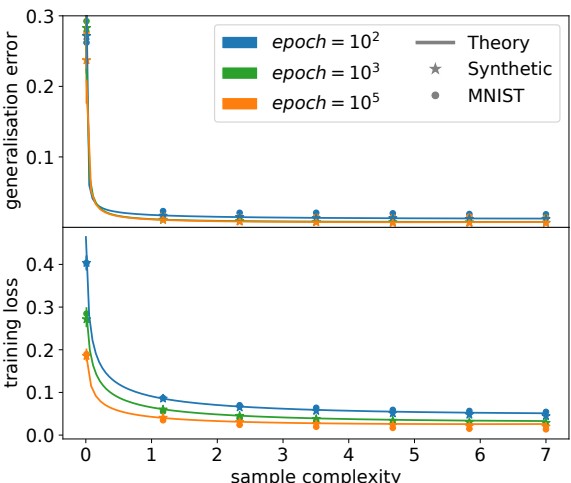

Figure 7: Generalisation error and training loss for logistic regression on MNIST with a feature map $\varphi$ obtained by training 2-layer fully connected neural network, with $\ell_2$ penalty and fixed $\lambda = 0.05$. The different curves show the performance at different stages of training.

**Multiclass vs. binary approximation –** In the cases previously discussed, we have considered a $K = 2$ cluster approximation $P_2$ to the empirical data distribution $\hat{P}$. However, the data sets considered here (MNIST and Fashion-MNIST) are originally composed of 10 classes, and therefore we should ask the question of whether a $K = 10$ cluster approximation $P_{10}$ where we fit the means and covariances of each original class is any different from the approximation studied above. In principle, these two approximations can have very different statistical properties. For instance, from Theorem 2 it follows that the generalisation and training errors of Gaussian mixtures only depend on the statistics of the local field $\lambda = \boldsymbol{W}\boldsymbol{x}$ conditioned on the labels, which in the binary setting considered here is $y \in \{+, -\}$. Conditioned on $y = \pm$, this local field is simply a Gaussian random variable under $P_2$, while it is a multi-modal random variable under $P_{10}$. Therefore, there is *a priori* no reason for these two approximations to give the same learning curves.

As an example, consider a $K = 4$ Gaussian mixture distribution with a common variance $\Sigma_k = \Delta \mathbf{I}_d$ and with means:

$$\boldsymbol{\mu}_1 = \boldsymbol{e}_1 + \boldsymbol{e}_2, \qquad \boldsymbol{\mu}_2 = \boldsymbol{e}_1 - \boldsymbol{e}_2, \qquad \boldsymbol{\mu}_3 = -\boldsymbol{e}_1 + \boldsymbol{e}_2, \qquad \boldsymbol{\mu}_4 = -\boldsymbol{e}_1 - \boldsymbol{e}_2 \qquad (232)$$

where $\boldsymbol{e}_i \in \mathbb{R}^d$ is the canonical basis vector of $\mathbb{R}^d$, with entries $e_{ij} = \delta_{ij}$. We consider two label assignments: a) a realisable case in which clusters 1 and 2 are assigned label $+1$, and clusters 3 and 4 are assigned $-1$ and b) a non-realisable case in which clusters 1 and 4 are assigned $+1$ and clusters 2 and 3 are assigned $-1$ (XOR function), see Fig. 8 (*top*) for an illustration. Now consider a dual $K = 2$ Gaussian mixture model with means and covariances $(\boldsymbol{\mu}_\pm, \boldsymbol{\Sigma}_\pm)$ chosen to match the class means and covariances of the $K = 4$ mixture, see Fig. 8 (*bottom*) for an illustration. In Fig. 9 we compare the learning curves of the $K = 4$ model with the $K = 2$ counterpart with matched class means and covariances. While in the realisable case $a)$ both have identical performance under the error bars, in the non-realisable case $b)$ the performance in are significantly different.

Indeed, a similar behaviour can be observed in the real data experiments. Fig. 10 compares the real learning curves of a MNIST 5v5 binary classification task (classifying five first digits vs. five last) with the two different Gaussian mixture approximations: $P_{10}$ where we fit the means and covariances of each individual cluster and $P_2$, where we fit only the class-wise means and covariances. While both approximations capture the high-level behaviour of the learning curves, $P_{10}$ is closer to the real learning curve than $P_2$.

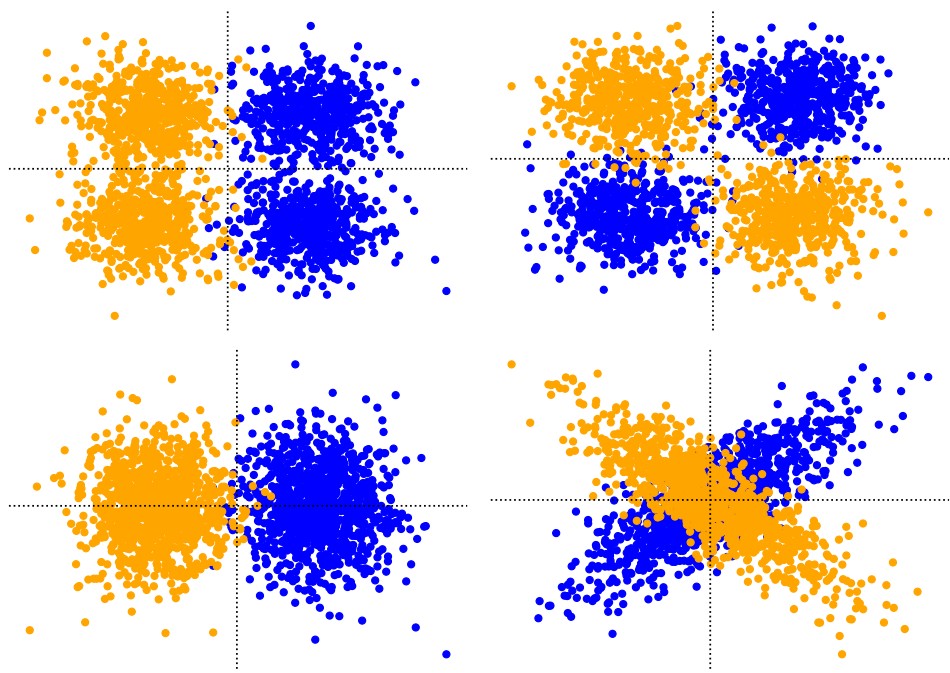

Figure 8: Two dimensional projection of the setting described in eq. (232). (**Left**) Realisable case, (**Right**) Non-realisable case (XOR function).

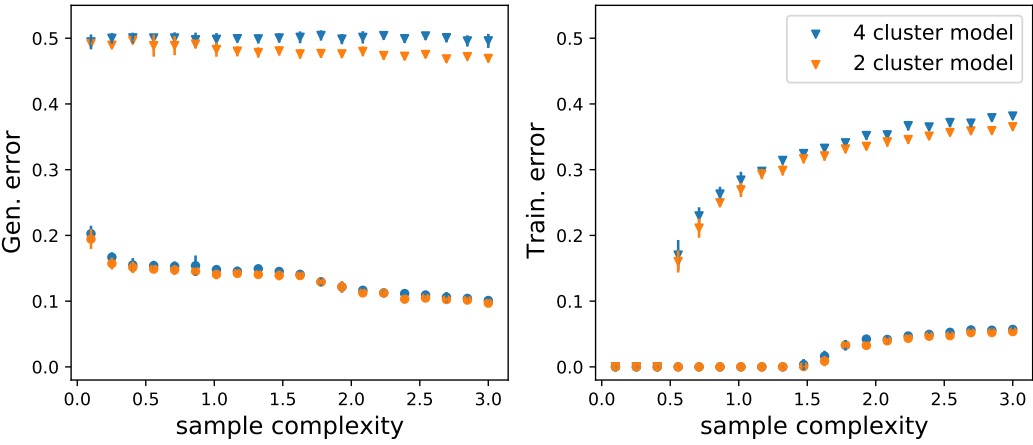

Figure 9: (**Left**) Generalisation and (**right**) training errors as a function of the sample complexity for logistic regression with $\ell_2$ penalty and $\lambda = 10^{-4}$ for the four models pictured in Fig. 8. Points denote the separable model (bottom curve), and triangles denote the non-realisable xor model (top curves). We have chosen a balanced scenario with $\Delta = 0.5$.

**Note on numerical instabilities —** When dealing with means and covariance matrices estimated from real data sets, we have observed that for small regularisation strength $\lambda \ll 1$ the self-consistent equations from Theorem 1 can develop spurious fixed points corresponding to negative values of the overlap parameters $q_\pm = \boldsymbol{W}^\top \boldsymbol{\Sigma}_\pm \boldsymbol{W}$ – which is clearly not possible since $\boldsymbol{\Sigma}_\pm$ is a positive-definite matrix. This is observed across different scenarios, and is independent of the choice of loss or the particular way the equations are solved. In fact, the minimum value of $\lambda$ below which the spurious fixed point develop seems to depend only on the conditioning number of the covariance matrices.

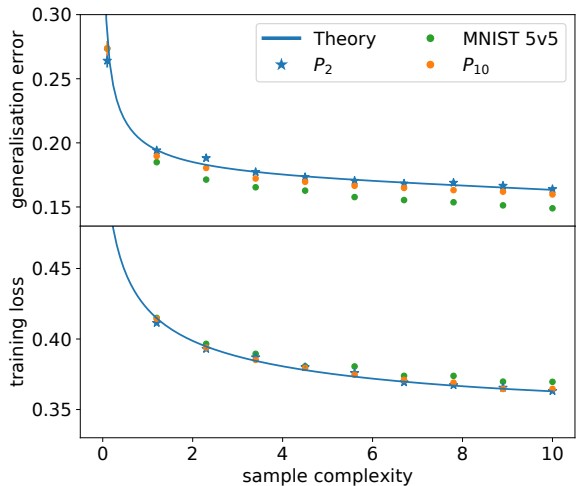

Figure 10: Generalisation error and training loss for logistic regression on the task of classifying $\{0, 1, 2, 3, 4\}$ vs $\{5, 6, 7, 8, 9\}$ digits of MNIST, as a function of the sample complexity for fixed $\ell_2$ penalty $\lambda = 0.1$. The blue curves show the 2-Gaussian cluster approximation $P_2$ (solid for theory, points for finite size simulations), while the orange points show the 10-Gaussian cluster approximation $P_{10}$, which lies systematically below. The green points denote simulations on the true data set.