# OpenReview forum: "Learning Gaussian Mixtures with Generalized Linear Models: Precise Asymptotics in High-dimensions"
_NeurIPS.cc/2021/Conference — NeurIPS 2021 Spotlight_

### Official Review · Reviewer_HUKh · 2021-07-11

**Rating:** 8
**Confidence:** 2

**Summary:**

This paper provides a general asymptotic characterization of the solution that comes from learning a mixture of K Gaussians under a supervised learning setting with arbitrary convex loss and regularization. The technique for proving the characterization is novel. Specifically, it constructs an approximate message passing sequence that admits state evolution equations, whose fixed point satisfies the optimality condition of the original problem. Equipped with the theoretical tools, the authors study both synthetic data (sparse Gaussian mixture and general K Gaussian mixtures) and real datasets to further understand (1) $\ell_{1}$ and $\ell_{2}$ regularization for sparse mixture models; (2) separability transition; and (3) quality of Gaussian mixtures as surrogate model. Interestingly, the experiments exhibit a high agreement between the theoretical and numerical values.

**Limitations And Societal Impact:**

How easy is it to construct an AMP sequence for a learning problem? Some comments on flexibility of the technique will be helpful. Some motivations of the AMP method and high level ideas to apply AMP method will also inspire other researchers in future, who might not be familiar with the technique.

**Main Review:**

Originality: This work utilizes the technique from the field of statistical physics to understand high dimensional inference problem in statistical learning. Specifically, an approximating message passing sequence is constructed so that the fixed point of that sequence coincides with the optimal solution of the original convex optimization problem with regularization. Using the existing theory for approximate message passing, the asymptotic value of the convex loss at the optimal solution can be derived. This method generalizes the previous work that derives the asymptotic result under some special setting for covariance structure and convex loss.

Quality: The paper is technically strong. Due to the space, the authors have to defer the proof to the supplementary material. The process to construct the AMP sequence such that the fixed point satisfies the optimality condition has been very carefully written.

Clarity: The submission is nicely written. A few questions to make sure I understand things correctly: (1) In the objective function, why does it not have $\frac{1}{n}$ factor for the loss term? (2) what is the choice of $d$ in the experiments for synthetic data?  How would choice of $d$ affect the gap between theoretical value and the experimental value? (3) The authors have mentioned that the current asymptotic result can be adapted to the finite sample setting, does that include both finite $n$ and $d$? (4) What the result might be if $n$ grows much faster than $d$, in the sense that $\frac{n}{d}$ is not a constant?

Significance: I think the result is very important. In particular, the technique of constructing approximate message passing algorithm can be potentially useful to other statistical learning setting.

**Time Spent Reviewing:**

10 hours

---

> ### Author Response · Authors · 2021-08-09
> **Reply to Reviewer HUKh**
>
> We thank the reviewer for the positive comments and for the interest in our work. We address each of her/his questions below.
>
> 1. We adopt as convention a risk function without the $\frac{1}{n}$ prefactor, so that $\mathcal R=O(n)$. The $\frac{1}{n}$ factor is however reintroduced when looking at the average training loss, i.e., in Eq. (4), which is the object of interest in the proof.
>
>
> 2. In the experiments for synthetic data, the value of the dimension is $d=1000$ for Section 2-3 and $d=784$ for Section 4. In particular, in Section 4 the covariances adopted in the synthetic experiments are the ones obtained by the real datasets and inherit their dimensionality. To exemplify how fast the gap between the numerical experiments and the theoretical predictions closes, in the Appendix we will add two new sets of plots about the classification task of $K=3$ Gaussian clouds with identical diagonal covariances, for both quadratic and logistic losses. The figures will show, at fixed $\alpha>1$, the differences between the training/test errors $\epsilon^{\rm exp}$ obtained by averaging over $2\cdot10^4$ experiments and the corresponding theoretical quantities $\epsilon^{\rm th}$ as function of $d$. In all cases, such differences $\Delta \epsilon=|\epsilon^{\rm exp}-\epsilon^{\rm th}|$ exhibit a fast decay, starting from $\frac{\Delta\epsilon}{\epsilon^{\rm th}}\simeq 10^{-1}$ for $d=10$ to $\frac{\Delta\epsilon}{\epsilon^{\rm th}}\sim 10^{-3}$ for $d=200$. We are happy to include these plots on OpenReview as well if it is technically possible in the rebuttal.
>
>
> 3. The existing finite sample size analysis (see [55]) of simpler instances of approximate message passing iterations are valid for $n$ and $d$ in the proportional regime. For separable loss and penalty functions, the concentration towards the asymptotic values is exponential in $d$, and in simple cases (typically i.i.d. Gaussian GLMs) a few hundreds is enough to obtain clean curves (to give a few examples, see Refs. [4,7,8,10,12,14,20,31,49,51]).
>
>
> 4. If $n$ grows much faster than $d$, we have a lot of samples to estimate a mixture of reasonable dimension. Thus we would estimate the centroids and covariances without error and the problem reduces to the one with known centroids and covariances which is much simpler. In order to make the problem non-trivial we would also need to make the centroids and covariances depend on $d$, which would be an interesting extension.
>
>
> 5. (comment on AMP) The construction of an AMP sequence relies on intuition from previous literature where AMP is derived based on graphical models. Our construction for the two-step iteration (24-25) starts with AMP for simpler instances of GLMs (i.i.d. Gaussian ones, see e.g.[53,54]), and then extends the construction to the Gaussian mixture case in the proof using the optimality and fixed point conditions.
>
>     There are many forms of AMP iterations, notably including many more than two steps corresponding to inference in trained deep neural networks (see e.g. [35, 38]). There is (yet) no generic rule to determine what AMP iteration fixed point should be matched with the optimality condition of the problem to get the best sequence, and it has to be constructed case-by-case. Notably, this is not known for a multi-layer perceptron or even a single but extensive hidden-layer with learned weights. In brief, a generic method to derive AMP iterations is still missing, and we agree with the referee that this is an important research direction. We will make an effort to add comments related to this discussion, as well as more motivation and high-level intuition to the paper.

---

### Official Review · Reviewer_DjLU · 2021-07-11

**Rating:** 7
**Confidence:** 3

**Summary:**

The paper provides precise asymptotic solutions for learning Gaussian mixtures using generalized linear models. This work generalizes and extends previous works in this direction by allowing any means/covariance of each Gaussian component, any convex loss and any regularization. Beyond the theoretical analysis, the paper also studied several examples, including comparing Lasso and Ridge regression, comparing the theoretical vs practical learning curves on real data.

**Limitations And Societal Impact:**

The paper addressed the limitations of the theory, i.e., the assumptions. Since this is theoretical paper, the societal impact is minimal.

**Main Review:**

The main contribution of this paper is a generalization of existing literature on asymptotic solutions for Gaussian mixtures. The results are original and novel. The paper is well-written and the theorem is clearly presented. The experimental result on real data looks surprisingly good to me.

My main concern is the complexity for solving the fixed point of Eq. (8). As far as I read, the main paper doesn't mention how to solve it. Even for the simplified setting, the solution to Eq (11) doesn't seem to be straight forward to me. The appendix seems to imply that Monte Carlo approximation and iterative algorithms will be involved in the computation. Note that these approaches may lead to some error in the asymptotic results.

Overall, I think this paper will be interesting to the theory community especially for those who are interested in high-dimensional asymptotic solutions.

**Time Spent Reviewing:**

3

---

> ### Author Response · Authors · 2021-08-09
> **Reply to Reviewer DjLU**
>
> We thank the referee for the appreciation of our work. We were also positively surprised by the good agreement on experimental data, and we think this makes the model even more interesting.
>
> About the comment on numerically solving the iterative equations, for the sake of completeness and transparency we will release a GitHub repository with a polished version of the code we used to solve the equations. Additionally, we welcome the referee suggestion and will clarify these points in the revised paper, adding a comment on the numerical procedure to solve the equations in the main text along the following lines:
>
> All the theoretical curves in the manuscript are obtained by iterating the fixed points in Eq. (8) in a self-consistent way, until a certain convergence criterion is reached (Frobenius norm of the variation of the order parameters between two subsequent iterations smaller than some tolerance $\epsilon\leq 10^{-5}$). In the simplest setting, i.e., the one discussed in Corollary 3, the update of $(\mathbf m_k,\mathbf Q_k,\mathbf V_k)$ is performed explicitly using Eq. (11), where $\mathbb E_{\vec\sigma,
> \vec\mu}[\bullet]$ is a shorthand for the sum over the eigenvalues and eigenvectors of the assigned covariance matrices. The update of $(\hat{\mathbf m}_k,\hat{\mathbf Q}_k,\hat{\mathbf V}_k)$ (right hand side of Eq. (8)) is more involved, as it requires the computation of the proximal $\vec f_k$ followed by a Gaussian average over $\vec\xi_k\in\mathbb R^K$, as in Eq. (7). This integral is performed with Monte Carlo, i.e., by solving the equation for $\vec f_k$ for a large number ($10^4-10^5$) of instances of $\vec\xi_k$ and averaging the solution. We remark that for the square loss $\vec f_k$ can be computed analytically and the integration can be performed explicitly, highly simplifying the fixed-point equations (see Eq. (169) in the Supplementary material). We agree with the referee that, whilst the convergence to the solution by iterative methods are guaranteed by the uniqueness of the fixed point, the precision of our theoretical predictions are in principle affected by the size of the Monte Carlo pool in performing the numerical integration. However, we have found that in practice fluctuations were small enough to give results that have much larger precision than direct numerical experiments.

---

### Official Review · Reviewer_Cx7X · 2021-07-16

**Rating:** 6
**Confidence:** 3

**Summary:**

The paper shows the exact asymptotic behavior of the GLM/1-layer-NN solution for features following a mixture of K Gaussian distribution.

**Limitations And Societal Impact:**

This is a theoretical paper.

**Main Review:**

My comments on the paper are mixed.
1) If we only look at a classification problem on a mixture of Gaussian distribution, to pick a convex approach, it seems natural to consider a linear model with a convex loss function. But we can also utilize a non-convex approach to first estimate the Gaussian parameters and then classify the sample which probably gives better results.
2) If we look at it as learning a neural network with a mixture of Gaussian as input, one layer is too simple to consider. If we think deep NN without the output layer produces a mixture of Gaussian and we just want to know the asymptotic of the final loss function, one layer seems to be fine.
3) If we look at the contribution to the AMP literature, I think it is great that we can extend the framework to a mixture of Gaussians. On the other hand, the result can be more general if it can extend to mixtures of concentrated distributions. The authors have mentioned that they expect the result should be similar and there are already existing results that extend Gaussian distribution to sub-Gaussian distribution. It looks like with some effort, the result in this paper can be more general.

In summary, the paper has some incremental contributions.

**Time Spent Reviewing:**

6

---

> ### Author Response · Authors · 2021-08-09
> **Reply to Reviewer Cx7X**
>
> We thank the reviewer for her/his very pertinent questions.
>
> 1. Indeed, there are many ways to fit a mixture of Gaussians, the lowest error can be obtained using directly a mixture of Gaussian and estimation based on the posterior distribution. This is indeed something we present in the appendix, section D, and is plotted for comparison with the ERM estimator in Fig. 3 (dashed lines in left and right panel).
>
>     However, as the referee anticipated, our main motivation was to understand how simple and tractable neural networks trained by minimising a loss function on the training data perform on this task.
>
>
> 2. Although we agree with the referee that a one-layer neural network trained on a general Gaussian mixture distribution is a simple setting, the solution we provide was not given in existing literature and hence it is a meaningful starting point. In addition, as discussed in the paper, a couple of popular machine learning problems are one-layer neural networks in disguise, e.g. kernel methods, deep neural networks in the lazy regime (Neural Tangent Kernels), random features model, etc.
>
>     We completely agree with the second part of the remark about the last layer of deep neural networks. Indeed, as mentioned in the paper, the neural collapse phenomenon  provides an additional motivation for the setting considered in our work.
>
>
> 3.  We completely agree that it is relevant to extend our AMP approach to mixtures of concentrated vectors. Indeed, we conjecture (section Universality, line 208) that our formula should apply to this case as well. However, proving this conjecture is in our view a rather challenging question (see, e.g., refs. [56,57]) requiring an extensive mathematical effort that goes well beyond the scope of the present paper. But we thank the referee for the suggestion and we hope to provide a proof in the near future.
>
> About the remark that our work is incremental, the AMP extension presented in the paper solves several challenges and shortcomings of previous proof techniques. In particular, it required using a considerable amount of non-trivial techniques including multi-layer AMP and spatial coupling. It extends the reach of previous proofs. As a result, we are able to take random design models close to very realistic settings. For all these reasons, we do not think the contributions of our paper are incremental.

---

### Decision · Program_Chairs · 2021-09-27

**Decision:**

Accept (Spotlight)

**Comment:**

This paper studies the problem of learning a mixture of k gaussians by performing empirical risk minimization on a convex loss with a regularization penalty. The main technical contribution of this work is an asymptotic characterization of the solution, through statistical physics techniques. The conceptual novelty is that they can work with mixtures of gaussians, and it seems promising that some of these ideas might ultimately be useful in even more general settings, such as mixtures of concentrated distributions. They also give some appealing applications to (1) the case with sparse means, where they study the efficiency of $\ell_1$ vs. $\ell_2$ penalties and (2) max margin multiclass classification, where they identify a phase transition in the maximum likelihood estimator. They also give some interesting empirical applications that work beyond synthetic data.